# Drug reinforcement impairs cognitive flexibility by inhibiting striatal cholinergic neurons

Himanshu Gangal[1,2], Xueyi Xie[1], Zhenbo Huang[1], Yifeng Cheng [1], Xuehua Wang[1], Jiayi Lu[1], Xiaowen Zhuang[1], Amanda Essoh[1], Yufei Huang[1,2], Ruifeng Chen[1,3], Laura N. Smith [1,2], Rachel J. Smith[2,4] & Jun Wang [1,2,3] ✉

Addictive substance use impairs cognitive flexibility, with unclear underlying mechanisms. The reinforcement of substance use is mediated by the striatal direct-pathway medium spiny neurons (dMSNs) that project to the substantia nigra pars reticulata (SNr). Cognitive flexibility is mediated by striatal cholinergic interneurons (CINs), which receive extensive striatal inhibition. Here, we hypothesized that increased dMSN activity induced by substance use inhibits CINs, reducing cognitive flexibility. We found that cocaine administration in rodents caused long-lasting potentiation of local inhibitory dMSN-to-CIN transmission and decreased CIN firing in the dorsomedial striatum (DMS), a brain region critical for cognitive flexibility. Moreover, chemogenetic and time-locked optogenetic inhibition of DMS CINs suppressed flexibility of goal-directed behavior in instrumental reversal learning tasks. Notably, rabies-mediated tracing and physiological studies showed that SNr-projecting dMSNs, which mediate reinforcement, sent axonal collaterals to inhibit DMS CINs, which mediate flexibility. Our findings demonstrate that the local inhibitory dMSN-to-CIN circuit mediates the reinforcement-induced deficits in cognitive flexibility.

Natural rewards and addictive substances, such as cocaine and alcohol, increase dopaminergic activity in the striatum, which governs over 90% of voluntary behaviors[1,2]. This elevation in dopamine levels facilitates corticostriatal synaptic plasticity in direct-pathway medium spiny neurons (dMSNs)[3–8], ultimately driving an animal to repeat actions that previously resulted in rewards and promoting reinforcement learning[5,9,10]. The projection of dMSNs to the substantia nigra pars reticulata (SNr) forms the direct pathway of the basal ganglia[11–13], which plays a crucial role in mediating reward-induced reinforcement[14]. Drug-free optogenetic studies have demonstrated that intracranial self-manipulation of dMSN→SNr activity alone is sufficient to induce reinforcement[15,16], confirming the direct pathway's

role in this behavior. Thus, positive rewards generally increase dMSN activity, promoting the repeated use of these rewards, with addictive substances causing more persistent neuroadaptations compared to natural rewards[17].

Reinforcement of drug (e.g., cocaine) and alcohol use has also been linked to reduced cognitive flexibility, contributing to the compulsive use of addictive substances[9,18–23]. However, the neurocircuitry involved in drug-induced inflexibility remains elusive. Intriguingly, there is also evidence indicating that excessive or chronic reinforcement of natural rewards can lead to a decline in decision-making and flexibility[24–26]. Cognitive flexibility enables adaptation to environmental changes to obtain rewards or avoid punishment[27]. In animal

[1]Department of Neuroscience and Experimental Therapeutics, School of Medicine, Texas A&M University Health Science Center, Bryan, TX 77807, USA. [2]Institute for Neuroscience, Texas A&M University, College Station, TX 77843, USA. [3]Interdisciplinary Faculty of Toxicology, Texas A&M University, College Station, TX 77843, USA. [4]Department of Psychological and Brain Sciences, Texas A&M University, College Station, TX 77843, USA. ✉e-mail: jwang188@tamu.edu

studies, cognitive flexibility can be measured using reversal learning tasks, where animals are trained on a set of action-outcome associations that are then reversed; animals are considered inflexible if they fail to learn the reversed associations[27]. The ability to modify action-outcome associations is critical for maintaining adaptive goal-directed behavior in a dynamic environment, which is a fundamental aspect of cognitive flexibility[28]. Substantial evidence suggests that an increase in striatal acetylcholine (ACh), released by cholinergic interneurons on thalamic stimulation, is necessary for reversal learning[27,29-31]. Furthermore, a recent study demonstrated an association between compulsive cocaine self-administration and reduced the activity of cholinergic interneurons (CINs)[32]. Considering these findings, we hypothesize that drug-induced cognitive inflexibility could be attributed to CIN dysfunction. Interestingly, CINs receive approximately 60% of total presynaptic inputs from GABAergic sources[33]. Additionally, striatal CINs receive extensive local afferents from the striatum itself[34], which is primarily composed of GABAergic dMSNs and indirect pathway MSNs (iMSNs)[11-13]. Thus, CINs are likely to receive inhibitory inputs from MSNs in the striatum. Given that positive reinforcers enhance striatal dMSN activity, the downstream inhibitory dMSN→CIN transmission could play a crucial role in reducing CIN activity mediating drug-induced suppression of cognitive flexibility.

Our findings indicate that cocaine exposure increased GABAergic inputs from dMSNs to CINs and suppressed CIN firing in the DMS, a brain region that regulates goal-directed behaviors and cognitive flexibility[30,31]. Chemogenetic or optogenetic inhibition of CINs reproduced cocaine-induced reductions in cognitive flexibility. Notably, SNr-projecting dMSNs in the direct pathway that mediate reinforcement also sent collaterals to inhibit CINs. Our data demonstrate that the inhibitory dMSN→CIN connection mediates the reinforcer-induced reduction in cognitive flexibility.

## Results

### Cocaine exposure impairs cognitive flexibility

To explore the impact of cocaine administration on cognitive flexibility, we intraperitoneally injected wild-type (WT) rats with cocaine or saline for seven days[35] (Fig. 1a). Following each injection, the rats exhibited hyperlocomotion (Fig. 1b; $F_{(1,17)} = 22.87$). We then trained the animals on a two-action (A)-outcome (O) reversal learning task for 11 sessions, wherein pressing the left lever (A1) resulted in the delivery of sucrose solution (O1) to a central magazine (A1→O1), and pressing the right lever (A2) dispensed food pellets (O2) in an adjacent magazine (A2→O2) (Fig. 1c)[30,31].

After the initial training, a devaluation test was conducted for two days to assess whether the rats had learnt the correct action-outcome contingencies (Fig. 1c, right). The rats then underwent reversal training for four sessions, followed by a second devaluation test (Fig. 1d). During reversal training, the action-outcome contingencies of the task were reversed, i.e., pressing the left lever (A1) led to food pellet delivery (A1→O2), while pressing the right lever (A2) led to the delivery of sucrose solution (A2→O1). Cocaine- and saline-administered rats exhibited no significant difference in lever-press rates during initial training (Fig. 1e; $F_{(1,17)} = 0.693$, $p = 0.417$), and both groups were sensitive to outcome devaluation (Fig. 1f). There was a main effect of devaluation ($F_{(1,16)} = 67.376$, $p < 0.001$) but no group x devaluation interaction ($F_{(1,16)} = 0.386$, $p = 0.543$). Additionally, lever-press rates during reversal training did not differ between groups (Fig. 1g; $F_{(1,17)} = 0.247$, $p = 0.626$). However, the saline, but not cocaine, group was sensitive to outcome devaluation after reversal training (Fig. 1h). Statistical analysis revealed a main effect of devaluation ($F_{(1,16)} = 29.762$, $p < 0.001$) and critically, a significant group x devaluation interaction ($F_{(1,16)} = 7.368$, $p = 0.015$). Further simple effects test revealed that, whereas the saline group pressed the valued lever significantly more than the devalued lever (Fig. 1h left; $F_{(1,16)} = 35.545$), the cocaine group did not (Fig. 1h right; $F_{(1,16)} = 3.381$, $p = 0.085$). These

data suggest that cocaine impaired the learning of new action-outcome contingencies.

To determine whether cocaine also induces reversal learning deficits in other species and reward types, we trained cocaine-exposed mice on the reversal learning task with purified food (O1) and grain-based food (O2) as the two available rewards (Fig. 1i; Supplementary Fig. 1a, $F_{(1,23)} = 35.38$). Both groups exhibited similar performance during initial training (Supplementary Fig. 1b; $F_{(1,22)} = 1.67$, $p = 0.21$) and were both sensitive to outcome devaluation (Fig. 1j). There was a main effect of devaluation ($F_{(1,21)} = 19.780$, $p < 0.001$) but no group x devaluation interaction ($F_{(1,21)} = 0.0436$, $p = 0.837$). Additionally, there was no difference between groups during reversal training (Supplementary Fig. 1c; $F_{(1,22)} = 0.918$, $p = 0.348$). However, the saline, but not the cocaine group, was sensitive to outcome devaluation after reversal training (Fig. 1k). Statistical analysis revealed a significant group x devaluation interaction ($F_{(1,21)} = 6.665$, $p = 0.017$). Further simple effects test revealed that while the saline group had significantly higher valued versus devalued lever presses (Fig. 1k left; $F_{(1,21)} = 4.949$), the cocaine group did not (Fig. 1k right; $F_{(1,21)} = 1.987$, $p = 0.173$). These results indicate that although cocaine administration does not alter the initial learning of action-outcome contingencies, it impairs reversal learning and reduces cognitive flexibility.

### Cocaine enhances the inhibition of CINs

Cognitive flexibility is modulated by CINs in the dorsomedial striatum (DMS)[30,31]. We investigated the effects of cocaine treatment on DMS CIN activity in choline acetyltransferase (ChAT)-eGFP mice (Fig. 2a-c; Supplementary Fig. 2a, $F_{(1,8)} = 82.376$). DMS slices were prepared two days after the final cocaine injection, and cell-attached recordings were conducted in GFP-positive CINs. We found that the spontaneous firing frequency of CINs was significantly lower in cocaine-administered mice than in saline controls (Fig. 2d, e; $t_{(19)} = 2.30$).

Since reversal learning (Fig. 1d) was tested approximately 24 days after the last cocaine injection, we also measured CIN activity at this time point. We discovered that the spontaneous firing frequency remained lower in the cocaine group (Fig. 2f; $t_{(19)} = 2.42$), suggesting that cocaine induces long-lasting inhibition of CIN activity. Cocaine-mediated suppression of DMS CIN activity could result from an increase in presynaptic inhibitory inputs[33,34]. Therefore, we examined whether repeated cocaine injections altered GABAergic inputs onto DMS CINs. Spontaneous inhibitory postsynaptic currents (sIPSCs) in CINs were recorded two days after the final injection (Fig. 2g). We found that sIPSC frequencies, but not amplitudes, were greater in cocaine-administered mice than in saline-injected controls (frequency: Fig. 2h, $t_{(28)} = -2.23$; amplitude: Fig. 2i, $t_{(26)} = 0.85$, $p = 0.405$). Additionally, cocaine decreased the paired-pulse ratio of electrically evoked IPSCs in CINs at multiple inter-stimulus intervals (Fig. 2j, k; $F_{(1,44)} = 14.27$), suggesting that cocaine increases presynaptic GABA release onto DMS CINs.

To model voluntary cocaine-seeking behavior, we trained ChAT-eGFP mice with intravenous self-administration (IVSA) of cocaine (Fig. 2l; Supplementary Fig. 2b, $F_{(1,18)} = 15.179$)[17,36]. The control group was yoked to receive saline infusions. Since we recorded DMS CIN firing two and 24 days after the last cocaine i.p. injection (Fig. 2e, f), we chose to record at a time point falling within this period for cocaine IVSA. Ten days after the final IVSA session, we found that the spontaneous firing frequency of DMS CINs was significantly lower in the cocaine IVSA group than in the yoked saline controls (Fig. 2m; $t_{(24)} = 3.31$). Additionally, we discovered that cocaine IVSA increased the sIPSC frequency, but not amplitude, in CINs (frequency: Fig. 2n, $t_{(36)} = -2.94$; amplitude: Fig. 2o, $t_{(34)} = -1.37$, $p = 0.178$). The amplitude of electrically evoked IPSCs was significantly greater in the cocaine group (Fig. 2p, q; $F_{(1,16)} = 18.02$). Consistently, cocaine IVSA also decreased the paired-pulse ratio of these IPSCs (Fig. 2r; $t_{(16)} = 3.62$), indicating that cocaine increased the probability of presynaptic GABA release onto CINs[37]. Taken together, these results suggest that both

experimenter-delivered and voluntary cocaine administration inhibit DMS CIN activity and potentiate GABAergic inputs onto DMS CINs.

## CINs receive striatal inputs primarily from dMSNs

Having shown that cocaine potentiated presynaptic GABAergic inputs onto DMS CINs, we next examined the source of these inputs. The majority of inputs to CINs originate from the striatum itself[34], which contains two equal populations of principal GABAergic MSNs that express either dopamine D1 receptors (D1Rs) or D2Rs[11–13]. These are respectively located in the direct and indirect pathways of the basal ganglia and are thus called dMSNs or iMSNs[11–13]. To compare dMSN versus iMSN inputs onto CINs, we used rabies-mediated monosynaptic retrograde tracing[38]. Helper viruses (AAV-DIO-TVA-mCherry and AAV-DIO-RG) were infused into the DMS of ChAT-Cre;D1-tdTomato mice,

where CINs expressed Cre recombinase, and dMSNs expressed tdTomato (Fig. 3a). Three weeks later, rabies-GFP was infused at the same location in the DMS (Fig. 3a right, b), leading to selective rabies-GFP infection of TVA-expressing CINs. RG allowed the transfer of rabies-GFP from these CINs to their presynaptic neurons (Fig. 3c). These "starter CINs" expressed mCherry and GFP. dMSNs presynaptic to the starter CINs contained tdTomato and GFP (Fig. 3d–h). To differentiate starter CINs from presynaptic dMSNs, we performed ChAT immunostaining to label all CINs using the far-red Alexa Fluor 647 (Fig. 3g). Since over 90% of striatal neurons are either dMSNs or iMSNs[11–13], we assumed that most of the "GFP only" striatal neurons were putative presynaptic iMSNs. We discovered a higher percentage of dMSNs than putative iMSNs presynaptic to the starter CINs (Fig. 3i; $t_{(3)}$ = 3.643). Additionally, we found a few CINs (co-labeled with GFP and far-red) that formed monosynaptic

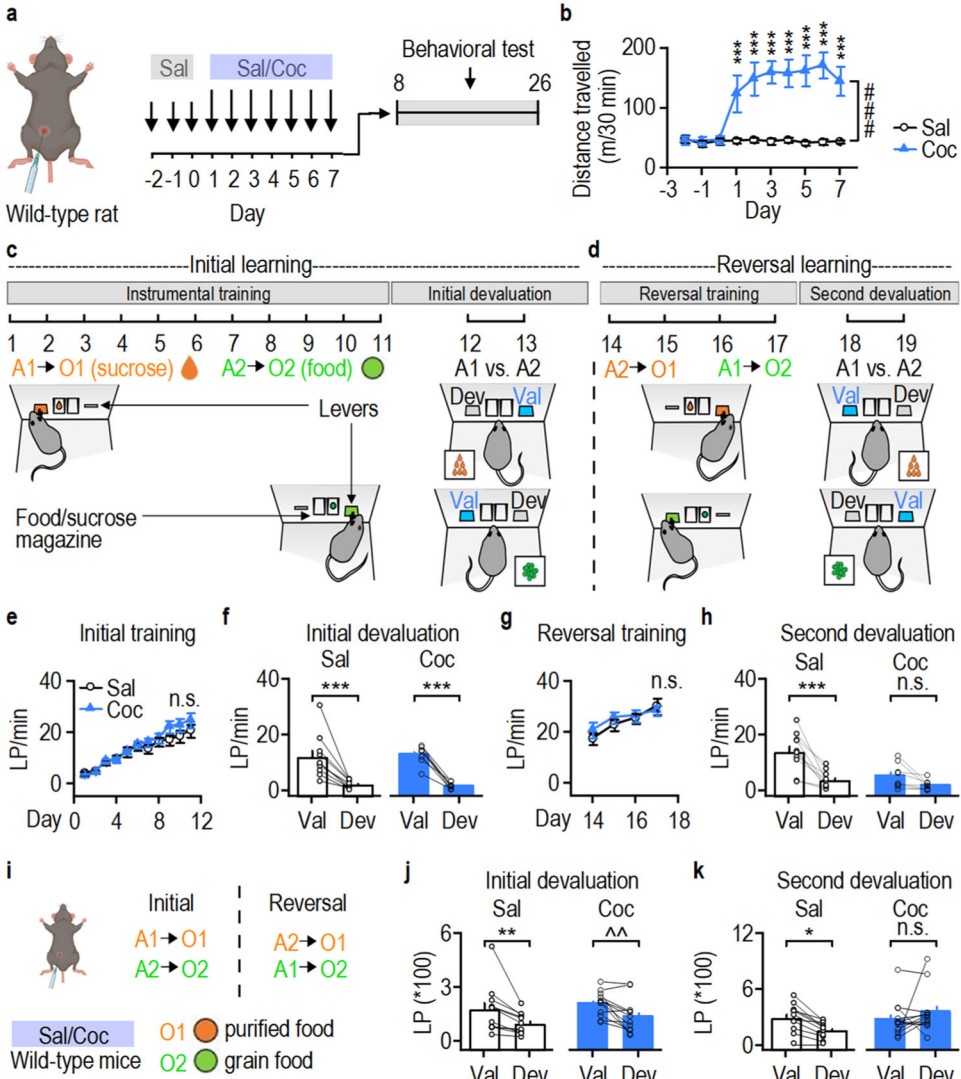

**Fig. 1 | Cocaine administration compromises reversal learning. a** Schematic of intraperitoneal (i.p.) injections of saline (Sal) or cocaine (Coc; 15 mg/kg) in wild-type rats. **b** Repeated cocaine injections caused hyperlocomotion in rats; ***$p < 0.001$ versus Sal, ###$p < 0.001$. **c** Initial instrumental training of action (A)-outcome (O) contingency: rats were trained to press the left lever (A1) to receive sucrose solution in a central magazine (O1; A1→O1) and the right lever (A2) to receive food pellets in an adjacent magazine (O2; A2→O2). The establishment of A-O contingencies was assessed via a devaluation test. **d** Reversal training (A2→O1, A1→O2) and 2nd devaluation. **e** Cocaine did not alter lever press rate during initial training. **f** Saline (left) and cocaine (right) groups were sensitive to outcome devaluation; ***$p < 0.001$. **g** Cocaine did not alter the lever-press rate during

reversal training. **h** The saline, but not cocaine, group was sensitive to outcome devaluation after reversal training; ***$p < 0.001$. **i** Schematic of i.p. injections of saline or cocaine (15 mg/kg) in wild-type mice and subsequent initial and reversal training followed by devaluation tests. **j** Saline and cocaine groups were sensitive to outcome devaluation after initial training; **$p = 0.004$, ^^$p = 0.006$. **k** The saline, but not cocaine, group was sensitive to outcome devaluation after reversal training; *$p = 0.037$. Val valued, Dev devalued, LP lever presses, n.s (not significant; **e, g, h, k**). Two-way RM ANOVA followed by Tukey *post-hoc* test (**b, e, g**), mixed model ANOVA followed by simple effects test (**f, h, j, k**). $n = 10$ (**a–h** Sal), 8 (**a–h** Coc), 11 (**i–k** Sal), 12 (**i–k** Coc). Data are presented as mean values ± SEM. Source data are provided as a Source Data file.

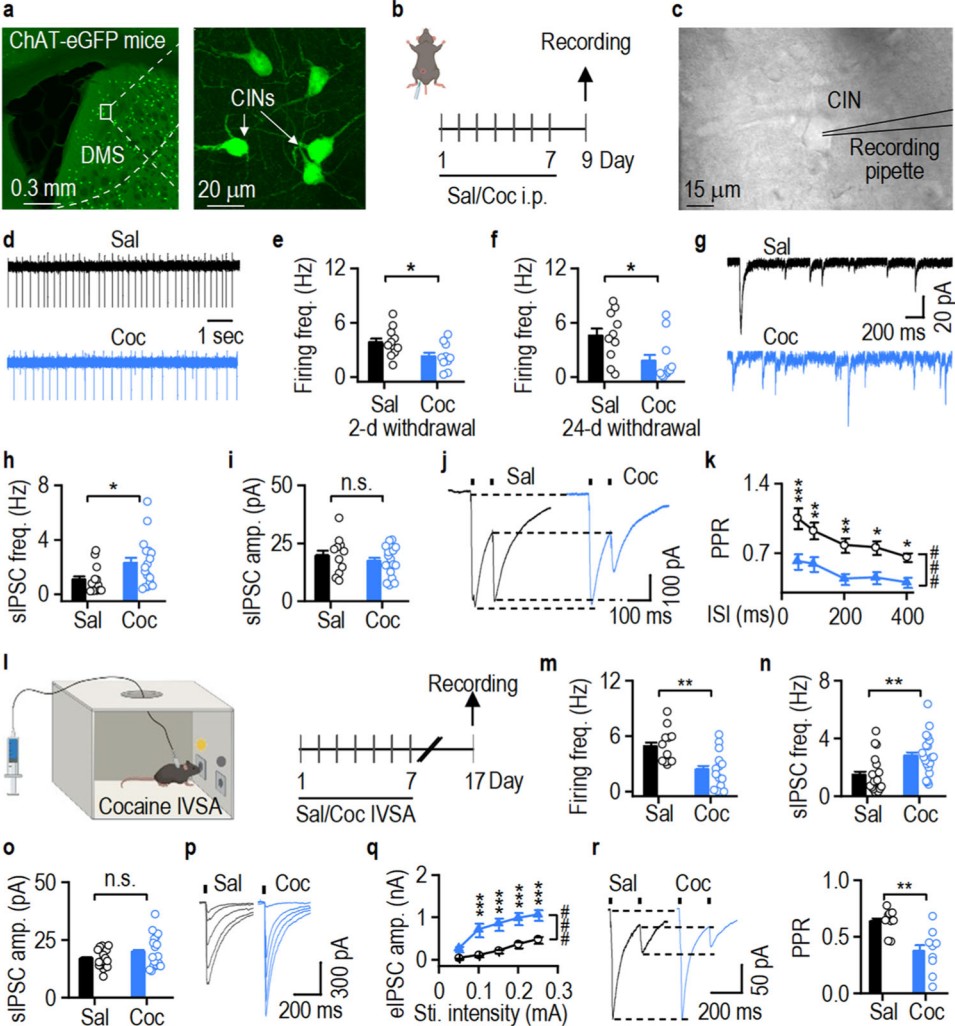

**Fig. 2 | Cocaine exposure potentiates GABAergic inputs onto DMS CINs.**
**a** Confocal images of cholinergic interneurons (CINs) in the dorsomedial striatum (DMS) in a ChAT-eGFP mouse. **b–e** Repeated cocaine (Coc, blue) but not saline (Sal, black) intraperitoneal (i.p.) injections reduced the spontaneous firing frequency (freq.) of DMS CINs 2 d after the last injection; *$p = 0.0328$. **f** Cocaine-induced inhibition of DMS CIN activity persisted 24 d after the last injection; *$p = 0.0257$. **g–i**, Sample spontaneous IPSCs (sIPSCs) traces (**g**). Cocaine increased the frequency (**h**; *$p = 0.0337$), but not amplitude (amp.) (**i**), of sIPSCs in DMS CINs. **j, k** Cocaine decreased paired-pulse ratios (PPR) of electrically evoked IPSCs (eIPSCs) in CINs; *$p < 0.05$, **$p < 0.01$, ***$p < 0.001$ versus Coc; ###$p < 0.001$. **l** ChAT-eGFP mice were trained with cocaine intravenous self-administration (IVSA) for 7 d;

DMS CINs were recorded 10 d after the last training session. **m**, Cocaine IVSA inhibited DMS CIN firing 10 d after the last exposure; **$p = 0.003$. **n, o** Cocaine IVSA increased the frequency (**n**; **$p = 0.006$), but not amplitude (**o**), of sIPSCs. **p, q** Cocaine IVSA potentiated eIPSCs in DMS CINs; ***$p < 0.001$ versus Sal, ###$p < 0.001$. **r** Cocaine IVSA decreased the eIPSC PPR in DMS CINs; **$p = 0.002$. Sti.: Stimulation, n.s. (not significant; **i**, **o**). Unpaired $t$ test (**e**, **f**, **h**, **i**, **m**, **n**, **o**, **r**), two-way RM ANOVA followed by Tukey *post-hoc* test (**k**, **q**). $n = 11$ neurons from 3 mice (11/3) (**e**, Sal), 10/3 (**e**, Coc; **f**, Sal), 11/4 (**f**, Coc), 18/4 (**h**, **i**; Sal), 20/4 (**h**, **i**; Coc), 26/3 (**k**, Sal), 23/3 (**k**, Coc), 12/4 (**m**, Sal), 14/4 (**m**, Coc), 13/4 (**n**, **o**; Sal), 17/4 (**n**, **o**; Coc), 9/4 (**q**, Sal & Coc; **r**, Sal & Coc). Data are presented as mean values ± SEM. Source data are provided as a Source Data file.

connections with the starter CINs[39,40]. These data suggest that CINs receive anatomic striatal inputs that mainly originate from dMSNs.

The dMSN→SNr pathway mediates the reinforcing effects of all rewards[14,16]. Consequently, we investigated whether SNr-projecting dMSNs also send collaterals to CINs. To test this, we infused AAVretro-DIO-ChR2-eGFP into the SNr of D1-Cre;ChAT-eGFP mice (Fig. 3j), which caused retrograde ChR2-eGFP expression in dMSNs (Fig. 3k)[41]. Optical ChR2 stimulation evoked picrotoxin-sensitive oIPSCs in CINs (Fig. 3l, Supplementary Fig. 3a). Additionally, optical burst stimulation of SNr-projecting dMSNs inhibited CIN firing (Fig. 3m). These findings suggest that SNr-projecting dMSNs send axonal collaterals to inhibit CINs. Furthermore, we examined the presence of axonal fibers of CIN-projecting dMSNs in the internal globus pallidus (GPi) and SNr regions. We discovered that the dMSNs (green and red) that innervated CINs also projected to the GPi (Supplementary Fig. 3b–d) and SNr (Supplementary Fig. 3e–g). Taken together, these results suggest that

dMSNs send axons to the SNr to mediate reinforcement and also send collateral axons to inhibit CINs.

**CINs receive stronger inhibition from dMSNs than from iMSNs**
Since MSNs are GABAergic, more anatomic inputs from dMSNs than from iMSNs indicate that CINs are primarily inhibited by dMSNs. To test this, we generated D1-Cre;Ai32 and A2a-Cre;Ai32 mice, in which dMSNs or iMSNs expressed ChR2 to allow selective optogenetic stimulation, respectively (Fig. 4a, left). CINs in DMS slices were identified by their large size, spontaneous firing, and characteristic sag at hyperpolarizing currents (Fig. 4a, right). Optical dMSN excitation induced a large picrotoxin-sensitive inhibitory postsynaptic current (oIPSC) in CINs (Fig. 4b). Importantly, we discovered that dMSN→CIN oIPSCs were larger than iMSN→CIN oIPSCs (Fig. 4c; $F_{(1,27)} = 11.21$).

To investigate the consequences of dMSN-mediated inhibition of CINs, we measured spontaneous CIN firing following burst stimulation

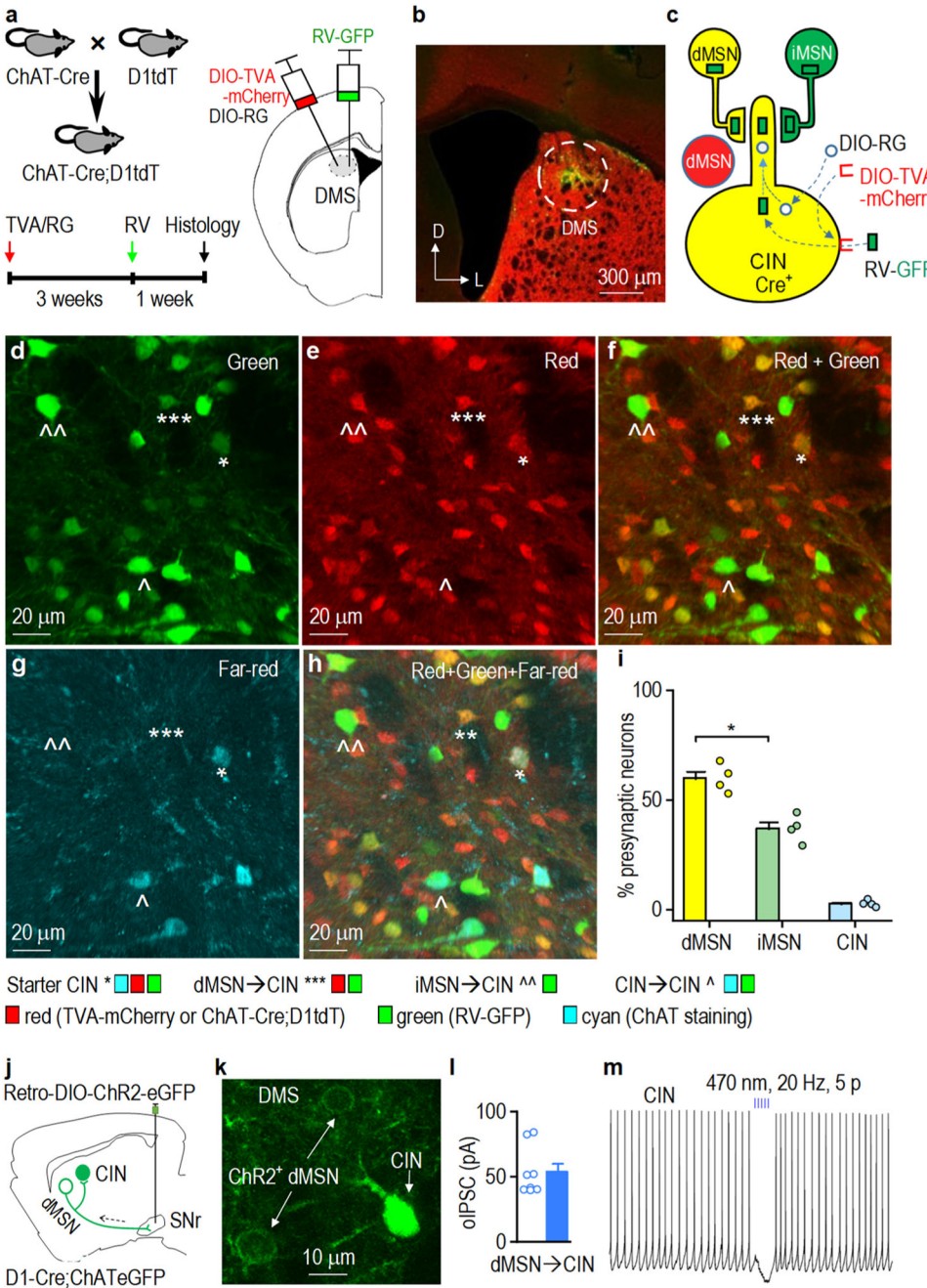

**Fig. 3 | CINs receive striatal inputs primarily from dMSNs. a** Schematic of experimental design. Cre-dependent helper viruses (AAV-DIO-TVA (EnvA receptor)-mCherry and AAV-DIO-RG (rabies glycoprotein)) were infused into the dorsomedial striatum (DMS) of ChAT-Cre;D1-tdTomato (tdT) mice, followed by rabies virus (RV-GFP) infusion at the same site 3 weeks later. Coronal sections were prepared 1 week after rabies virus infusion and were stained for choline acetyltransferase (ChAT; far-red, pseudo colored with cyan). **b** Confocal micrograph showing the injection site and viral expression in the DMS. L lateral, D dorsal. **c** Schematic showing viral expression and retrograde spread of RV-GFP. AAV-DIO-TVA-mCherry and AAV-DIO-RG infected Cre-positive cholinergic interneurons (CINs). RV-GFP infected TVA-positive CINs (starter cells expressed GFP and mCherry), labeling their presynaptic neurons with GFP. Since direct-pathway medium spiny neurons (dMSNs) were labeled red (from D1-tdT), dMSNs presynaptic to the starter CINs were yellow (red

and green overlap), whereas putative indirect pathway MSNs (iMSNs) were green. **d–h** Confocal micrographs of a DMS section; *Starter CINs, ***dMSN→CIN, ^^iMSN→CIN, ^CIN→CIN. **i** Summarized data showing that significantly more dMSNs (yellow) than iMSNs (green only) project to CINs; *$p = 0.036$. **j** AAVretro-DIO-ChR2-eGFP (Channelrhodopsin-2) was infused into the substantia nigra pars reticulata (SNr) of D1-Cre;ChAT-eGFP mice and DMS CINs were recorded. **k** Image of the DMS demonstrating ChR2-eGFP expression in dMSNs (cell membrane) and eGFP expression in CINs (cytoplasm). **l** Summarized optically-induced inhibitory postsynaptic currents (oIPSCs) recorded from CINs. **m** A burst of light stimulation (470 nm, 20 Hz, 5 pulses) of SNr-projecting dMSNs inhibited CIN firing. Unpaired $t$ test (**i**). $n = 4$ mice (**i**) and 8/3 (**l**). Data are presented as mean values ± SEM. Source data are provided as a Source Data file.

of dMSNs in DMS slices from D1-Cre;Ai32 mice (Fig. 4d). We observed a strong pause followed by a rebound in CIN firing (Fig. 4e; baseline versus pause: $t_{(43)} = 7.250$, pause versus rebound: $t_{(43)} = -11.907$, rebound versus baseline: $t_{(43)} = -5.407$). We also discovered that

optical dMSN stimulation inhibited CIN firing evoked by current injection (Supplementary Fig. 4a). Additionally, picrotoxin abolished the inhibitory effect of dMSN stimulationn on CIN firing (Supplementary Fig. 4b). Similarly, burst iMSN stimulation inhibited CIN firing in

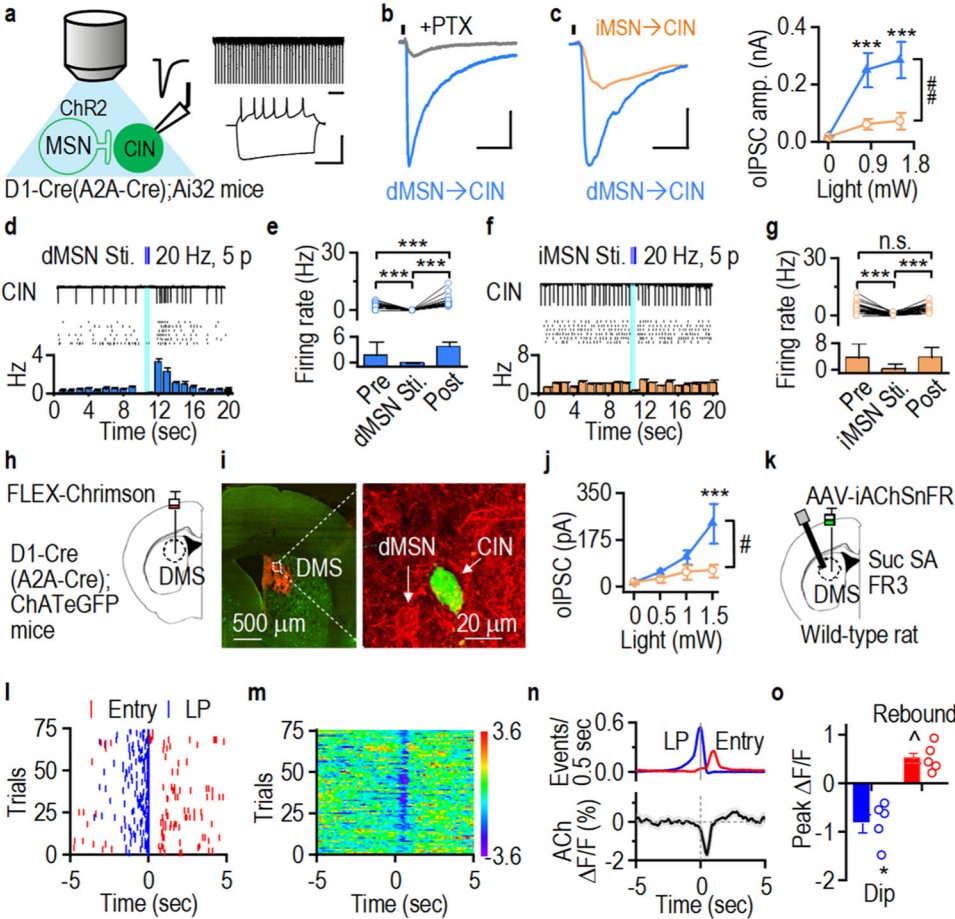

**Fig. 4 | CINs receive stronger functional GABAergic inputs from dMSNs than from iMSNs. a** Optogenetic stimulation of ChR2-expressing direct pathway (dMSNs) or indirect pathway MSNs (iMSNs) in D1-Cre;Ai32 or A2A-Cre;Ai32 mice respectively (left). Spontaneous firing (top right) and characteristic sag (bottom right) in DMS CINs. **b** Picrotoxin (PTX; gray)-sensitive optically evoked dMSN→CIN inhibitory postsynaptic current (oIPSC, blue). **c** oIPSCs were greater at dMSN→CIN (blue) than at iMSN→CIN synapses (orange); ***$p < 0.001$ versus iMSN→CIN, ##$p = 0.002$. **d** dMSN stimulation (Sti.; 20 Hz, 5 ms, 5 pulses) induced pause-rebound firing in a CIN. Top, firing; middle, multiple sweeps; bottom, corresponding histogram. **e** Summarized data comparing CIN firing before (pre), during and after (post) dMSN-burst stimulation; ***$p < 0.001$. **f** Representative histogram demonstrating a pause but no rebound of CIN firing on iMSNs stimulation (20 Hz, 5 ms, 5 pulses). **g** Summarized data demonstrating iMSN-mediated inhibition of CIN firing; ***$p < 0.001$. **h** Infusion of AAV-FLEX-Chrimson-tdTomato in the DMS of D1-

Cre;ChAT-eGFP and A2A-Cre;ChAT-eGFP mice. **i** Micrograph demonstrating Chrimson-tdTomato expression in dMSNs and GFP in CINs in D1-Cre;ChAT-eGFP mice. **j** Greater oIPSCs at dMSN→CIN than at iMSN→CIN synapses; ***$p < 0.001$ versus iMSN→CIN, #$p = 0.039$. **k** Wild-type (WT) rats infused with AAV-iAChSnFR in the DMS were trained to self-administer sucrose (Suc SA) using a fixed ratio 3 (FR3) schedule. **l** Sample lever pressing (LP, blue) and magazine entry (Entry, red) events, trials aligned to reward delivery. **m** Corresponding heat map of ACh activity. **n** Averaged ACh signal. **o** Summarized data demonstrating a significant dip (blue) and rebound (red) in ACh activity; *$p = 0.014$, ^$p = 0.015$ versus baseline. n.s. (not significant; **g**). Scale bars: 1 s (**a**, top right); 100 ms, 50 mV (**a**, bottom right); 50 ms, 200 pA (**b**); 10 ms, 250 pA (**c**). Two-way RM ANOVA followed by Tukey *post-hoc* test (**c**, **j**), paired *t* test (**e**, **g**, **o**). $n = 13/5$ (**c**, dMSNs), 16/5 (**c**, iMSNs), 44 sweeps/8 neurons (**i**), 32 sweeps/8 neurons (**g**), 10/3 (**j**, dMSNs), 14/5 (**j**, iMSNs) and 5 rats (**o**). Data are presented as mean values ± SEM. Source data are provided as a Source Data file.

A2A-Cre;Ai32 mice (Fig. 4f, g; baseline versus pause: $t_{(31)} = 7.47$). However, iMSN stimulation induced a weaker pause than did dMSN stimulation (Supplementary Fig. 4c; $t_{(12)} = 2.41$), and iMSN stimulation did not induce a rebound in CIN firing (Fig. 4f, g; rebound vs baseline: $t_{(31)} = -0.33$, $p = 0.743$).

The above studies used transgenic mice (Ai32) to express ChR2 in the whole brain, including the striatum. To selectively stimulate dMSNs or iMSNs in the DMS, we virally expressed AAV-Flex-Chrimson-tdTomato in D1-Cre;ChAT-eGFP or A2a-Cre;ChAT-eGFP mice, respectively (Fig. 4h, i). Whole-cell recordings revealed greater dMSN→CIN oIPSCs than iMSN→CIN oIPSCs (Fig. 4j; $F_{(1,22)} = 4.82$). Interestingly, dMSN stimulation inhibited CIN activity whereas iMSN stimulation did not (dMSN: Supplementary Fig. 4d; $t_{(12)} = 4.53$; iMSN: Supplementary Fig. 4e; $t_{(11)} = 0.15$, $p = 0.886$). These results suggest that CINs receive stronger inhibitory inputs from dMSNs than from iMSNs.

Reinforcement of reward seeking is mediated by dMSNs, and dMSN activity increases during action initiation of seeking behavior

(e.g., lever presses) in an instrumental task[1]. Since CINs receive strong inhibition from dMSNs and release acetylcholine (ACh), we hypothesized that striatal ACh levels decrease during reward seeking. To test this, we infused an acetylcholine sensor, AAV-iAChSnFR, and optical fibers in the DMS of WT rats (Fig. 4k, Supplementary Fig. 4f). Seven days after fiber implantation, the rats underwent training to lever press for sucrose rewards under a FR3 schedule (Fig. 4l). We found that ACh levels decreased immediately after lever presses and rebounded after reward deliveries (Fig. 4m–o; dip: $t_{(4)} = -4.20$, rebound: $t_{(4)} = 4.13$). These results suggest that lever pressing, which is associated with dMSN excitation[1], inhibits CIN activity.

### Drug reinforcement or dMSN self-stimulation potentiates inhibitory dMSN→CIN transmission

Repeated exposure to addictive substances potentiates dMSN activity[6,11–13,42]. Therefore, we next tested whether the exposure potentiated dMSN→CIN transmission. Firstly, D1-Cre;Ai167;ChAT-eGFP

mice were injected with cocaine for 7 d, and DMS dMSNs were recorded 2 d after the last injection (Fig. 5a). We found that cocaine injections increased AMPAR/NMDAR ratios (Fig. 5b; $t_{(38)} = -3.16$) and AMPA-induced currents in dMSNs (Fig. 5c; $t_{(15)} = -3.03$)[43]. Next, we recorded from DMS CINs in the same cocaine-injected mice (Fig. 5a). Cocaine significantly increased dMSN→CIN oIPSCs 2 and 24 d after the last cocaine injection (2 d: Fig. 5d, $F_{(1,42)} = 6.952$; 24 d: Fig. 5e, $t_{(22)} = -2.22$). These data suggest that cocaine potentiates dMSN activity and, thus, dMSN→CIN transmission.

dMSN activity is also potentiated by self-administration of natural rewards, like food or sucrose[8]. However, addictive substances, but not natural rewards, induce long-lasting synaptic plasticity[17]. Therefore, we next tested whether cocaine-induced suppression of CINs by dMSNs is long-lasting (Fig. 5f). D1-Cre;Ai167;ChAT-eGFP mice were trained to self-administer cocaine or sucrose for 11 days. Both groups earned similar numbers of reinforcers (Fig. 5g; $F_{(1,14)} = 0.45$, $p = 0.513$). dMSN→CIN oIPSCs were recorded three weeks after the last training session. We found that oIPSCs were higher in the cocaine group than in sucrose controls (Fig. 5h; $F_{(1,40)} = 4.36$). oIPSC PPRs were marginally reduced in the cocaine group (Supplementary Fig. 5a; $t_{(21)} = 1.53$, $p = 0.14$). These data suggest that cocaine-induced dMSN→CIN suppression is long-lasting.

Many addictive substances cause behavioral reinforcement via similar neuronal mechanisms[6,7]. We next tested whether alcohol could also alter GABAergic transmission onto DMS CINs. Transgenic mice were trained to consume alcohol using an intermittent-access two-bottle choice (2BC) procedure[44] (Fig. 5i) for minimum 8 weeks and recordings were performed 1 d after the last alcohol session. We discovered that alcohol intake increased the frequency, but not amplitude, of spontaneous inhibitory postsynaptic currents (sIPSCs) in DMS CINs (frequency: Fig. 5j, $t_{(29)} = -2.28$; amplitude: Fig. 5k, $t_{(29)} = -1.32$, $p = 0.198$). Additionally, it potentiated dMSN→CIN oIPSCs (Fig. 5l; $F_{(1,9)} = 10.33$). These data suggest that chronic alcohol consumption potentiates dMSN→CIN transmission.

dMSN activity can also be potentiated by optogenetic intracranial self-stimulation (oICSS) of dMSNs[16]. Therefore, we next examined whether oICSS of DMS dMSNs potentiated dMSN→CIN transmission. We virally expressed Chrimson in DMS dMSNs in D1-Cre rats and then trained them to lever press to receive laser stimulation. These rats exhibited escalated oICSS across sessions, whereas the control group (no laser stimulation) did not (Fig. 5m, n; $F_{(1,9)} = 22.431$). One day after the last session, dMSN→CIN oIPSCs were recorded. We found that dMSN oICSS induced greater dMSN→CIN oIPSCs and marginally reduced oIPSC PPR (Fig. 5o, $F_{(1,25)} = 6.11$; Supplementary Fig. 5b, $t_{(33)} = 1.68$, $p = 0.10$).

Taken together, these findings indicate that reinforcement of addictive substances (cocaine and alcohol) and dMSN self-stimulation potentiate inhibitory dMSN→CIN transmission.

## Time-locked optogenetic CIN inhibition impairs reversal and extinction learning

We discovered that exposure to addictive substances enhanced inhibitory dMSN→CIN transmission, likely resulting in increased suppression of CIN activity. Given that CINs regulate reversal learning[30,31], we examined whether chemogenetic inhibition of DMS CINs disrupted the cognitive flexibility of instrumental learning (Supplementary Fig. 6a–d). During initial training, mice showed a preference for nose-poking for food rather than sucrose (Supplementary Fig. 7a). We then analyzed the action-outcome contingency learning for food rewards in the task. Initial training results for nose-poke rates, magazine entries and contingency learning showed no difference between groups (Supplementary Fig. 6e, f, Supplementary Fig. 7b). However, on the first day of reversal training, the hM4Di group displayed more magazine entries than the control group, while there were no differences in nose pokes and rewards earned between the groups (Supplementary

Fig. 6g–i, Supplementary Fig. 7c–e). The return map of inter-entry intervals was concentrated near the origin for the hM4Di group but scattered for controls, indicating that the hM4Di group entered the magazine more frequently (Supplementary Fig. 6j–l). Furthermore, the hM4Di group had fewer learned action-outcome sequences (Supplementary Fig. 6m–o). We also observed that the the hM4Di group did not enter the magazine to collect rewards immediately after delivery during reversal training, unlike the control group (Supplementary Fig. 6p–r). Since both groups eventually learned the reversed action-outcome associations (Supplementary Fig. 7f), this finding suggests that chemogenetic CIN inhibition slowed down the reversal learning process. More information about this experiment can be found in the supplementary discussion.

Since chemogenetic CIN inhibition lasts for hours after CNO injection, it does not allow us to identify exactly when CINs regulate reversal learning. CINs are activated by salient environmental stimuli[45], including changes in rewards delivered (e.g., food→sucrose) during the reversal task. Therefore, we hypothesized that DMS CIN activity during reward delivery regulates reversal learning. To test this, we optogenetically inhibited DMS CINs when rewards were delivered during reversal training in rats that demonstrated strong goal-directed learning[30]. eNpHR was virally expressed in DMS CINs of ChAT-Cre;tdT rats (Fig. 6a–c). Rats were then trained on the reversal learning task as explained in Fig. 1 (Fig. 6d). During initial training, the eNpHR and eYFP (control) groups did not differ in lever pressing or magazine entries (lever pressing: Supplementary Fig. 8a, $F_{(1,15)} = 0.49$, $p = 0.495$; entries: Supplementary Fig. 8b, $F_{(1,15)} = 0.045$, $p = 0.835$). Both groups were sensitive to outcome devaluation (Fig. 6e). There was a main effect of devaluation ($F_{(1,16)} = 48.994$, $p < 0.001$) but no group x devaluation interaction ($F_{(1,16)} = 1.304$, $p = 0.270$). Devaluation indices were calculated for rats in both groups as (Val-Dev)/(Val+Dev). Both groups exhibited similar devaluation indices (Fig. 6f; $t_{(14)} = 0.73$, $p = 0.48$).

The rats then underwent reversal training. Constant light stimulation (590 nm) was time-locked to reward deliveries to inhibit CIN activity (Fig. 6g). We discovered that although lever pressing was unaffected (Supplementary Fig. 8c; $F_{(1,15)} = 0.951$, $p = 0.345$), the eNpHR group entered the magazine more than controls during the first two sessions (Supplementary Fig. 8d; $F_{(1,14)} = 4.18$). We further classified magazine entries into effective and ineffective entries, depending on whether the rats entered the magazine during reward delivery or at another time, respectively. The ineffective entries on the first day of reversal training were significantly higher in the eNpHR group (Supplementary Fig. 8e; $t_{(18)} = -2.24$), whereas effective entries did not differ between groups (Supplementary Fig. 8f; $t_{(18)} = -0.57$, $p = 0.577$). Additionally, the return maps of inter-entry intervals revealed that the center of the data points was closer to the origin for the eNpHR group (Supplementary Fig. 8g–i), suggesting that the eNpHR rats checked the magazine for a reward more frequently. Importantly, the eYFP, but not the eNpHR, group was sensitive to outcome devaluation after reversal training (Fig. 6h). Statistical analysis revealed a significant group x devaluation interaction ($F_{(1,16)} = 10.937$, $p = 0.004$). Further simple effects test revealed that, whereas the eYFP group pressed the valued lever significantly more than the devalued lever (Fig. 6h left; $F_{(1,16)} = 8.805$), the eNpHR group did not (Fig. 6h right; $F_{(1,16)} = 2.923$, $p = 0.107$). Consequently, the devaluation index was lower in the eNpHR group (Fig. 6i; $t_{(14)} = 2.34$), suggesting that inhibition of DMS CINs during reward delivery impaired reversal learning. In contrast, optogenetic CIN inhibition time-locked to lever presses did not affect reversal learning (Supplementary Fig. 8j; $t_{(9)} = 0.72$, $p = 0.49$).

Reversal learning involves inhibition of previously learned action-outcome contingencies (extinction). Therefore, we next tested whether inhibition of DMS CINs altered extinction learning. CINs were inhibited when a reward was supposed to be delivered (Fig. 6j). We observed higher lever-press rates in the eNpHR group on the first day

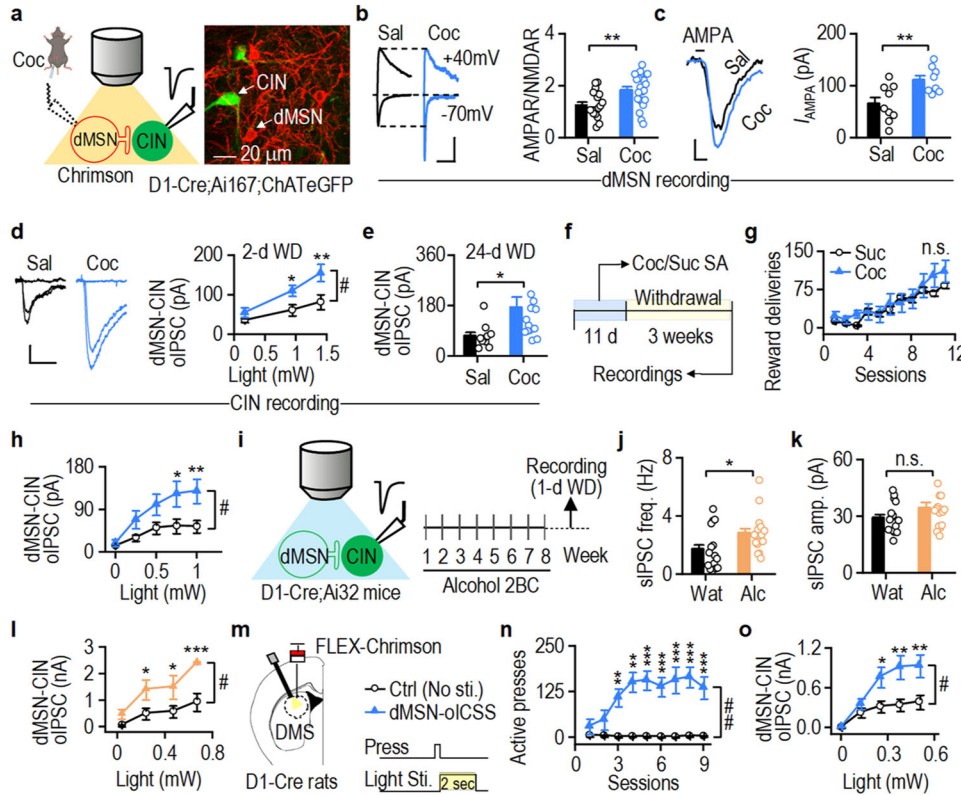

**Fig. 5 | Drug reinforcement potentiates GABAergic dMSN→CIN transmission in the DMS. a, b** Repeated cocaine (Coc) but not saline (Sal) injections increased AMPAR/NMDAR ratios in DMS dMSNs of D1-Cre;Ai167;ChAT-eGFP mice, wherein dMSNs expressed Chrimson-tdTomato and CINs expressed eGFP; **$p = 0.003$. **c** Cocaine potentiated AMPA-induced currents ($I_{AMPA}$) in dMSNs; **$p = 0.008$. **d, e** Cocaine potentiated dMSN→CIN oIPSCs after 2 d (**d**; *$p = 0.056$, **$p < 0.01$ versus Sal; #$p = 0.012$) and 24 d (**e**; *$p = 0.037$) withdrawal (WD). **f, g** D1-Cre;Ai167;ChAT-eGFP mice self-administered (SA) cocaine or sucrose and were recorded from after 3 weeks. **h** Cocaine self-administration caused greater potentiation of dMSN→CIN oIPSCs than sucrose self-administration; *$p < 0.05$, **$p < 0.01$ versus Suc; #$p = 0.043$. **i** D1-Cre;Ai32 and ChAT-eGFP mice consumed alcohol (Alc; orange), controls received only water (Wat; black). **j, k** Alcohol consumption increased sIPSC frequency (freq.) (**j**; *$p = 0.030$), but not amplitude (amp.) (**k**), in DMS CINs of ChAT-eGFP mice. **l** Alcohol potentiated dMSN→CIN oIPSCs in D1-Cre;Ai32 mice; *$p < 0.05$,

***$p < 0.001$ versus Wat; #$p = 0.011$. **m, n** D1-Cre rats infused with AAV-FLEX-Chrimson-tdTomato in the DMS pressed levers for optical intracranial self-stimulation (oICSS; blue) of dMSNs (2-s constant), controls (Ctrl; black) did not receive stimulation; **$p < 0.01$, ***$p < 0.001$ versus Ctrl; ##$p = 0.001$. **o** dMSN self-stimulation potentiated dMSN→CIN oIPSCs 1 d after the last training session; *$p < 0.05$, **$p < 0.01$ versus Ctrl; #$p = 0.021$. n.s. (not significant; **g, k**). Scale bars: 100 ms, 50 pA (**b**); 50 s, 30 pA (**c**); 50 ms, 30 pA (**d**). Two-way RM ANOVA followed by Tukey *post-hoc* test (**d, g, h, l, n, o**), unpaired *t* test (**b, c, e, j, k**). $n = 18/4$ (**b**, Sal), 22/5 (**b**, Coc), 9/3 (**c**, Sal), 8/3 (**c**, Coc), 22/5 (**d**, Sal), 22/7 (**d**, Coc), 10/3 (**e**, Sal), 14/4 (**e**, Coc), 7 mice/group (**g**), 20/5 (**h**, Suc), 22/5 (**h**, Coc), 15/3 (**j, k**; Wat), 16/4 (**j, k**; Alc), 6/4 (**l**, Wat), 5/3 (**l**, Alc), 6 rats/group (**n**), 14/5 (**o**, Ctrl), and 13/6 (**o**, dMSN-oICSS). Data are presented as mean values ± SEM. Source data are provided as a Source Data file.

---

of extinction (Fig. 6k; $p = 0.006$), indicating delayed extinction learning. Time-course analysis of lever pressing for the two groups revealed a difference in lever presses made between the 10 and 20 min time points of the extinction session (Fig. 6l, m; $t_{(15)} = -2.95$). Taken together, these results indicate that time-locked inhibition of DMS CINs during reward delivery impairs reversal and extinction learning.

## Discussion

This study identified a striatal mechanism underlying the cognitive flexibility deficits induced by the reinforcement of addictive substances. We found that cocaine exposure impaired cognitive flexibility, as evidenced by reversal learning deficits in a two-action-outcome instrumental task. Additionally, cocaine administration attenuated DMS CIN activity, which was associated with an increase in inhibitory presynaptic inputs onto CINs. Furthermore, we discovered that dMSNs provide the primary inhibitory inputs to CINs. Cocaine administration potentiated glutamatergic inputs to DMS dMSNs and enhanced inhibitory dMSN→CIN transmission. Moreover, chemo- or optogenetic inhibition of DMS CINs reduced cognitive flexibility. Importantly, we discovered that SNr-projecting dMSNs, which mediate reinforcement, also sent axonal collaterals to inhibit CINs, which mediate cognitive

flexibility. These findings imply that the local inhibitory dMSN→CIN transmission mediates the cognitive flexibility deficits induced by addictive substances.

We discovered that experimenter-delivered cocaine impaired reversal learning in an instrumental task. We observed that while cocaine did not alter initial learning of action-outcome contingencies, it caused deficits in reversal training. This reversal learning deficit is unlikely to result from a cocaine-induced decrease in motivation to consume natural rewards, as both cocaine- and saline-treated groups exhibited similar initial learning. Interestingly, several studies have reported that exposure to cocaine or amphetamine impairs the initial acquisition of action-outcome contingencies[46-48]. The discrepancy in the impact of drug exposure on initial learning between these reports and our study is unlikely to result from differences in drug dose or intensity, as most of these studies employed a similar drug treatment intensity as ours. However, those studies showing initial learning deficits utilized a random interval (RI) schedule, rather than the random ratio (RR) paradigm during training. In contrast, when using the RR schedule, two previous studies also observed that drug exposure did not lead to deficits in goal-directed learning following initial training[49,50], which aligns with our findings. Notably, one study

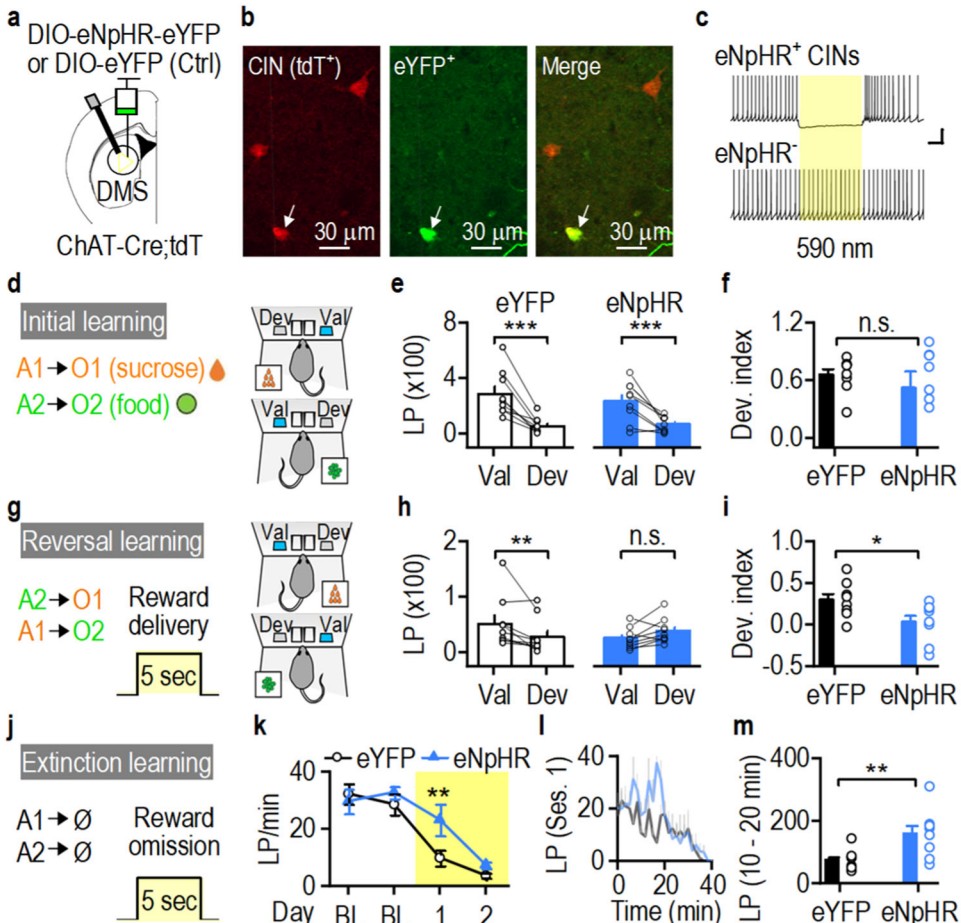

**Fig. 6 | Time-locked optogenetic inhibition of DMS CINs impairs reversal and extinction learning. a–c** AAV-DIO-eNpHR-eYFP (halorhodopsin; blue) or AAV-DIO-eYFP (Ctrl: control; black) was infused into the dorsomedial striatum (DMS) of ChAT-Cre;tdTomato (tdT) rats (**a**). tdT-positive cholinergic interneurons (CINs) expressed eNpHR-eYFP (**b**) and were inhibited by yellow light stimulation of eNpHR in slice recordings (**c**). **d** Initial action (A)-outcome (O) contingency training (A1→O1; A2→O2) (left) and the initial devaluation test (right). **e** eYFP and eNpHR groups were sensitive to outcome devaluation after initial training; ***$p < 0.001$. **f** The initial devaluation (Dev.) index did not differ between groups. **g** Reversal training (A1→O2; A2→O1) (left) and the devaluation test after reversal (right). Reward deliveries were paired with laser stimulation. **h** The eYFP (left), but not the eNpHR (right), group was sensitive to outcome devaluation after reversal training (left); **$p = 0.009$. **i** The

devaluation index was lower in the eNpHR group than in controls; *$p = 0.034$. **j** Extinction training: lever presses did not lead to reward deliveries (Ø). Reward omissions were paired with laser stimulation. **k** Lever-press rate was higher in the eNpHR group than in controls during the first extinction session; **$p = 0.006$ versus eYFP. **l** Averaged lever pressing for the two groups during the first extinction session (Ses. 1). **m** Summarized data showing that lever presses during 10–20 min of the extinction session were higher in the eNpHR group than the eYFP controls; **$p = 0.009$. Val valued, Dev devalued, LP lever presses, BL baseline, n.s. (not significant; **f**, **h**). Scale bar: 1 s, 20 pA (**c**). Unpaired $t$ test (**f**, **i**, **m**), mixed model ANOVA followed by simple effects test (**e**, **h**), two-way RM ANOVA followed by Tukey *post-hoc* test (**k**). $n = 9$ rats/group. Data are presented as mean values ± SEM. Source data are provided as a Source Data file.

employed both RR and RI schedules in separate groups and found that RR training, but not RI training, preserved goal-directed learning[51]. Additionally, when a fixed-ratio schedule was used, short (1 h/day) but not long (6 h/day) cocaine IVSA training maintained goal-directed learning[52]. While there are some similarities, the neural mechanisms involved in initial learning and reversal learning are distinct. For instance, studies have demonstrated that ablation of DMS CINs impairs reversal learning without affecting initial learning[30,31]. This suggests that drug exposure selectively disrupts cholinergic activity, which is required for reversal learning but not for initial learning.

We found that both non-contingent and contingent cocaine exposure led to a long-lasting reduction in the spontaneous firing of DMS CINs. Interestingly, recent studies have reported the downregulation of CIN activity in the ventral striatum after acute amphetamine exposure[53], and D2R overexpression in Nac CINs after cocaine self-administration[32], which aligns with our finding that addictive substances suppress striatal CIN activity. Surprisingly, acute cocaine exposure in vivo or ex vivo has been shown to increase striatal cholinergic activity[54]. However, long-term effects have not been reported before.

The opposing regulation of CIN firing by acute and repeated cocaine exposure suggests that cocaine withdrawal could lead to adaptations in CIN firing. Given that CINs are usually excited by salient stimulation to inhibit corticostriatal transmission[13,55,56], cocaine-induced reduction of CIN firing would impair such inhibitor control, which may contribute to the compulsive/inflexible decision-making observed in drug-exposed subjects. ACh also binds to nAChRs on striatal dopaminergic terminals, leading to local dopamine release[57], which affects corticostriatal synaptic plasticity in reinforcement learning.

CINs receive various inputs, including glutamatergic[56], dopaminergic[58], recurrent[40], substance-P, and enkephalin[33,59,60]. Notably, glutamatergic inputs from the parafascicular thalamic nuclei induce D2R-mediated burst-pause firing in CINs, which is essential for behavioral flexibility[30,31]. However, in our study, we report GABA$_A$R-mediated pause-rebound firing in CINs caused by dMSN excitation that regulates drug-induced behavioral inflexibility. In line with reduced CIN activity, we found that cocaine or alcohol enhanced GABAergic transmission in CINs due to an increase in presynaptic GABAergic inputs. Our rabies study suggests that the primary source

of these inputs is striatal D1R-expressing neurons. Striatal tyrosine hydroxylase interneurons also express D1Rs and send GABAergic inputs to DMS CINs[39]. However, there are far fewer interneurons than D1R-MSNs. CINs also receive GABAergic inputs from the external globus pallidus[61], but they do not express D1Rs[62]. Also, slice recordings confirmed that dMSNs send stronger GABAergic inputs to CINs than iMSNs. This distinction between dMSN and iMSN GABAergic inputs to CINs has not been reported previously. However, earlier studies noted that there were more substance P (co-released by dMSNs)-positive synapses than enkephalin (co-released by iMSNs)-positive synapses onto CINs[59,60]. These observations are consistent with our findings, indicating that dMSNs provide the primary GABAergic inputs to CINs.

Addictive substances and intracranial dMSN self-stimulation enhanced GABAergic dMSN-CIN transmission. Since we did not observe any postsynaptic changes in GABA$_A$R responsiveness, the enhanced dMSN→CIN transmission likely resulted from a potentiation in dMSN activity. Consistently, cocaine administration increased AMPA-induced currents and the AMPAR/NMDAR ratio in DMS dMSNs. Interestingly, alcohol administration, intracranial self-stimulation of dopamine neurons, and self-administration of natural rewards have all been shown to potentiate striatal dMSN activity by increasing AMPAR trafficking[6–8,11,15,63]. The present study found that drug self-administration led to greater inhibitory dMSN→CIN transmission than sucrose self-administration, and this difference was observed 3 weeks after the last drug exposure. This increased inhibition may be attributed to the selective facilitation of NMDA receptor and extracellular signal-regulated kinase (ERK) activity in dMSNs by drugs[64–67]. In contrast, sucrose self-administration did not alter NMDAR activity[42]. The selective enhancement of NMDA receptor activity by drugs, as opposed to natural rewards, may explain the persistent strengthening of dMSN→CIN transmission observed following drug exposure. Taken together, these findings indicate that reinforcement of addictive substances potentiates dMSN activity and strengthens GABAergic dMSN→CIN transmission, which reduces CIN activity.

Previous research has demonstrated that glutamatergic projections from the thalamus→CINs mediate reversal learning[30,31,55]. In the present study, we show that the inhibitory dMSN→CIN transmission plays a modulatory role in regulating reversal learning, and that this inhibition is upregulated after drug exposure, causing enhanced suppression of CIN activity, and thereby reducing cognitive flexibility. Secondly, dMSNs are not always excited in vivo, rather dMSN activity increases during action-initiations[1]. Therefore, inhibition of CINs via dMSNs is not consistently maintained. However, long-lasting dMSN excitation (via drug exposure or excessive reinforcing behaviors) could lead to a decline in CIN activity.

To summarize, we demonstrate that cocaine induces long-lasting potentiation of inhibitory dMSN→CIN transmission, thus reducing cognitive flexibility in a two-action-outcome reversal learning task. This cocaine-mediated deficit in cognitive flexibility is associated with increased GABAergic inputs to CINs, which reduces DMS CIN activity. Chemo- or optogenetic inhibition of DMS CINs, conducted to mimic the reinforcement effect of addictive substances, also reduced cognitive flexibility. Importantly, SNr-projecting dMSNs, which mediate reinforcement, send axonal collaterals to inhibit DMS CINs, which regulate behavioral flexibility. Our findings demonstrate that the local inhibitory dMSN→CIN circuit mediates the reinforcer-induced reduction in cognitive flexibility.

## Methods
### Animals
Transgenic mice were on the C57BL/6 (Mutant Mouse Regional Resource Centers) background, whereas transgenic rats were on the Long-Evans (Harlan Laboratories) background. All electrophysiology, histology and behavioral experiments were conducted in 4- to 7-month-old male and female mice/rats. Both mice (4–5 mice per cage)

and rats (2 rats per cage) were group-housed and maintained in a temperature (72 degrees fahrenheit)- and humidity (54 percent)-controlled environment. In total, 116 male and 82 female animals were used for the study and sex was not considered in the study design. This was because it has been shown that cocaine administration impairs reversal learning in both male and female animals equally[22]. For the behavioral experiments, animals were kept in a 12:12 hr. light: dark cycle, and the experiments were performed in the dark cycle. Details regarding the single transgenic mice and rats used in this study are specified in Supplementary Tables 2 and 3, respectively. Animals were crossed as per the requirement of the experiment, Fig. 1a–h: wild-type rats, Fig. 1i–k: wild-type mice, Fig. 2: ChAT-eGFP mice, Fig. 3a–i: ChAT-Cre;D1tdT mice, Fig. 3j–m: D1-Cre;ChAT-eGFP mice, Fig. 4a–g: D1-Cre;Ai32;ChAT-eGFP and A2A-Cre;Ai32;ChAT-eGFP mice, Fig. 4h–j: D1-Cre;ChAT-eGFP and A2A-Cre;ChAT-eGFP mice, Fig. 4k–o: wild-type rats, Fig. 5a–h: D1-Cre;Ai167;ChAT-eGFP mice, Fig. 5j–k: ChAT-eGFP mice, Fig. 5l: D1-Cre;Ai32 mice, Fig. 5m–o: D1-Cre rats, Fig. 6: ChAT-Cre;tdT rats. All animal care and experimental procedures were approved by the Texas A&M University Institutional Animal Care and Use Committee and were conducted in agreement with the National Research Council Guide for the Care and Use of Laboratory Animals (AUP# 2022-0198).

### Reagents
Cocaine solutions were prepared with (−)-cocaine hydrochloride (NIDA Drug Supply Program; cat #9041-001) in saline. CNO was obtained from the NIMH chemical synthesis and drug supply program. DNQX disodium salt (AMPA/kainate antagonist) was obtained from Abcam (ab120169). AP5 was obtained from R&D systems. All other reagents were obtained from Sigma. Details regarding viruses are provided in Supplementary Table 1.

### Stereotaxic virus infusions
Animals were anesthetized with 3–4% (vol/vol) isoflurane at 1.0 L/min and placed on a heating pad to maintain body temperature. The head was leveled, and craniotomies were performed using stereotaxic coordinates from the mouse or rat brain atlas[68,69]. Viruses were infused at the following coordinates. DMS1-AP: 0.48, ML: ±2.5, DV: −4.7; DMS2-AP: −0.24, ML: ±3, DV: −4.7 (Fig. 4k–o, Fig. 5m–o, Fig. 6; rats). DMS1-AP: 0.38, ML: ±1.55, DV: −2.9; DMS2-AP: 0.02, ML: ±1.8, DV: −2.7 (Supplementary Fig. 6; mice). DMS-AP: 0.38, ML: ±1.55, DV: −2.9 (Fig. 3a–i, Fig. 4h–j, mice). SNr-AP: −3.28, ML: ±1.2, DV: −4.5 (Fig. 3j–m, mice). 0.8-1 μL (for rats) and 0.4−0.5 μL (for mice) virus was infused at each injection site at a rate of 0.1 μL/min. Injectors were kept in place for 10 min to allow the virus to diffuse. In Fig. 3, rabies-GFP was infused with the injectors at a 10-degree angle to avoid contamination of the infusion track.

### Optical fiber implantations
Rats were anesthetized using 2–3% (vol/vol) isoflurane and were mounted on a stereotaxic frame (Kopf instruments). An incision was made, and bilateral optical fiber implants (300-nm core fiber secured to a 2.5-mm ceramic ferrule with 5 mm of fiber extending past the end of the ferrule) were lowered into the DMS at an angle of 5 degrees at the coordinates of AP: 0.12, ML: ± 3.2, DV: −4.55 from Bregma. Implants were secured to the skull using metal screws and dental cement (Henry Schein) and covered with denture acrylic (Lang Dental). The incision was closed around the head cap, and the skin was vet-bonded to the head cap. Rats were monitored for 1 week or until they had resumed normal activity. The methods were adapted from Ma et al.[42].

### Jugular vein catheter implantation surgery
The jugular vein catheter was constructed in-house, with the main parts purchased from SAI Infusion Technologies. Prior to the surgery, all catheters were tested for leaks using sterile water, and the

intracardial tubing end was trimmed to 1.2 mm from the anchoring bead. Catheter implantation surgery was performed under oxygen/sevoflurane vapor and based on protocols described in Thomsen & Saine[70]. Briefly, the catheter base was placed above the mid-scapular region, and the intracardial tubing was passed subcutaneously over the right shoulder and inserted 1.0–1.2 mm into the jugular vein. Two suture threads were tied around the vein above and below the anchoring beads to secure the intracardial tubing. Following surgery, mice were rested for 24 h. They received daily administration of heparin and cefazolin solution (30 USP units/mL heparin and 67 mg/mL cefazolin in 0.02 mL of 0.9% saline) through the catheter for 7 d. Antibiotic ointment was applied daily to the skin at the surgery site. Thereafter, catheters were flushed daily with 0.03 mL heparinized saline (0.9%) throughout the self-administration sessions. Catheter patency was examined weekly by using it for injections of 0.03 mL of 15 mg/mL ketamine and 0.75 mg/mL midazolam, with loss of righting reflex required within 5 s[71].

## Slice electrophysiology

Coronal sections of the dorsomedial striatum (thickness: 250 μm) were cut at a speed of 0.14 mm/s in ice-cold cutting solution containing (in mM) 40 NaCl, 148.5 sucrose, 4.5 KCl, 1.25 NaH$_2$PO$_4$, 25 NaHCO$_3$, 0.5 CaCl$_2$, 7 MgSO$_4$, 10 dextrose, 1 sodium ascorbate, 3 myo inositol and 3 sodium pyruvate, and bubbled with 95% O$_2$ and 5% CO$_2$. The slices were then incubated for 45 min in a 1:1 mixture of cutting and external solution containing (in mM) 125 NaCl, 4.5 KCl, 2.5 CaCl$_2$, 1.3 MgSO$_4$, 1.25 NaH$_2$PO$_4$, 25 NaHCO$_3$, 15 sucrose and 15 glucose, and saturated with 95% O$_2$ and 5% CO$_2$.

Potassium-based intracellular solution was used for all cell-attached and current-clamp recordings in Figs. 2d–f, 2m, 3m, 4d–g, Supplementary Fig. 4a–e, Supplementary Fig. 6b, Fig. 6c; this contained (in mM) 123 potassium gluconate, 10 HEPES, 0.2 EGTA, 8 NaCl, 2 MgATP and 0.3 NaGTP, with the pH adjusted to 7.3 using KOH. Cesium intracellular solution was used for all other whole-cell recordings; this contained (in mM) 119 CsMeSO$_4$, 8 TEA.Cl, 15 HEPES, 0.6 ethylene glycol tetraacetic acid (EGTA), 0.3 Na$_3$GTP, 4 MgATP, 5 QX-314.Cl and 7 phosphocreatine, with the pH adjusted to 7.3 using CsOH. The recording bath was maintained at 32 °C, and the perfusion speed was set to 2–3 mL/min.

NBQX (10 μm) and AP5 (50 μm) were used to block glutamatergic transmission for IPSC recordings (Figs. 2, 4 and Supplementary Figs. 4, 5d–o, Supplementary Fig. 5). Picrotoxin (100 μm) was used to block all GABAergic transmission for EPSC recordings (Fig. 5b, c). CINs in slices were identified by their labeled color, large size, spontaneous firing, characteristic sag and were held at −60 mV, whereas dMSNs were held at −70 mV. Spontaneous cell-attached CIN firing activity was recorded for 5 min to calculate the average firing frequency. sIPSCs were recorded for 3 min to calculate the average sIPSC frequency and amplitude. For electrically evoked IPSC/EPSC recordings, the stimulation electrode was placed 100-150 μm away from the neuron of interest. For optically evoked IPSC recordings, 2 ms flash of light was used (470 nm, ChR2; 590 nm, Chrimson). For paired-pulse ratio (PPR) recordings, two stimulations (electrical/optical) separated by an inter-stimulation interval were applied and the ratio of peak response 2 over peak response 1 was calculated. The parameters of optical burst stimulation were 470/590 nm, 20 Hz, 5 ms, 5 pulses.

For the measurement of AMPAR/NMDAR ratio, the peak currents of AMPAR-mediated EPSCs were measured at a holding potential of −70 mV and the NMDAR-mediated EPSCs were estimated as the EPSCs at + 40 mV, 30 ms after the peak AMPAR EPSCs, when the contribution of the AMPAR component was minimal[42]. The AMPA/NMDA ratio was calculated by dividing the AMPAR EPSCs by NMDAR EPSCs. To measure AMPA-induced currents in dMSNs, AMPA (5 μM) was bath-applied to the slices for 15 s, and holding currents were recorded at a frequency of 0.2 Hz[43]. Electrophysiology data were acquired using Clampex (in

pClamp 10.7, Molecular Devices), data was analyzed using using Clampfit (in pClamp 10.7, Molecular Devices) and Mini Analysis (Mini60, Synaptosoft Inc.).

## Behavior

Behavioral experiments were conducted in 4- to 7-month-old animals that were housed at 23 °C with a 12-h light: dark cycle. All behavioral experiments were performed during the dark phase.

### Cocaine-induced hyperlocomotion

Saline was injected intraperitoneally (i.p.) in wild-type mice/rats for 3 d to habituate them to the injection procedure and the open-field locomotion test. After habituation, half of the animals continued to receive saline, while the other half received daily i.p. cocaine (15 mg/kg) injections over the next 7 d. Total mean intake per mouse -105 mg/kg (Fig. 2a–k). After each injection, animals were placed in an open-field chamber, and locomotion was measured for 30 min. Any locomotor activity resulted in beam breaks, which were detected and recorded by the software (Kinder Scientific, CA).

### Cocaine intravenous self-administration

Operant chambers (Coulbourn Instruments, MA) containing two nose-poke ports were each housed in light- and sound-attenuating chambers. The chambers had cue lights inside and above each port, and a cannula that ran through the ceiling of the chamber could be attached to the implanted jugular vein catheter. Mice were trained to self-administer an intravenous infusion of cocaine (1 mg/kg/infusion in sterile 0.9% saline). The drug infusion volume was 0.56 mL/kg, and the concentration was 0.18 mg/mL. The length of each infusion was calculated using weight (kg)/0.01 s. The experiment ran for 7 d, with one 3-h session per day. The maximum number of cocaine infusions that the animals could receive during each session was 30. Illumination of the house light signaled reward availability, and the light was turned off following each infusion. There was a 20-s timeout period following each infusion, during which an effective nose-poke did not trigger an infusion. The acquisition phase was conducted with an FR1 schedule, where each effective nose-poke on the active port resulted in one drug infusion coupled with activation of the cue light inside the nose-poke hole[71]. Total mean intake per animal -84 mg/kg (Fig. 2l–r). In Fig. 5f–h, the concentration of cocaine used for IVSA training was 0.5 mg/kg/infusion.

### Intracranial optogenetic self-stimulation of dMSNs

Optical fibers implanted in the rats were connected to the laser via patch cables before the rats were placed in an operant chamber containing two levers with cue lights above them. An active lever press activated the corresponding cue light (10 pulses; 0.5-s ON and 0.5-s OFF); this was paired with constant laser stimulation (590 nm, 2 s). An inactive lever press had no effect. Lever pressing during laser stimulation had no effect, but the presses were recorded. Each session lasted for 30 min. The control rats were introduced into the chambers, but lever pressing did not result in cue light/laser stimulation. Rats were mildly food-deprived to increase their activity.

### Sucrose self-administration

For Fig. 4k, rats were introduced into the operant chamber with two lever presses. Pressing the left lever resulted in the delivery of sucrose solution (20%, Tocris) and a cue light to signal delivery, whereas pressing the right lever had no effect. Rats were trained on a continuous reinforcement schedule for 3 days before moving them to a FR3 schedule. Photometry recordings were performed on the 3rd FR3 training session. The training session lasted for 30 min.

For Fig. 5g, mice were introduced into the operant chambers where a left nose poke resulted in the delivery of sucrose solution (10%,

Tocris) and a cue light to signal delivery; right nose pokes had no effect. Each session lasted for 2 h.

## Reversal learning in rats (Figs. 1a–h and 6a–i)
This procedure was adapted from Bradfield et al.[30].

**Apparatus design.** The operant chambers (Coulbourn Instruments) contained a house light, a white noise speaker, and two retractable levers on the right wall. Between the two levers, there were adjacent food (pellet) and sucrose (solution) magazines, which shared a common sensor. The training protocols were written in Graphic State, and the chambers were controlled using Graphic State RT.

**Food deprivation.** All rats were food-deprived for at least 3 d before training to bring their body weights down to 85% of their free weight. Bodyweight was maintained at 85% thereafter.

**Magazine training (4 d).** Levers were retracted, and rats were trained to enter the food or sucrose magazines to collect either food pellets (45 mg; Dustless Precision Pellets Rodent, Grain-Based; Bio-Serv) or sucrose solution (20% sucrose (Tocris) in drinking water, 5-sec availability). Each session started with the illumination of the house light. Twenty deliveries of either food pellets or sucrose solution were made in the respective magazine using the RT60 schedule. Only one type of reward (either food or sucrose) was delivered during each session. The reward type was distributed evenly over the four magazine training sessions. 200 total magazine entries were set as the threshold to gauge sufficient motivation in rats before moving on to the instrumental training phase. Each rat was trained in the same operant chamber throughout.

**Initial training (11 d).** Each training session had 4 sub-sessions (2 food and 2 sucrose sub-sessions in random order) separated by a 2.5 min timeout period when the house light was turned off, and levers were retracted. During each sub-session, only one type of reward (food or sucrose) and the corresponding lever was available to press, and rats were required to lever-press to receive reward delivery. A sub-session ended either when the rats were able to acquire 10 rewards or when 10 min of the sub-session had elapsed, whichever occurred first. The associations were: left lever-press (A1) for sucrose delivery (O1) and right lever-press (A2) for food delivery (O2). For the first 2 d, rats were trained on the FR1 schedule, where one lever press resulted in one reward delivery. Subsequently, the rats were trained on RR5 ($p = 0.2$ for 3 d), RR10 ($p = 0.1$ for 3 d) and RR20 ($p = 0.05$ for 3 d) schedules.

**Reversal training (4 d).** This was conducted as described in initial training (above), except that left lever presses (A1) resulted in food delivery (O2), and right lever presses (A2) resulted in sucrose delivery (O1). The training employed the RR20 ($p = 0.05$) schedule.

**Outcome-specific devaluation (2 d).** To induce satiety, rats were given free access to either food or sucrose (selected randomly) for 1 h prior to testing in new cages kept in a new room. After feeding, rats were placed in the operant chamber. The house light was turned on, and their total numbers of left or right lever presses were recorded for 10 min, with neither resulting in reward delivery. This was repeated on the following day, except that the rats were pre-fed with the other reward.

## Extinction learning
After the second devaluation test, rats were retrained on the reversed A-O contingencies at RR20 schedule (Fig. 6g) without laser stimulation until they exhibited stable performance for at least 2 d. During extinction training (2 sessions), rewards were

removed from the operant setup, while everything else in the training protocol remained the same. Optical stimulation (590 nm, 5 s) was provided to inhibit DMS CINs when the reward was supposed to be delivered.

## Reversal learning in mice (Fig. 1i–k)
The procedure was adapted from Matamales et al.[31].

**Apparatus design.** The mice operant chambers (Med Associates) contained a house light, a white noise speaker, and two retractable levers on the right wall. Between the two levers, there was a single food magazine that could deliver either grain-based (20 mg; Dustless Precision Pellets Rodent, Grain-Based; Bio-Serv) or purified food pellets (20 mg; Dustless Precision Pellets Rodent, Purified; Bio-Serv). The training protocols were written and controlled using the Med-PC software. Initial training contingencies: left lever-press (A1) for purified pellets (O1) and right lever-press (A2) for grain-based pellet delivery (O2). Reversal training contingencies: A2→O1, A1→O2.

Procedures for food deprivation, magazine training, timeline of training schedule, and outcome-specific devaluation testing were the same as specified in the section *Reversal Learning in Rats* (Figs. 1a–h and 6a–i).

**Initial training (11 d; A1→O1, A2→O2).** Each training session had 2 consecutive sub-sessions (one for each reward type). The order of the sessions (grain/purified first) was randomly selected each day. During each sub-session, only one type of reward and the corresponding lever was available to press, and mice were required to lever-press to receive a reward delivery. A sub-session ended either when the mice could acquire 20 rewards or when 30 min of the sub-session had elapsed, whichever occurred first.

**Reversal training (4 d; A2→O1, A1→O2).** This was conducted as described in the initial training (above).

## Reversal learning in mice (Supplementary Fig. 6)
The operant chambers (Coulbourn Instruments, MA) contained a house light, a white noise speaker, and two nose-poke holes (one on the left wall, A1 and the other on the right wall, A2). Similarly, it had two magazines for reward delivery (a food magazine (O1, grain-based) on the left wall and a sucrose magazine (O2) on the right wall). Initial training contingencies: A1→O1, A2→O2. Reversal training contingencies: A2→O1, A1→O2. CNO (3–5 mg/kg) was administered 30 min before each reversal training session. All other procedures were the same as specified in the section *Reversal Learning in Mice* (Fig. 1i–k). The only difference being that mice were required to nose-poke rather than lever-press to acquire rewards.

## Intermittent-access to alcohol two-bottle choice drinking
One bottle containing drinking water and one bottle containing 20% alcohol were introduced to the mouse home cages at noon on Mondays, Wednesdays, and Fridays. The alcohol bottle was replaced with a bottle containing only drinking water at noon on Tuesdays, Thursdays, and Saturdays. This schedule was maintained for 8 weeks, following which we measured the amount of alcohol each mouse drank per day. Electrophysiology recordings from alcohol-exposed mice were performed 1 d after the last alcohol exposure.

## Histology
Mice and rats were anesthetized and perfused intracardially with 4% paraformaldehyde (PFA) in phosphate-buffered saline (PBS). The brains were then extracted and submerged in 4% PFA/PBS solution for one day at 4 °C, following which they were transferred to a 30% sucrose solution in PBS. Once the brains completely sank in the sucrose solution, they were cut into 50-μm thick sections using a cryostat.

The slices were stored in a PBS bath at 4 °C prior to mounting on slides for imaging using a confocal laser-scanning microscope (Fluoview-1200, Olympus). All images were processed using Imaris 8.3.1 (Bit-plane, Zurich, Switzerland). ChAT-positive neurons were identified by staining with the anti-ChAT antibody[72] (EMD milliipore AB144P, LOT #2947408, dilution 1:200 (2 nights)) and labeled with far-red fluorescence (647 nm) using a donkey anti-goat antibody[73] (Invitrogen A21447, LOT #2273668, dilution 1:500 (2 nigh)).

## Fiber photometry

The fiber-photometry setup was adapted from Cui et al.[1]. In a 30-min sucrose self-administration session, the spectrum data was recorded continuously at 10 Hz. 488 nm laser was delivered to excite iAChSnFR. The percentage $\Delta F/F$ was calculated by $100 \times (F - F_{mean})/F_{mean}$, where $F_{mean}$ was the mean fluorescence intensity. At the same time, behavior data were collected. Fiber-photometry data was collected using OceanView 1.6.7. To analyze the fiber-photometry data in the context of rat behavior, MATLAB scripts were developed.

## Data analysis and statistics

Finalized data was organized in MS Excel 16.0 and graphed using Origin Pro 2019 (Origin lab). Each experiment was repeated in 3–7 animals, and the resulting data were pooled for statistical analysis. The minimum sample size requirement for electrophysiology recordings was set at 3 (animals). Outliers and unstable recordings with a change in series resistance of more than 10% were excluded from the analyses. For behavioral experiments, the control and experimental groups were age-matched. Data collection was randomized. No data were excluded unless stated otherwise. All data are expressed as mean ± s.e.m. Statistical analyses were performed using SigmaPlot 12.0 (Systat Software Inc.). Normal distribution was tested and unpaired $t$ test, paired $t$ test or two-way RM ANOVA followed by *post-hoc* Tukey test were used to determine statistical significance as appropriate, with an alpha value of 0.05. Mixed model ANOVA followed by simple effects test was conducted for Fig. 1f, h, j, k and Fig. 6e, h in SPSS 29.0[30]. All statistical tests conducted in this study were two-sided.

In Fig. 3, fluorescent cells were counted manually in Imaris. We counted the total presynaptic dMSNs, presynaptic iMSNs, and presynaptic CINs in 4-5 DMS coronal sections in both the left and right hemispheres and calculated their respective percentages. This gave us 8–10 data points (respective percentages) per mouse. Then, we calculated the averaged percentages of presynaptic dMSNs, iMSNs, and CINs for each mouse. Since we had 4 animals, this gave us 4 data points for pre-dMSNs/iMSNs/CINs. We then perform a statistical comparison between the 4 presynaptic dMSN percentages and the 4 presynaptic iMSN percentages (Fig. 3i).

## Reporting summary

Further information on research design is available in the Nature Portfolio Reporting Summary linked to this article.

# Data availability

Source data are provided with this paper.

# Code availability

The code used for analysis of fiber-photometry data in the current study is available online at the Zenodo public repository [https://doi.org/10.5281/zenodo.7948766][74].

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

## Acknowledgements

We thank Giles Johnson and Arian Rivera for technical support and Dr. David Lovinger for his comments on the manuscript. This research was supported by NIAAA R01AA027768 (J.W.), U01AA025932 (J.W.), NIDA R01DA046457 (R.J.S.), and by an X-grant from the Presidential Excellence Fund at Texas A&M University (J.W.). Figures 1a left, 1i, 2b, l, Supplementary Fig. 6c, d, m were created with BioRender.com.

## Author contributions

J.W. and H.G. conceived the project and designed the experiments. H.G., X.X., and Y.H. conducted the behavioral experiments and analyzed the corresponding data. H.G., Z.H, J.L. and X.Z. performed the electrophysiological experiments and analyzed the corresponding data. R.C. and H.G. carried out the fiber-photometry experiments and analyzed the corresponding data. Y.C., H.G., and J.W. performed the histology experiments. X.W. was responsible for animal breeding in all experiments. H.G., J.W., and A.E. wrote the manuscript with significant contributions from R.J.S. and L.N.S.

## Competing interests

The authors declare no competing interest.
