## [Peer Review File · Nature Communications]

Drug Reinforcement Impairs Cognitive Flexibility by Inhibiting Striatal Cholinergic NeuronsReviewers' comments:

Reviewer #1 (Remarks to the Author):

This is a generally well done and interesting paper exploring how cocaine, alcohol, and more generically 'reinforcement' can drive adaptive changes in cholinergic neuron function to alter cognitive flexibility. The authors use a number of converging approaches to test their model. I have a few concerns, which relate to the low n across multiple experiments.

1) The n for some of the ephys experiments is low for animals (as low as $n = 2$ for figure 4 panel D, Figure 5 panel A) It would be helpful to know what cells come from each animal to see if the changes that are seen are driven by single animals. I am sensitive to the amount of time it takes to generate these mice, but also, I have concern that 2 animals provides insufficient support for the hypothesis. Further in panel 5I, 4 cells are taken from a single mouse. this is too low of a sample size.

2) For statistical comparison of the rabies mapping, it appears the authors used 'brain slices' as the basis of comparison vs. animals. This would artificially create an n of 35 instead of 4. This seems like a non rigorous way to perform this analysis, and I suggest the authors use a different approach that would take in to account this is only an n of 4.

Reviewer #2 (Remarks to the Author):

This is an interesting paper which examines changes in the activity of cholinergic interneurons (CINs) in the dorsomedial striatum after exposure to cocaine and alcohol. The authors convincingly demonstrate that the activity of these neurons is decreased by this exposure and that the decrease is in some part due to the activity of direct pathway striatonigral GABAergic projection neurons (dSPNs) that send collaterals to the CINs. They examine projections from these neurons and indirect pathway SPNs to CINs and find the former maintain stronger projections and produce a stronger inhibitory influence on activity. They use monosynaptic rabies virus tracing, DREADDs, optogenetics, in combination with electrophysiology to come to these conclusions which for the most part seems sound. They also examine the effects of drug self-administration and find similar effects to noncontingent exposure.

The main weakness of the paper is the attempt subsequently to confirm that these effects have functional relevance. It has previously been shown that CINs, and indeed dSPNs, play an important role in goal-directed action, particularly after the specific action-outcome contingencies change; e.g., in extinction or when the outcome identities associated with specific actions are reversed. The authors want to use a similar task to assess this same question after CIN manipulations but, unfortunately, use one that is highly ambiguous. They do not implement previous tasks in the same way and this raises a great many issues.

In their task they train mice to perform two distinct nose poke (NP) responses, one earning a sucrose outcome and the other a food pellet outcome. However, these outcomes (O1 and O2) are presented at separate food delivery sites: e.g., NP1O1 at site 1, which is on one side of the box, and NP2-O2 at site 2, which is on the other side of the box; see Figure 1C. This means that when the outcome identities are reversed: NP1-O2 and NP2-O1 the outcome identities change but so does the location of the outcomes relative to the action; i.e., the spatial configuration of the action-outcome retrieval sequence is completely altered. All of the effects the authors describe using this task could, therefore, reflect effects on this sequence performance. In addition, the authors go to some pains to report magazine entries (entries to the food delivery sites) but do so without separating entries to the delivery site for O1 and for O2. The pattern of changes in entries could simply reflect the confusion as to the spatial layout of the box or the location of specific rewards and have nothing to do with goal-directed action.

Finally, most of the inference that this tasks and the effects of reversal and extinction influence

"cognitive flexibility" is made from changes in the pattern of performance whereas previous studies that the authors appear to be modelling (e.g., Matamales et al, 2016; Bradfield et al, 2013) used lever pressing into a central magazine into which both outcomes were delivered and employed tests such as outcome devaluation to assess changes in the encoding of the relationship between action and outcome.

There are significant issues here: The DREADDs and halorhodopsin experiments manipulating CIN activity are not designed to examine the effects of reversal learning on action-outcome association, just changes in performance. Thus, statements about contingency learning are not relevant. The increased magazine entries could reflect perseveration but could also reflect a failure to learn the change spatially. NOTE that spatial relationships between nosepoke and magazine entry are closely related before the shift but not after the shift. Into which magazine are the entries recorded? Both or only the 'correct' magazine? The identity of the outcome and its spatial location are confounded in this task.

The authors need to present more data on the sites of magazine entry and its relationship to lever pressing and be more cautious and thoughtful in the assessment of the functional effects of their manipulations of CINS and in their conclusions.

Other issues:

1. The word reinforcement is used very loosely. What do the authors mean by it? The use of the term 'reinforcement' appears to be used as if it is synonymous with 'increase in performance' rather than strengthening stimulus-response associations (which is its more usual meaning). Thus, the rationale for this study is very vague. dSPN activity changes the degree of performance. This could be true of any reward or any treatment that affects dSPN activity. Thus, the fact that it is achieved by a D1 agonist cocaine is no guarantee that it has anything to do with drug addiction. To show that the authors need to compare changes in activity to cocaine and alcohol with that induced by other rewards.

2. The authors state that "CINs primarily receive GABAergic inputs and extensive local afferents from the striatum itself which is composed of GABAergic dMSNs and indirect-pathway MSNs (iMSNs). This is true of GABAergic inputs but of course there are many other influences on these neurons that are not described, including glutatergic, dopaminergic and modulatory effects of enkephalin, substance P and so on. Pausing in CINs has been documented by Surmeier (Ding et al, 2010) following stimulation of thalamic afferents. Is this pause the same as that induced by dSPN activity? This is not discussed. Are those effects due to direct thalamic inputs to dSPNs or to thalamic inputs to CINS?

3. The final section of the results documenting collaterals of striatonigral neurons should be presented earlier as a part of the mapping of dSPN inputs to CINS. It seems out of place at the end of the results.

4. The discussion focusses on chronic vs acute exposure to drugs although 7 days of non-contingent exposure is hardly chronic. However, what would have been more interesting and welcome would be some consideration of what the consequences for drug induced maladaptive excitation of dSPNs and consequent suppression of CIN activity should be thought to be for the circuitry as a whole in an addicted animal. What effects should CIN suppression be thought to have on local circuitry and how should these effects be thought to generate the specific maladaptive effects of drug addiction (or at least the effects of drug exposure)? This question is not addressed and seems to hang as a major issue given the authors' rationale.

Reviewer #3 (Remarks to the Author):

This manuscript looks to explore the role of medium spiny neuron inhibitory regulation of cholinergic interneurons (CINs) in the dorsomedial striatum (DMS) in cognitive flexibility with a secondary focus on how this inhibitory regulation is impacted by cocaine and alcohol. Results indicate that direct pathway MSNs more potently regulate CINs and that cocaine and alcohol potentiate this inhibition. Further, as reported, the authors suggest that dMSN inhibition of CINs

suppress cognitive flexibility in an instrumental reversal learning task. Overall the manuscript is highly interesting and provides novel insight into the role of CIN modulation of behavior. Further, it provides novel and interesting data that support a cell-type specific modulation of CINs. The manuscript is a little sporadic with how drugs are administered (IVSA vs i.p.) what drugs are tested (cocaine vs EtOH), whether studies are done in mice or rats. While these aren't necessary concerns, it does make for an often difficult read. One way to deal with this would be reorganization (see comments below) as well as more clear experimental schematics for each figure or within panels. Further, there are numerous instances where information is missing that draws significant concern about result interpretation and reduces overall enthusiasm for the manuscript.

1. It isn't clear how the behavioral assessment is measuring cognitive flexibility as described. The data only show that there is a reduction in total nose poking following cocaine. At best that might suggest a reduction in motivation for two types of appetitive rewards. It is assumed that the mice nose poke more for the sucrose versus food reward prior to but there is no data on that. If for example it was their preference and they showed perseverant nose poking on the previous sucrose port that might be suggestive of inflexibility but even then, not definitive as we don't know how cocaine impacts motivational states or hedonics. This is a major design flaw of the study as it stands. A secondary concern is that the authors state that initial learning is not influenced. How is learning being measured? The animals do in fact learn that a nose poke delivers but beyond that this is simple conditioned responding and to some degree an initial measure of approach motivation and subsequent motivation for a reward based on the use of a random ratio schedule of reinforcement.

Similar issues regarding either the model or clarification about the data/model for experiments presented in figure 6

2. The inclusion of IVSA and a 10-d withdrawal period after IVSA seems out of place. While it is certainly nice to see that contingent and non-contingent cocaine exposure produce similar adaptations in firing and inhibitory signaling selection of the timepoint needs justification. Do the authors have behavior data in these mice as well? If so, and it is similar to the non-contingent this would significantly increase a causal link. Providing the total mean cocaine intake in a dose manner would be helpful to compare with the amount of cocaine these mice receive versus the non-contingent.

3. The inclusion of the EtOH data in figure 2 also seems like an afterthought. It is certainly interesting that similar plasticity is in place but without assessment of firing (as well as a behavioral effect that is similar) conclusions that EtOH produce the same adaptations in ChAT-IN can't be made. What was the justification for allowing mice to drink for 8 weeks?

4. There are numerous points that need clarifying with regard to the slice physiology.

It would seem entirely unlikely that a K-gluconate based internal solution would be used for collection of spontaneous and evoked IPSCs. Please clarify which solutions were used for IPSCs and A/N ratio recordings.

What was the justification for a 200 ms interval being the only interval chosen? Typically PPR measure use at least 2 or 3 time points to gain a better picture of the release probability.

The authors need to provide the mean input-output curve data for the Chrimson oIPSCs.

The n-size for the oIPSCs in panel indicate that only 2 cells were collected from 2 mice. This is too low to make any meaningful statistical conclusion.

Additional information is needed with regard to how the NMDAR components of the EPSC at +40 was extracted to ensure AMPAR components were also not included in the amplitude.

What was the rationale for applying AMPA to the slice for only 15 seconds. It would be helpful if the authors provided a citation where this approach to assess AMPAR-specific signaling was used as it is not a typical approach compared to AMPAR I/O curves or sEPSCs. Less of an issue but there seems to be significant noise in the representative traces.

Why was the red-shifted Chrimson opsin used for data collected in Figure 5a-e but the Ai32 line was used in studies related to alcohol administration in Figure 5f?

While data showing that cocaine self-administration augments dMSN-dependent IPSCs is interesting, it does not definitively show that the cocaine potentiation of AMPAR signaling in dMSNs is the driving factor without simulating glutamate release at dMSNs and recording from CINs. Notably, this doesn't detract from the findings, but the authors should consider rewording the results/interpretation

The impact of dMSN stimulation was assessed but not stimulation of iMSNs. This would seem like an important piece of information to have. While iMSNs clearly do not innervate the CINs as much as dMSN they still do provide inhibitory input. Thus knowing whether or not this inhibitory input is also sufficient to inhibit CINs or if it is selective for dMSNs.

5. No details regarding the approach used to quantify inputs to CIN are given. How these data were quantitatively collected is important.

6. No information is provided about the background (or anything for that matter) of the mice and rats. Are they all male or a mix of male and female? What was the breeding strategy for the double transgenics and where were they obtained from? What age were they? Etc.

7. There are no controls for in vivo optogenetic studies show in figure 5.

8. It isn't clear what virus was used as a control for chemogenetic inhibition studies.

9. Figure 6a shows representative spike firing but it isn't clear what that is? Is it ex vivo recordings?

10. Please clarify what was used for control in chemogenetic inhibition studies.

11. What control virus or parameters were used for "controls" in the optogenetic inhibition studies.
Minor:

1. While I personally don't think the inclusion of the EtOH data is helpful, if the authors feel the need to keep it in, the manuscript would benefit from the reorganization by grouping EtOH-related studies into a separate figure/results section rather than randomly inserted among cocaine studies in its current format.

We thank the reviewers for their thoughtful comments and appreciate the opportunity to revise the manuscript. We have carefully revised the manuscript and provides our detailed responses below. The original concerns expressed by the reviewers are shown in *italics*.

Reviewer 1

This is a generally well done and interesting paper exploring how cocaine, alcohol, and more generically 'reinforcement' can drive adaptive changes in cholinergic neuron function to alter cognitive flexibility. The authors use a number of converging approaches to test their model. I have a few concerns, which relate to the low n across multiple experiments.

Concern 1

1) The n for some of the ephys experiments is low for animals (as low as n = 2 for figure 4 panel D, Figure 5 panel A). It would be helpful to know what cells come from each animal to see if the changes that are seen are driven by single animals. I am sensitive to the amount of time it takes to generate these mice, but also, I have concern that 2 animals provides insufficient support for the hypothesis. Further in panel 5I, 4 cells are taken from a single mouse. this is too low of a sample size.

Response: We agree with the reviewer and have increased the sample size of animals per group to at least 3 for all electrophysiology experiments. Additionally, we have confirmed that we include a minimum of 2-4 neurons per animal for analysis to ensure that single animals do not drive our results and conclusions. In the following table, we list all the figures and the corresponding sample size in the new manuscript.

Table 2. Summary of figure panels and corresponding sample size

Figures	Approaches	Group 1 (Neuron#, Animal#)	Group 2 (Neuron#, Animal#)
1a-h	Behavior	Saline (9 rats)	Cocaine (7 rats)
1i-k	Behavior	Saline (11 mice)	Cocaine (12 mice)
2e	Electrophysiology	Saline (11,3)	Cocaine (10,3)
2f	Electrophysiology	Saline (10,3)	Cocaine (11,4)
2h-i	Electrophysiology	Saline (18,4)	Cocaine (20,4)
2j-k	Electrophysiology	Saline (26,3)	Cocaine (23,3)
2m	Electrophysiology	Saline (11,4)	Cocaine (12,4)
2n-o	Electrophysiology	Saline (13,4)	Cocaine (17,4)
2p-r	Electrophysiology	Saline (9,4)	Cocaine (9,4)
3a-i	Histology	4 mice (within-subject comparison)	
3l	Electrophysiology	(8,3) (no comparison)	
4c	Electrophysiology	dMSN→CIN (13,5)	iMSN→CIN (16,5)
4e	Electrophysiology	8 neurons (within neuron comparison)	
4g	Electrophysiology	8 neurons (within neuron comparison)	
4j	Electrophysiology	dMSN→CIN (10,3)	iMSN→CIN (14,5)
4l-o	In-vivo imaging	5 rats (within-subject comparison)	
5b	Electrophysiology	Saline (18,4)	Cocaine (22,5)
5c	Electrophysiology	Saline (9,3)	Cocaine (8,3)
5d	Electrophysiology	Saline (22,5)	Cocaine (25,7)
5e	Electrophysiology	Saline (10,3)	Cocaine (14,4)
5g	Behavior	Sucrose (7)	Cocaine (7)
5h	Electrophysiology	Sucrose (20,5)	Cocaine (22,5)
5j-k	Electrophysiology	Water (15,3)	Alcohol (16,4)
5l	Electrophysiology	Water (6,4)	Alcohol (5,3)

5n	Behavior	Control (6)	dMSN-oICSS (6)
50	Electrophysiology	Control (14,5)	dMSN-oICSS (13,6)
6	Behavior	Control (9)	hM4Di (9)
7	Behavior	Control (8)	eNpHR (8)

Reviewer 1, concern 2

2) For statistical comparison of the rabies mapping, it appears the authors used 'brain slices' as the basis of comparison vs. animals. This would artificially create an n of 35 instead of 4. This seems like a non rigorous way to perform this analysis, and I suggest the authors use a different approach that would take in to account this is only an n of 4.

Response: We appreciate the reviewer’s suggestion and have made the following change (Fig. 3i). For each animal, we first calculated the averaged percentages of presynaptic dMSNs, iMSNs, and CINs. Since we have 4 animals, this gives us 4 data points for presynaptic dMSNs, iMSNs, and CINs. We then performed a statistical comparison between the 4 presynaptic dMSN percentages and the 4 presynaptic iMSN percentages. This led to a comparison among 4 data points (one data point representing one mouse) rather than 35 data points (previous data where each data point represented a slice). This revision did not affect our statistics and conclusion that CINs receive striatal inputs that mainly originate from dMSNs (Fig. 3i).

Reviewer 2

Concern 1

This is an interesting paper which examines changes in the activity of cholinergic interneurons (CINs) in the dorsomedial striatum after exposure to cocaine and alcohol. The authors convincingly demonstrate that the activity of these neurons is decreased by this exposure and that the decrease is in some part due to the activity of direct pathway striatonigral GABAergic projection neurons (dSPNs) that send collaterals to the CINs. They examine projections from these neurons and indirect pathway SPNs to CINs and find the former maintain stronger projections and produce a stronger inhibitory influence on activity. They use monosynaptic rabies virus tracing, DREADDs, optogenetics, in combination with electrophysiology to come to these conclusions which for the most part seems sound. They also examine the effects of drug self-administration and find similar effects to noncontingent exposure.

The main weakness of the paper is the attempt subsequently to confirm that these effects have functional relevance. It has previously been shown that CINs, and indeed dSPNs, play an important role in goal-directed action, particularly after the specific action-outcome contingencies change; e.g., in extinction or when the outcome identities associated with specific actions are reversed. The authors want to use a similar task to assess this same question after CIN manipulations but, unfortunately, use one that is highly ambiguous. They do not implement previous tasks in the same way and this raises a great many issues.

*In their task they train mice to perform two distinct nose poke (NP) responses, one earning a sucrose outcome and the other a food pellet outcome. However, these outcomes (O1 and O2) are presented at separate food delivery sites: e.g., NP1 → O1 at site 1, which is on one side of the box, and NP2-O2 at site 2, which is on the other side of the box; see Figure 1C. This means that when the outcome identities are reversed: NP1-O2 and NP2-O1 the outcome identities change but so does the location of the outcomes relative to the action; i.e., the spatial configuration of the action-outcome retrieval sequence is completely altered. All of the effects the authors describe using this task could, therefore, reflect effects on this sequence performance (**see Response #1 below**). In addition, the authors go to some pains to report magazine entries (entries to the food delivery sites) but do so without separating entries to the delivery site for O1 and for O2 (**see Response #2 below**). The*

pattern of changes in entries could simply reflect the confusion as to the spatial layout of the box or the location of specific rewards and have nothing to do with goal-directed action (**see Response #3 below**).

Finally, most of the inference that this tasks and the effects of reversal and extinction influence “cognitive flexibility” is made from changes in the pattern of performance whereas previous studies that the authors appear to be modelling (e.g., Matamales et al, 2016; Bradfield et al ,2013) used lever pressing into a central magazine into which both outcomes were delivered and employed tests such as outcome devaluation to assess changes in the encoding of the relationship between action and outcome.

There are significant issues here: The DREADDs and halorhodopsin experiments manipulating CIN activity are not designed to examine the effects of reversal learning on action-outcome association, just changes in performance. Thus, statements about contingency learning are not relevant (**see Response #4 below**). The increased magazine entries could reflect perseveration but could also reflect a failure to learn the change spatially. NOTE that spatial relationships between nosepoke and magazine entry are closely related before the shift but not after the shift. Into which magazine are the entries recorded? Both or only the ‘correct’ magazine? The identity of the outcome and its spatial location are confounded in this task (**see Response #5 below**). The authors need to present more data on the sites of magazine entry and its relationship to lever pressing and be for more cautious and thoughtful in the assessment of the functional effects of their manipulations of CINS and in their conclusions (**see Response #6 below**).

Response #1: We thank the reviewer for raising this concern. At the time of the experiment, we only had access to mouse operant chambers from Coulbourn Instruments, in which food and sucrose dispensers were on opposite walls (Fig. 1A below). However, our lab now has mouse operant chambers from Med Associates (Fig. 1B below), and we exactly emulated the 2-food reversal-learning task as specified in Matamales et al. (2020; 2016). In these new chambers, one food dispenser is located in between two pressable levers. Using this equipment, we conducted a **new experiment** to study the effect of cocaine administration on reversal learning in mice (Fig. 1i-k, Supplementary Fig. 1). We found that although both groups of mice were sensitive to outcome devaluation after initial training (Fig. 1j), only the saline-exposed mice, but not the cocaine-exposed animals, were sensitive to outcome devaluation after reversal training (Fig. 1k). These results suggest that cocaine exposure disrupte reversal learning in mice and that the spatial configuration in Figure 6 (and Old Fig. 1) unlikely contributes to a cocaine-induced reversal learning deficits.

As an alternative, we also used rat operant chambers from Coulbourn Instruments (Fig. 1C below), in which the food and sucrose dispensers were adjacent to each other and were on the same side of the wall between the levers; these chambers were used our recent study on reversal learning deficits (Fig. 1 and 8 in Ma et al., 2022). Using these chambers, we conducted another new experiment to study the effect of cocaine administration on reversal learning in rats (Fig. 1a-h). Again, we found that cocaine exposure disrupted reversal learning in rats (Fig. 1h). Please note that we also used these chambers for Figure 7.

Using both of these two tasks, we eliminated the issue of a drastic change in spatial configuration as highlighted by the reviewer and still found that cocaine impaired reversal learning in mice and rats. Additionally, in Figure 6 where we chemogenetically inhibited CINs during reversal training, we have **added 15 new panels** of data analysis to support our conclusions, which is discussed in detail in our Response #2-6 to Reviewer 2, Concern 1. In summary, we have made the following updates in the manuscript (MS) for our behavior experiments.

Table 3. Hypothesis and the presence of spatial change in revised manuscript (MS).

Figure	Questions Addressed	Spatial Change (Old MS)	Spatial Change (Revised MS)	Data Provided
1a-h	Does cocaine administration disrupt reversal learning in rats?	Not tested	No	Devaluation
1i-k	Does cocaine administration disrupt reversal learning in mice?	Yes	No	Devaluation

6	Does chemogenetic CIN inhibition disrupt reversal learning?	Yes	Yes	Devaluation, Performance
7	Does optogenetic CIN inhibition disrupt reversal learning?	No	No	Devaluation

Figure 1. **A.** Mouse chamber used in Old manuscript Figures 1 and 6. 1) during initial training, nose-poking on the left hole (A1) led to delivery of a food pellet (O1; A1→O1), while nose-poking on the right hole (A2) led to delivery of sucrose solution (O2; A2→O2). 2) reversal training, A1→O2 and A2→O1. **B.** New mouse chambers were used to repeat the experiment in Old manuscript Figure 1. 1) during initial training, pressing on the right lever (A1) will result in the delivery of a grain pellet (O1; A1→O1) and pressing on the left lever (A2) will result in the delivery of a purified pellet (O2; A2→O2). Both O1 and O2 will be delivered in the same magazine which is located between two levers on the same wall. 2) during reversal learning, A1→O2 and A2→O1. Since O1 and O2 are delivered in the same port, there is no spatial issue. **C.** Rat operant chamber used in Figures 1 & 7. The food-pellet port and sucrose-solution port are adjacent and located between two levers on the same wall.

Response #2: Figure 6 shows that the mice displayed goal-directed learning (Fig. 6f) as well as high motivation (Supplementary Fig. 6a) to nose-poke for the food reward (O1) versus the sucrose reward (O2) during initial training. Therefore, we specifically analyzed the action-outcome associations for food (O1) in the whole experiment. Thus, the reported magazine entries are specific for O1 (Fig. 6i). We have now also added the data corresponding to the magazine entries at delivery sites for O2 during the reversal training in Supplementary Figure 6e.

In **Figure 7**, both food and sucrose magazines were adjacent to each other on the same side of the wall and shared a common sensor. Therefore, it is not possible to separate the entries for O1 and O2 (Supplementary Fig. 7d). However, for this experiment, we used outcome-specific devaluation after reversal training to conclude that optogenetic CIN inhibition impaired reversal learning (Fig. 7h, i).

Response #3: We appreciate the reviewer’s concern about our result interpretation in Figure 6. To investigate the involvement of DMS CINs in regulating reversal learning, we first infused AAV-DIO-hM4Di-mCherry in the DMS of ChAT-Cre mice, whereas the controls were injected with AAV-DIO-mCherry. Six weeks later, both groups underwent training on a two-action-outcome reversal learning task (Fig. 6c-d). During initial training, animals were required to nose-poke at the A1 site and collect food rewards from the O1 delivery site (A1→O1). However,

upon contingency reversal, animals were required to nose-poke at the A2 site and collect food rewards from the O1 delivery site (A2→O1). CNO was administered 30 min before each reversal training session. In reversal session 1, we observed the following:

- 1) Total food magazine entries (O1) were higher in the hM4Di group than in controls (Fig. 6i). Additionally, the return map of inter-entry (O1) intervals for the hM4Di group was clustered near the origin (Fig. 6k), and consequently, the mean inter-entry interval for the hM4Di group was lower than controls (Fig. 6l). These data suggest that chemogenetic CIN inhibition causes animals to frequently check for rewards via excessive magazine entries upon contingency reversal. A similar increase in head entries during instrumental tasks has been linked to perseveration and behavioral inflexibility in other studies (Lhost et al., 2021; Son et al., 2013).
- 2) Reversal training required animals to learn the A2 (nose-poke) → O1 (entry) association. To quantify this association, we counted the number of times an animal entered the O1 magazine within 10 sec of a nose poke at the A2 site (Fig. 6n). We found that the number of such entries was lower in the hM4Di group than in controls (Fig. 6o). This result suggests that the hM4Di group was slower in learning the A2 (nose-poke) → O1 (entry) association than controls.
- 3) The control animals immediately entered the magazine to collect food rewards upon delivery, whereas the hM4Di group failed to do so (Fig. 6q). We calculated the percentage of rewards collected (reward delivery followed by at least 1 magazine entry) over total rewards delivered and found a lower percentage for the first 2 reversal training sessions in the hM4Di group than in the controls (Fig. 6r). However, by the 4th session, both groups exhibited similar percentages (Fig. 6r). Similar deficits in reward collection behavior have previously been linked to behavioral inflexibility (Lhost et al., 2021).

Since CNO was administered during all reversal training sessions, the aforementioned effects are unlikely to be due to any direct locomotion effects of chemogenetic CIN inhibition. To summarize, as the reviewer suggested, we have performed additional analysis to support our findings and revised the interpretation of our results (Fig. 6).

Response #4: We agree with the reviewer that the outcome devaluation test is critical to assess action-outcome association learning. Accordingly, in the revised manuscript, we only use outcome devaluation data to make claims regarding any deficits in contingency learning. We have performed four reversal learning experiments in the study (Fig. 1a-h, 1j-1k, 6, and 7). Their results are summarized in Table 3 below.

- **Figure 1a-h.** We conducted outcome devaluation tests after initial (Fig. 1f) and reversal training (Fig. 1h) in rats. Both saline- and cocaine-exposed rats were sensitive to outcome devaluation after initial training. However, after reversal training, cocaine-exposed animals were not sensitive to outcome devaluation, but the saline animals could, suggesting that cocaine leads to reversal learning deficits in rats.
- **Figure 1i-k.** This is similar to Figure 1a-h, but we instead used mice and the 2-food reversal learning task for the experiment. We found that cocaine-exposed mice were not sensitive to outcome devaluation after reversal training, but the saline animals were, suggesting that cocaine leads to reversal learning deficits in mice (Fig. 1k).
- **Figure 6.** We conducted devaluation tests both after initial (Fig. 6f) and reversal training (Supplementary Fig. 6f). Interestingly, we found that both control and hM4Di groups were sensitive to outcome devaluation after initial and reversal training, suggesting that chemogenetic CIN inhibition only disrupts the early learning of new action-outcome associations in mice. Please also see our response #3 to Reviewer 2, concern 1.
- **Figure 7.** We conducted devaluation tests after initial (Fig. 7e, f) and reversal training (Fig. 7h, i). The control group, but not the eNpHR group, was sensitive to outcome devaluation after reversal training, suggesting that optogenetic DMS CIN inhibition impairs reversal learning in rats.

Regarding the effect on extinction learning, multiple studies have used instrumental performance as a measure to assess extinction (Chen et al., 2016; Matamalas et al., 2020; Shalev et al., 2001). However, we have performed additional within-session analysis to support our result (Fig. 7l, m). This **new analysis** found that

there was significantly higher lever pressing in the halorhodopsin rats than in controls during the 10 – 20 min segment of the extinction session, suggesting delayed extinction learning in the eNpHR group.

Table 3. Summary of devaluation tests in reversal learning experiments

Fig	Treatment	1st devaluation	2nd devaluation
1a-h	Cocaine	Saline and cocaine-exposed rats were sensitive to outcome devaluation (Fig. 1f)	Saline, but not cocaine-exposed rats, were sensitive to outcome devaluation. (Fig. 1h)
1i-k	Cocaine	Saline and cocaine-exposed mice were sensitive to outcome devaluation (Fig. 1j)	Saline, but not cocaine-exposed mice, were sensitive to outcome devaluation (Fig. 1k)
6	hM4Di	Control and hM4Di mice were sensitive to outcome devaluation (Fig. 6f)	Control and hMDi4 mice were sensitive to outcome devaluation (Supplementary Fig. 6f)
7	eNpHR	Control and halorhodopsin rats were sensitive to outcome devaluation (Fig. 7e, f)	Control, but not halorhodopsin rats, were sensitive to outcome devaluation (Fig. 7h, i)

Response #5: We observed an increase in magazine entries during reversal training in our chemogenetic (Fig. 6i) and optogenetic manipulation experiments (Supplementary Fig. 7d).

For Figure 6, we agree with the reviewer that the increase in magazine entries could reflect perseveration but could also represent a failure to learn the spatial change. Please note that we observed this increase in entries into the “correct” magazine (O1). As explained in our response #3 to Reviewer 2, Concern 1, and in Figure 6, we conclude 3 points. 1) The inter-entry interval was lower in the hM4Di group than in controls, suggesting frequent, excessive magazine entries (Fig. 6j-l). 2) The number of successful action-outcome associations (Nose poke [A2] → Food magazine entry [O1]) was lower in the hM4Di group than in controls (Fig. 6n, o). 3) Upon delivery, the control group collected the rewards in a timely manner, whereas the hM4Di group did not (Fig. 6q, r). These data suggest that chemogenetic downregulation of DMS CINs transiently leads to 1) frequent, excessive magazine entries, 2) slowed reversal learning, and 3) impaired reward collection upon contingency reversal.

In Figure 7, both food and sucrose magazines were adjacent on the same side of the wall. Thus there is not a drastic spatial shift upon reversal in this experiment. Since food and sucrose magazines shared a common sensor, we could only conclude that we saw an increase in total magazine entries in the eNpHR group during reversal training (Supplementary Fig. 7d).

Response #6: We appreciate the reviewer’s suggestions. For Figure 6, we added 15 new panels to demonstrate our results. Specifically, we added details on the sites of magazine entry (Fig. 6i-l, Supplementary Fig. 6d, e), its relationship to nose-poking (Fig. 6m-o), and its relationship to reward delivery during reversal training (Fig. 6q, r). Additionally, we revised our conclusion of the result to “Chemogenetic downregulation of DMS CINs transiently leads to 1) frequent, excessive magazine entries, 2) slowed reversal learning, and 3) impaired reward collection upon contingency reversal.” (Fig. 6).

For Figure 7, we moved all the performance-based data to Supplementary Figure 7 and only kept the initial (Fig. 7e, f) and reversal devaluation data (Fig. 7h, i) in the main figure to conclude that optogenetic CIN inhibition impairs reversal learning in rats.

Reviewer 2, other concern 1

Other issues: 1. The word reinforcement is used very loosely. What do the authors mean by it? The use of the term ‘reinforcement’ appears to be used as if it is synonymous with ‘increase in performance’ rather than strengthening stimulus-response associations (which is its more usual meaning). Thus, the rationale for this study is very vague. dSPN activity changes the degree of performance. This could be true of any reward or any

treatment that affects dSPN activity. Thus, the fact that it is achieved by a D1 agonist cocaine is no guarantee that it has anything to do with drug addiction. To show that the authors need to compare changes in activity to cocaine and alcohol with that induced by other rewards.

Response: We agree with the reviewer that while increasing performance, dMSN excitation does not necessarily cause addiction. Numerous studies found that addictive substances (e.g., cocaine and alcohol) and natural rewards (e.g., sucrose) induce distinct changes in dMSN activity (Bobadilla *et al.*, 2020; Ma *et al.*, 2018). Thus, it is critical to compare addictive substance-induced versus sucrose-induced CIN changes. Since most studies in the manuscript were conducted following cocaine administration, we conducted a **new experiment** to test whether cocaine IVSA or sucrose self-administration elicits distinct changes in dMSN→CIN activity (Fig. 5f-h, Supplementary Fig. 5a). It has been shown that both cocaine and sucrose self-administration can elicit similar plasticity one day after the last respective training session. However, cocaine-induced plasticity is long-lasting (at least 3 weeks), whereas sucrose-induced plasticity is not (Chen *et al.*, 2008). Therefore, we measured GABAergic dMSN→CIN transmission 3 weeks after the last training session and found it to be higher in the cocaine group (Fig. 5f-h). This result suggests that the drug-induced increase in dMSN→CIN transmission is stronger and lasts longer than that induced by natural rewards.

Reviewer 2, other concern 2

*2. The authors state that “CINs primarily receive GABAergic inputs and extensive local afferents from the striatum itself which is composed of GABAergic dMSNs and indirect-pathway MSNs (iMSNs). This is true of GABAergic inputs but of course there are many other influences on these neurons that are not described, including glutamatergic, dopaminergic and modulatory effects of enkephalin, substance P and so on. Pausing in CINs has been documented by Surmeier (Ding *et al.*, 2010) following stimulation of thalamic afferents. Is this pause the same as that induced by dSPN activity? This is not discussed. Are those effects due to direct thalamic inputs to dSPNs or to thalamic inputs to CINS?”*

Response: We thank the reviewer for the constructive comments. We updated the discussion section to include “all the presynaptic inputs to DMS CINs”. Specifically, we added “glutamatergic (Aceves Buendia *et al.*, 2019; Assous *et al.*, 2017; Ding *et al.*, 2010; Doig *et al.*, 2014; Kosillo *et al.*, 2016; Matamales *et al.*, 2016; Sciamanna *et al.*, 2012; Stalnaker *et al.*, 2016; Thorn and Graybiel, 2010), dopaminergic (Brown *et al.*, 2012; Chuhma *et al.*, 2014; Lozovaya *et al.*, 2018; Straub *et al.*, 2014; Tritsch and Sabatini, 2012; Wieland *et al.*, 2014), recurrent (Sullivan *et al.*, 2008), substance-P and enkephalin (Bolam *et al.*, 1986; Francis *et al.*, 2019; Gonzales *et al.*, 2013; Martone *et al.*, 1992) inputs”.

We do not believe that the pause in CIN firing we observed is similar to that reported in Ding *et al.*, (2010). **First**, our pause was induced by GABAergic inhibition, whereas that in Ding *et al.* was mediated by glutamatergic excitation. In Figure 4d, we show that optogenetic stimulation of GABAergic dMSNs causes a pause in CIN firing. This pause was abolished by a GABA_AR antagonist, picrotoxin (Supplementary Fig. 4b). Ding *et al.* conducted their study in the presence of GABA_AR and GABA_BR antagonists. Therefore, they did not study MSN-derived pause in CIN firing. Instead, Ding *et al.* demonstrates a burst-pause response in CINs following stimulation of thalamic inputs, which are glutamatergic. This pause is dopamine D2 receptor-dependent because burst firing of CINs increases the release of ACh, which activates nAChRs on striatal DA terminals and triggers dopamine release to inhibit CINs via their D2Rs (Ding *et al.*, 2010; Liu *et al.*, 2022). **Second**, our inhibition-induced pause was followed by a strong rebound (Fig. 4d, **pause-rebound**), whereas thalamus mediated CIN excitation did not induce rebound after the pause in Ding *et al.* (**burst-pause**). Therefore, our research elucidates an innovative mechanism of CIN inhibition that mediates drug-induced inflexibility. Table 5 below summarizes the difference between Ding *et al.* and our studies. We added part of the above statements in the Discussion section.

Table 5. Differences in burst-pause and pause-rebound in CINs

Burst-pause (Ding et al., 2010)	Pause-rebound (this manuscript)
Induced by glutamatergic thalamic inputs.	Induced by GABAergic dMSN inputs.
No rebound after the pause.	The rebound comes after the pause.
The pause is D2R dependent.	The pause is GABA _A R-dependent.
Pause length is correlated with the number of spikes in the burst.	Pause length is dependent on the strength of inhibitory inputs.

Reviewer 2, other concern 3

3. *The final section of the results documenting collaterals of striatonigral neurons should be presented earlier as a part of the mapping of dSPN inputs to CINS. It seems out of place at the end of the results.*

Response: We appreciate the reviewer’s suggestion and appended the results documenting the collaterals of striatonigral neurons to Figure 3, where we showed that dMSNs preferentially innervate CINs.

Reviewer 2, other concern 4

4. *The discussion focusses on chronic vs acute exposure to drugs although 7 days of non-contingent exposure is hardly chronic (See response #1 below). However, what would have been more interesting and welcome would be some consideration of what the consequences for drug induced maladaptive excitation of dSPNs and consequent suppression of CIN activity should be thought to be for the circuitry as a whole in an addicted animal. What effects should CIN suppression be thought to have on local circuitry and how should these effects be thought to generate the specific maladaptive effects of drug addiction (or at least the effects of drug exposure)? This question is not addressed and seems to hang as a major issue given the authors’ rationale (See response #2 below).*

Response #1: We agree with the reviewer’s critique and replaced the term “chronic” with “the 7-day non-contingent cocaine administration” (3 instances). Additionally, we cited a previous study showing that D2R abundance in NAc CINs is increased after cocaine self-administration to reduce CIN activity (Lee *et al.*, 2020), which supports our conclusion.

Response #2: We thank the reviewer for this valuable suggestion. Previous research (Ding *et al.*, 2010) and our recent study (Ma *et al.*, 2022) found that CIN activation inhibits dMSNs and potentiates iMSNs. During salient environment stimulation, the thalamus sends excitatory projections to DMS CINs, causing a burst in striatal acetylcholine. This ACh burst transiently inhibits cortico-striatal synapses onto both dMSNs (“Go” pathway) and iMSNs (“No-Go” pathway) by binding to presynaptic M2 muscarinic receptors. Subsequently, ACh then preferentially allows cortical activation of D2-MSNs (“No-Go”) via postsynaptic M1 muscarinic receptors (Ding *et al.*, 2010). After drug exposure, the reduced striatal ACh would impair the inhibitory control of CINs on the cortico-striatal synapses after salient stimulation. This, in part, should contribute to the compulsive/inflexible decision-making observed in drug-exposed subjects. ACh also binds to nAChRs on striatal dopaminergic afferents, leading to local dopamine release (Cachope *et al.*, 2012), which not only supports learning but also affects synaptic plasticity of cortico-striatal synapses (Partridge *et al.*, 2002). Thus, a deficiency in ACh signaling after drug exposure could regulate striatal circuits in a variety of ways. We added a part of the above response to the discussion section. We think that exploring how drug-induced maladaptive CIN activity affects the local circuitry is out of scope of this study.

Reviewer 3

Concern 1

This manuscript looks to explore the role of medium spiny neuron inhibitory regulation of cholinergic interneurons (CINs) in the dorsomedial striatum (DMS) in cognitive flexibility with a secondary focus on how this inhibitory regulation is impacted by cocaine and alcohol. Results indicate that direct pathway MSNs more potently regulate CINs and that cocaine and alcohol potentiate this inhibition. Further, as reported, the authors suggest that dMSN inhibition of CINs suppress cognitive flexibility in an instrumental reversal learning task. Overall the manuscript is highly interesting and provides novel insight into the role of CIN modulation of behavior. Further, it provides novel and interesting data that support a cell-type specific modulation of CINs. The manuscript is a little sporadic with how drugs are administered (IVSA vs i.p.) what drugs are tested (cocaine vs EtOH), whether studies are done in mice or rats. While these aren't necessary cons, it does make for an often difficult read. One way to deal with this would be reorganization (see comments below) as well as more clear experimental schematics for each figure or within panels (see Response #1 below). Further, there are numerous instances where information is missing that draws significant concern about result interpretation and reduces overall enthusiasm for the manuscript.

1. It isn't clear how the behavioral assessment is measuring cognitive flexibility as described (see Response #2 below). The data only show that there is a reduction in total nose poking following cocaine. At best that might suggest a reduction in motivation for two types of appetitive rewards (see Response #3 below). It is assumed that the mice nose poke more for the sucrose versus food reward prior to but there is no data on that (see Response #4 below). If for example it was their preference and they showed perseverant nose poking on the previous sucrose port that might be suggestive of inflexibility but even then, not definitive as we don't know how cocaine impacts motivational states or hedonics. This is a major design flaw of the study as it stands (see Response #5 below). A secondary concern is that the authors state that initial learning is not influenced. How is learning being measured? The animals do in fact learn that a nose poke delivers but beyond that this is simple conditioned responding and to some degree an initial measure of approach motivation and subsequent motivation for a reward based on the use of a random ratio schedule of reinforcement (see Response #6 below).

Similar issues regarding either the model or clarification about the data/model for experiments presented in figure 6 (see Response #7).

Response #1: We thank the reviewer for their constructive comments. We improved our schematics and reorganized our manuscript based on reviewer comments which are summarized below.

Table 6. Summary of animals, treatments, and tests in all figures

Figure	Heading	panel	mice /rats	Treatment	IVSA /i.p.	Experiment type
1	Reversal	a-h	rats	cocaine	i.p.	behavior
	Reversal	i-k	mice	cocaine	i.p.	behavior
2	Coc→CIN	a-k	mice	cocaine	i.p.	physiology
	Coc→CIN	l-r	mice	cocaine	IVSA	physiology
3	dMSN/iMSN→CIN	a-i	mice	n/a	n/a	histology
	dMSN→CIN	j-m	mice	n/a	n/a	physiology
4	dMSN/iMSN→CIN	a-j	mice	n/a	n/a	physiology
	ACh imaging	k-o	rats	n/a	n/a	photometry
5	dMSN→CIN	a-e	mice	cocaine	i.p.	physiology
	dMSN→CIN	f-h	mice	cocaine	IVSA	physiology
	dMSN→CIN	i-l	mice	alcohol	oral	physiology
	dMSN→CIN	m-o	rats	dMSN-SA	Optical	physiology
6	Reversal	a-p	mice	n/a	n/a	behavior
7	Reversal	a-n	mice	n/a	n/a	behavior

Response #2: We thank the reviewer for this question. Cognitive flexibility is the ability of an animal to adjust to environmental changes to obtain rewards or avoid punishment. In our behavioral design, the operant chamber consisted of 2 lever presses (Fig. 1c: left, A1; right, A2) and 2 reward dispensers (sucrose, O1; food, O2). Animals were extensively trained to press the left lever to receive a sucrose reward (A1→O1) and the right lever to receive a food reward (A2→O2). Since animals were food restricted, obtaining the food and sucrose rewards in the task was of high value. Once the animals learned these action-outcome associations (A1→O1 and A2→O2), we switched them, i.e., the animals were now required to press the right lever to receive a sucrose reward (A2→O1) and the left lever to receive a food reward (A1→O2).

If the animals were to continue receiving rewards, they would need to inhibit the old behavioral contingencies and learn new associations. An inability to effectively adapt to this association switch would be classified as reduced cognitive behavior flexibility. Additionally, reduced cognitive flexibility after the switch can manifest in multiple ways: A) The animals might give up nose-poking or reward collection. B) The animals might continue using the old behavior strategy, leading to 1) incorrect action-outcome associations (as measured by outcome-specific devaluation tests), 2) erroneous nose pokes, 3) perseverative reward entries, 4) higher latency to seek- or collect- reward, 5) incorrect outcome associations, etc. For these reasons, this design presents a holistic model to study cognitive flexibility.

This behavioral design has been previously used to assess cognitive flexibility in literature (Bradfield and Balleine, 2017; Bradfield *et al.*, 2013; Matamales *et al.*, 2020; Matamales *et al.*, 2016).

Response #3: We agree with the reviewer that a reduction in nose pokes in the cocaine group following reversal is not necessarily a sign of behavioral inflexibility. Behavioral flexibility is the ability to learn new action-outcome contingencies, which is determined by outcome-specific devaluation tests and not only by nose-poke performance (Bradfield and Balleine, 2017; Bradfield *et al.*, 2013; Matamales *et al.*, 2020; Matamales *et al.*, 2016) (please see our response #4 to Reviewer 2, Concern 1). Thus, we conducted 2 new experiments and found that although both saline- and cocaine-exposed rats were able to learn the initial action-outcome contingencies, only the saline group, and not the cocaine group, was able to learn the reversed contingencies (Fig. 1a-h). Additionally, we were able to replicate this result in mice in a similar 2-food reversal learning task (Fig. 1i-k, see our response #1 to Reviewer 2, Concern 1). These data suggest that cocaine impairs cognitive flexibility. Notably, we do not see any significant reduction in the nose pokes anymore, which could be because we used a different apparatus for the experiments in the new manuscript (Fig. 1 above).

Response #4: We thank the reviewer for raising this concern. We found that mice nose-poked more for food than the sucrose reward. The corresponding data are presented in Supplementary Figure 6a.

Response #5: Please see our Response #3 to Reviewer 3, Concern 1 where we address this question.

Response #6: We agree with the reviewer's concern that instrumental response is not a standalone indicator of learning in our task. In the revised manuscript, we have used the outcome-specific devaluation test as a measure of goal-directed learning for all behavioral experiments (please see our Response #4 to Reviewer 2, Concern 1).

Response #7: We thank the reviewer for raising this concern. Regarding the question of whether the behavioral task is suitable for studying cognitive flexibility, please see our Response #2 to Reviewer 3, Concern 1. Additionally, please see our Response #3 to Reviewer 2, Concern 1 for the clarification of our result in Figure 6. To summarize, we have added 15 new panels of analysis in Figure 6 and have 3 main conclusions. During reversal

training, 1) the inter-entry interval was lower in the hM4Di group than in controls, suggesting frequent, excessive magazine entries (Fig. 6j-l). 2) The number of successful action-outcome associations (Nose poke [A2] → Food magazine entry [O1]) was lower in the hM4Di group than in controls (Fig. 6n, o). 3) Upon delivery, the control group collected the rewards in a timely manner, whereas the hM4Di group did not (Fig. 6q, r). These data suggest that chemogenetic downregulation of DMS CINs transiently leads to 1) frequent, excessive magazine entries, 2) slowed reversal learning, and 3) impaired reward collection upon contingency reversal.

Reviewer 3, concern 2

2. The inclusion of IVSA and a 10-d withdrawal period after IVSA seems out of place. While it is certainly nice to see that contingent and non-contingent cocaine exposure produce similar adaptations in firing and inhibitory signaling selection of the timepoint needs justification (see Response #1 below). Do the authors have behavior data in these mice as well? If so, and it is similar to the non-contingent this would significantly increase a causal link. Providing the total mean cocaine intake in a dose manner would be helpful to compare with the amount of cocaine these mice receive versus the non-contingent (see Response #2 below).

Response #1: We thank the reviewer for raising this concern. In Figure 2, we show that intraperitoneal cocaine administration suppressed the activity of DMS CINs 2 d after the last cocaine administration. Additionally, this suppression was also observed 24 d after the last cocaine administration, which is the approximate time point when reversal learning was assessed in live animals (Fig. 1). For our IVSA study, we wanted to choose a time point between 2 d and 24 d after the last cocaine administration. Therefore, we did our recordings 10 d after the last cocaine IVSA session.

The reason we performed the IVSA experiment was to investigate whether contingent or non-contingent cocaine exposure affects the DMS CINs similarly. We added this justification in the result section of Figure 2.

Response #2: We appreciate the reviewer for this suggestion. We do have data for total nose-pokes for cocaine IVSA sessions (Supplementary Fig. 2b). On average, the nose pokes for cocaine intravenous self-administration across sessions were 11.9 (Session 1, S1) + 5.7 (S2) + 5.7 (S3) + 8.8 (S4) + 8(S5) + 15.1 (S6) + 28.1 (S7) ≈ 84 nose pokes. The concentration of infused cocaine was 1 mg/kg/infusion, which corresponds to an average of 84 mg/kg in intravenous cocaine administration across the IVSA session per mouse. For cocaine i.p. administration, we used a concentration of 15 mg/kg for 7 sessions which is equal to 105 mg/kg per mouse.

This shows that the total cocaine administration in the i.p. versus the IVSA administration is not notably different. We have added this quantification in the methods section of Figure 2.

Reviewer 3, concern 3

3. The inclusion of the EtOH data in figure 2 also seems like an afterthought. It is certainly interesting that similar plasticity is in place but without assessment of firing (as well as a behavioral effect that is similar) conclusions that EtOH produce the same adaptations in ChAT-IN can't be made (see Response #1 below). What was the justification for allowing mice to drink for 8 weeks? (see Response #2 below)

Response #1: We thank the reviewer and agree with their critique. Since we do not demonstrate in the paper that alcohol exposure also causes deficits in reversal learning and CIN firing, we removed instances from the paper where we mention that alcohol exposure also causes deficits in reversal learning via reduced cholinergic activity.

Response #2: Mice were administered with ethanol via the intermittent-access to 20% alcohol two-bottle choice drinking procedure (Cheng *et al.*, 2017; Hwa *et al.*, 2011; Warnault *et al.*, 2013). As per Hwa *et al.*, 2011, the intermittent-access protocol can reliably induce alcohol dependence in mice in 4 weeks, and much more potent as compared to continuous access to alcohol (Fig. 2 below). However, we extended this period to 8 weeks for our study to ensure that the mice were subjected to chronic alcohol exposure.

Figure 2. Hwa *et al.*, 2011; “Ethanol intake (g/kg) over 24 hours for male C57BL/6J mice given continuous access (CA, gray circles, n = 12) or intermittent access (IA, black circles, n = 15) to alcohol. Ethanol concentrations were faded from 3, 6, and 10% ethanol to be maintained on 20% ethanol for the remainder of the experiment. Mice receiving alcohol intermittently had significantly higher alcohol consumption during 3-week maintenance phase on 20% ethanol. Data are mean intake \pm SEM. ** $p < 0.01$ difference between groups.”

Reviewer 3, concern 4

4. There are numerous points that need clarifying with regard to the slice physiology.

It would seem entirely unlikely that a K-gluconate-based internal solution would be used for collecting spontaneous and evoked IPSCs. Please clarify which solutions were used for IPSCs and A/N ratio recordings (see Response #1 below).

What was the justification for a 200 ms interval being the only interval chosen? Typically PPR measure use at least 2 or 3 time points to gain a better picture of the release probability (see Response #2 below).

The authors need to provide the mean input-output curve data for the Chrimson oIPSCs (see Response #3 below).

The n-size for the oIPSCs in panel indicate that only 2 cells were collected from 2 mice. This is too low to make any meaningful statistical conclusion (see Response #4 below).

Additional information is needed with regard to how the NMDAR components of the EPSC at +40 was extracted to ensure AMPAR components were also not included in the amplitude (see Response #5 below).

What was the rationale for applying AMPA to the slice for only 15 seconds. It would be helpful if the authors provided a citation where this approach to assess AMPAR-specific signaling was used as it is not a typical approach compared to AMPAR I/O curves or sEPSCs. Less of an issue but there seems to be significant noise in the representative traces (see Response #6 below).

Why was the red-shifted Chrimson opsin used for data collected in Figure 5a-e but the Ai32 line was used in studies related to alcohol administration in Figure 5f? (see Response #7 below)

While data showing that cocaine self-administration augments dMSN-dependent IPSCs is interesting, it does not definitively show that the cocaine potentiation of AMPAR signaling in dMSNs is the driving factor without simulating glutamate release at dMSNs and recording from CINs. Notably, this doesn't detract from the findings, but the authors should consider rewording the results/interpretation (see Response #8 below).

The impact of dMSN stimulation was assessed but not stimulation of iMSNs. This would seem like an important piece of information to have. While iMSNs clearly do not innervate the CINs as much as dMSN they still do provide inhibitory input. Thus knowing whether or not this inhibitory input is also sufficient to inhibit CINs or if it is selective for dMSNs (**see Response #9 below**).

Response #1: K-gluconate-based internal solution was used to measure the spontaneous firing of DMS CINs in cell-attached recordings and also for all current-clamp recordings (action potentials). Cesium methanesulfonate-based internal solution was used for all voltage-clamp recordings (IPSCs, AMPA/NMDA ratio recordings). We updated the methods for electrophysiology experiments with clear specifications of the intracellular solution used for each experiment.

Response #2: We thank the reviewer and agree with their suggestion. We repeated the PPR measurement in Old Figure 2i at multiple inter-stimulation times (Fig. 2j, k). We found that the PPR of electrically evoked IPSCs in CINs was lower in the cocaine group than in saline controls at all inter-stimulation intervals. This **new experiment** was done in mice that were i.p. injected with cocaine.

Response #3: We have now added mean input-output curve data for the Chrimson oIPSCs (Fig. 4j).

Response #4: We have now increased the sample sizes for dMSN→CIN oIPSC (n = 13/5) and iMSN→CIN oIPSC (n = 16/5) for Figure 4c.

Response #5: We thank the reviewer for this concern. For the measurement of AMPAR/NMDAR ratio, the peak currents of AMPAR-mediated EPSCs were measured at a holding potential of -70 mV, and the NMDAR-mediated EPSCs were estimated as the EPSCs at $+40$ mV, 30 ms after the peak AMPAR EPSCs, when the contribution of the AMPAR component was minimal (Ma *et al.*, 2018). The AMPA/NMDA ratio was calculated by dividing the AMPAR EPSCs by NMDAR EPSCs. We added this explanation in the methods section of the new manuscript.

Response #6: We observed that bath application of AMPA for 15 sec elicited approximately 80-120 pA current in multiple postsynaptic neurons that we tested at the beginning of the experiment. This allowed us to avoid the ceiling and floor effect, as this current range is neither high nor low. Therefore, we chose a bath application duration of 15 seconds. The reviewer is correct that current induced by AMPA bath application is not a very typical approach to assess AMPAR-specific signaling, as compared to AMPAR I/O curves or sEPSCs. However, it has been used to demonstrate that *in vivo* cocaine exposure induces long-term potentiation in dopamine neurons (Ungless *et al.*, 2001). It has also been used in other addiction-related studies (Borgland *et al.*, 2004; Dong *et al.*, 2004; Wang *et al.*, 2015). In Figure 5c, we provide improved representative traces for the AMPA-induced current.

Response #7: We appreciate the reviewer for their critique. We initially performed the experiment in Figure 5l, i.e., to measure dMSN→CIN GABAergic transmission using D1-Cre;Ai32 mice. However, in these mice, visualizing CINs for patch-clamp electrophysiology was challenging because CINs were not fluorescently labeled. We then bred D1-Cre;Ai32;ChATeGFP mice. However, the ChR2⁺ fibers intensely contained GFP, which made the visualization of CINs (also containing GFP) challenging. Henceforth, we had to rely on cell size and biophysical properties (spontaneous firing, sag) of the patched neuron to identify CINs.

To improve productivity, we bred D1-Cre;Ai167;ChATeGFP mice, which expressed red-shifted Chrimson-tdTomato in dMSNs. In these mice, we were able to identify CINs (GFP⁺) in slices because the Chrimson-expressing fibers were red and CINs were green. These mice were used in recordings in Figure 5a-h.

Response #8: We thank the reviewer for this input. We have removed instances from the manuscript where we suggested that the driving factor of reduced CIN activity and behavior flexibility is because of the cocaine potentiation of AMPAR signaling in dMSNs. Instead, we write “these findings indicate that reinforcement of

addictive substance use potentiates dMSN activity and strengthens GABAergic dMSN→CIN transmission, which reduces CIN activity.”

Response #9: We performed **two new experiments** to assess the effect of iMSN stimulation on CIN firing. We found that in the triple transgenic animal, A2A-Cre;Ai32;ChATeGFP, light stimulation was able to inhibit the CINs. However, this iMSN-mediated inhibition was weaker as compared to that observed after dMSN stimulation (Fig. 4d-g, Supplementary Fig. 4c). Additionally, in A2A-Cre;ChATeGFP mice infused with AAV-flex-Chrimson-tdT in the DMS, we found that light stimulation was not able to inhibit CIN activity, while dMSN stimulation was sufficient to inhibit CIN firing (Supplementary Fig. 4d-e).

Reviewer 3, concern 5

5. No details regarding the approach used to quantify inputs to CIN are given. How these data were quantitatively collected is important.

Response: We thank the reviewer for the critical advice. Here is the detailed approach that we used to quantify the inputs to DMS CINs, and to classify them as being either dMSNs or iMSNs:

- 1) Genotype: ChAT-Cre;D1tdT (n = 4 mice)
- 2) Bilateral DMS infusion of
 - a) Helper viruses: AAV8-FLEX-TVA-mCherry and AAV8-FLEX-RG
 - b) Rabies-GFP
- 3) ChAT staining (Cyan)

This results in the color coding of the different cell types in the DMS as follows:

- 1) Presynaptic dMSNs: Red (D1tdT⁺) Green (GFP⁺)
- 2) Presynaptic iMSNs: Green (GFP⁺)
- 3) Presynaptic CINs: Green (GFP⁺) + Cyan (ChAT staining)
- 4) Starter CINs: Red (mCherry⁺) + Green (GFP⁺) + Cyan (ChAT staining)

The sections were imaged using confocal microscopy at 10x optical resolution. Images were analyzed using Imaris software. All counting was conducted manually. We counted the total presynaptic dMSNs, presynaptic iMSNs, and presynaptic CINs in 4-5 dorsomedial striatum coronal sections in both the left and right hemispheres and calculated their respective percentages for better analysis. This gave us 8-10 data points (respective percentages) per mouse. We performed this analysis on all 4 mice. Eventually, this gave us 35 data points from the 4 mice, which we plotted in Old Figure 3i. However, as suggested by Reviewer 1 (please see our Response to Reviewer 1, Concern 2), we now conducted the statistical test on the cumulative data obtained from the 4 mice (4 data points) and not on the data from the 35 sections. We have added this detail in the methods section.

Reviewer 3, concern 6

6. No information is provided about the background (or anything for that matter) of the mice and rats. Are they all male or a mix of male and female? What was the breeding strategy for the double transgenics and where were they obtained from? What age were they? Etc.

Response: We thank the reviewer for their constructive comment. Transgenic mice were of the C57BL/6 (Mutant Mouse Regional Resource Centers) background, whereas transgenic rats were of the Long-Evans (Harlan Laboratories) background. All electrophysiology and histology experiments were conducted on a mix of 4- to 7-month-old male and female mice. Behavioral experiments were conducted in 4- to 7- month-old male and female mice/rats. Both mice (4-5 mice per cage) and rats (2 rats per cage) were group-housed and were kept in a temperature- and humidity-controlled environment. For the behavioral experiments, animals were kept in a 12:12 h light: dark cycle, and the experiments were performed in the dark cycle. We provide a list of all single transgenic mice and rats in the supplementary section (copied here), which were crossed as desired to obtain the

genotypes used in this study. We have updated in detail the information regarding animals in our methods and supplementary sections.

Table 7. Vendors of the transgenic mouse/rat lines

ID	Genotype (Mice)		Vendor	Lot #
1	C57BL/6J	Mice	Jackson Laboratories	00664
2	ChAT-Cre(+/+)	Mice	Jackson Laboratories	028861
3	Δ -neo ChAT-Cre(+/+)	Mice	Jackson Laboratories	031661
4	ChATeGFP(+/+)	Mice	Jackson Laboratories	007902
5	Ai14(+/+)	Mice	Jackson Laboratories	007914
6	D1tdTomato(+/-)	Mice	Jackson Laboratories	016204
7	Ai32(+/+)	Mice	Jackson Laboratories	012569
8	Drd1-Cre(+/-)	Mice	MMRRC	29178
9	Ai167(+/-)	Mice	Allen Brain Institution	N/A
10	A2A-Cre(+/-)	Mice	MMRRC	036158-UCD
1	D1-Cre(+/-)	Rats	RRRC	856
2	ChAT-Cre(+/-)	Rats	RRRC	658
3	tdTomato/Hom-KI(+/+)	Rats	Envigo Horizon	TGRL9660

Reviewer 3, concern 7

7. *There are no controls for in vivo optogenetic studies show in figure 5.*

Response: All D1-Cre rats (n = 12) were infused with AAV-flex-Chrimson-tdT and implanted with optical fibers. However, during instrumental training, only half of them were provided with 2 sec of continuous light (590 nm) stimulation after an active lever press. The other half never received any optical stimulation but were introduced to the chamber and were free to press. These rats, which did not receive light stimulation on an active lever press, became the controls. In Figure 5n, we show the average number of active lever presses for the two groups across training sessions.

Reviewer 3, concern 8

8. *It isn't clear what virus was used as a control for chemogenetic inhibition studies.*

Response: AAV-EF1a-DIO-mCherry was used as a control for AAV-EF1a-DIO-hM4Di-mCherry in the chemogenetic study in Figure 6. We have now specified the control virus in Figure 6a.

Reviewer 3, concern 9

9. *Figure 6a shows representative spike firing but it isn't clear what that is? Is it ex vivo recordings?*

Response: Yes, Figure 6b represents *ex vivo* whole-cell current-clamp recordings of spontaneous firing in a DMS CIN that expressed AAV-EF1a-DIO-hM4Di-mCherry in ChAT-Cre mice before and after bath application of CNO. The genotype of animals used is the same as in the schematic of Figure 6a, i.e., infusion of AAV5-EF1a-DIO-hM4Di-mCherry in ChAT-Cre mice. In Figure 6b, we have specified that the representative spike firing traces are CIN *ex vivo* recordings.

Reviewer 3, concern 10

10. Please clarify what was used for control in chemogenetic inhibition studies.

Response: The control group consisted of ChAT-Cre mice infused with AAV5-EF1a-DIO-mCherry in the DMS. We added the name of the control virus in the schematic of Figure 6a.

Reviewer 3, concern 11

11. What control virus or parameters were used for “controls” in the optogenetic inhibition studies.

Response: The virus used for controls in the optogenetic inhibition study was AAV-DIO-eYFP in ChAT-Cre; tdTomato rats. The following changes were made in Figure 7:

- 1) We specified the control virus in the schematic of Figure 7a.
- 2) We added a representative spiking trace in Figure 7c, showing that yellow (590 nm) light stimulation did not affect DMS CIN firing in the control (eNpHR-) group.

Reviewer 3, minor concern 1

1. While I personally don't think the inclusion of the EtOH data is helpful, if the authors feel the need to keep it in, the manuscript would benefit from the reorganization by grouping EtOH-related studies into a separate figure/results section rather than randomly inserted among cocaine studies in its current format.

Response: We appreciate and agree with the reviewer's suggestion. We grouped the ethanol-related studies and arranged them in Figure 5j-l.

Other changes:

1. All changes in the text use a blue font

Reference list

Aceves Buendia, J.J., Tiroshi, L., Chiu, W.H., and Goldberg, J.A. (2019). Selective remodeling of glutamatergic transmission to striatal cholinergic interneurons after dopamine depletion. *Eur J Neurosci* 49, 824-833. 10.1111/ejn.13715.

Assous, M., Kaminer, J., Shah, F., Garg, A., Koos, T., and Tepper, J.M. (2017). Differential processing of thalamic information via distinct striatal interneuron circuits. *Nature communications* 8, 15860. 10.1038/ncomms15860.

Bobadilla, A.C., Dereschewitz, E., Vaccaro, L., Heinsbroek, J.A., Scofield, M.D., and Kalivas, P.W. (2020). Cocaine and sucrose rewards recruit different seeking ensembles in the nucleus accumbens core. *Molecular psychiatry* 25, 3150-3163. 10.1038/s41380-020-00888-z.

Bolam, J.P., Ingham, C.A., Izzo, P.N., Levey, A.I., Rye, D.B., Smith, A.D., and Wainer, B.H. (1986). Substance P-containing terminals in synaptic contact with cholinergic neurons in the neostriatum and basal forebrain: a double immunocytochemical study in the rat. *Brain Res* 397, 279-289. 10.1016/0006-8993(86)90629-3.

- Borgland, S.L., Malenka, R.C., and Bonci, A. (2004). Acute and chronic cocaine-induced potentiation of synaptic strength in the ventral tegmental area: electrophysiological and behavioral correlates in individual rats. *J Neurosci* *24*, 7482-7490.
- Bradfield, L.A., and Balleine, B.W. (2017). Thalamic control of dorsomedial striatum regulates internal state to guide goal-directed action selection. *J Neurosci* *37*, 3721-3733. 10.1523/JNEUROSCI.3860-16.2017.
- Bradfield, L.A., Bertran-Gonzalez, J., Chieng, B., and Balleine, B.W. (2013). The thalamostriatal pathway and cholinergic control of goal-directed action: interlacing new with existing learning in the striatum. *Neuron* *79*, 153-166. 10.1016/j.neuron.2013.04.039.
- Brown, M.T., Tan, K.R., O'Connor, E.C., Nikonenko, I., Muller, D., and Luscher, C. (2012). Ventral tegmental area GABA projections pause accumbal cholinergic interneurons to enhance associative learning. *Nature* *492*, 452-456. 10.1038/nature11657.
- Cachope, R., Mateo, Y., Mathur, B.N., Irving, J., Wang, H.L., Morales, M., Lovinger, D.M., and Cheer, J.F. (2012). Selective activation of cholinergic interneurons enhances accumbal phasic dopamine release: setting the tone for reward processing. *Cell reports* *2*, 33-41. 10.1016/j.celrep.2012.05.011.
- Chen, B.T., Bowers, M.S., Martin, M., Hopf, F.W., Guillory, A.M., Carelli, R.M., Chou, J.K., and Bonci, A. (2008). Cocaine but not natural reward self-administration nor passive cocaine infusion produces persistent LTP in the VTA. *Neuron* *59*, 288-297. 10.1016/j.neuron.2008.05.024.
- Chen, W., Wang, Y., Sun, A., Zhou, L., Xu, W., Zhu, H., Zhuang, D., Lai, M., Zhang, F., Zhou, W., and Liu, H. (2016). Activation of AMPA receptor in the infralimbic cortex facilitates extinction and attenuates the heroin-seeking behavior in rats. *Neurosci Lett* *612*, 126-131. 10.1016/j.neulet.2015.11.024.
- Cheng, Y., Huang, C.C.Y., Ma, T., Wei, X., Wang, X., Lu, J., and Wang, J. (2017). Distinct synaptic strengthening of the striatal direct and indirect pathways drives alcohol consumption. *Biological psychiatry* *81*, 918-929. 10.1016/j.biopsych.2016.05.016.
- Chuhma, N., Mingote, S., Moore, H., and Rayport, S. (2014). Dopamine neurons control striatal cholinergic neurons via regionally heterogeneous dopamine and glutamate signaling. *Neuron* *81*, 901-912. 10.1016/j.neuron.2013.12.027.
- Ding, J.B., Guzman, J.N., Peterson, J.D., Goldberg, J.A., and Surmeier, D.J. (2010). Thalamic gating of corticostriatal signaling by cholinergic interneurons. *Neuron* *67*, 294-307. S0896-6273(10)00475-7 [pii] 10.1016/j.neuron.2010.06.017.
- Doig, N.M., Magill, P.J., Apicella, P., Bolam, J.P., and Sharott, A. (2014). Cortical and thalamic excitation mediate the multiphasic responses of striatal cholinergic interneurons to motivationally salient stimuli. *J Neurosci* *34*, 3101-3117. 10.1523/JNEUROSCI.4627-13.2014.
- Dong, Y., Saal, D., Thomas, M., Faust, R., Bonci, A., Robinson, T., and Malenka, R.C. (2004). Cocaine-induced potentiation of synaptic strength in dopamine neurons: behavioral correlates in GluRA(-/-) mice. *Proc Natl Acad Sci U S A* *101*, 14282-14287. 10.1073/pnas.0401553101.
- Francis, T.C., Yano, H., Demarest, T.G., Shen, H., and Bonci, A. (2019). High-Frequency Activation of Nucleus Accumbens D1-MSNs Drives Excitatory Potentiation on D2-MSNs. *Neuron* *103*, 432-444.e433. 10.1016/j.neuron.2019.05.031.
- Gonzales, K.K., Pare, J.F., Wichmann, T., and Smith, Y. (2013). GABAergic inputs from direct and indirect striatal projection neurons onto cholinergic interneurons in the primate putamen. *J Comp Neurol* *521*, 2502-2522. 10.1002/cne.23295.

- Hwa, L.S., Chu, A., Levinson, S.A., Kayyali, T.M., DeBold, J.F., and Miczek, K.A. (2011). Persistent escalation of alcohol drinking in C57BL/6J mice with intermittent access to 20% ethanol. *Alcohol Clin Exp Res* 35, 1938-1947. 10.1111/j.1530-0277.2011.01545.x.
- Kosillo, P., Zhang, Y.F., Threlfell, S., and Cragg, S.J. (2016). Cortical Control of Striatal Dopamine Transmission via Striatal Cholinergic Interneurons. *Cereb Cortex*. 10.1093/cercor/bhw252.
- Lee, J.H., Ribeiro, E.A., Kim, J., Ko, B., Kronman, H., Jeong, Y.H., Kim, J.K., Janak, P.H., Nestler, E.J., Koo, J.W., and Kim, J.H. (2020). Dopaminergic Regulation of Nucleus Accumbens Cholinergic Interneurons Demarcates Susceptibility to Cocaine Addiction. *Biological psychiatry* 88, 746-757. 10.1016/j.biopsych.2020.05.003.
- Lhost, J., More, S., Watabe, I., Louber, D., Ouagazzal, A.M., Liberge, M., and Amalric, M. (2021). Interplay Between Inhibitory Control and Behavioural Flexibility: Impact of Dorsomedial Striatal Dopamine Denervation in Mice. *Neuroscience* 477, 25-39. 10.1016/j.neuroscience.2021.09.026.
- Liu, C., Cai, X., Ritzau-Jost, A., Kramer, P.F., Li, Y., Khaliq, Z.M., Hallermann, S., and Kaeser, P.S. (2022). An action potential initiation mechanism in distal axons for the control of dopamine release. *Science* 375, 1378-1385. 10.1126/science.abn0532.
- Lozovaya, N., Eftekhari, S., Cloarec, R., Gouty-Colomer, L.A., Dufour, A., Riffault, B., Billon-Grand, M., Pons-Bennaceur, A., Oumar, N., Burnashev, N., et al. (2018). GABAergic inhibition in dual-transmission cholinergic and GABAergic striatal interneurons is abolished in Parkinson disease. *Nature communications* 9, 1422. 10.1038/s41467-018-03802-y.
- Ma, T., Cheng, Y., Roltsch Hellard, E., Wang, X., Lu, J., Gao, X., Huang, C.C.Y., Wei, X., Ji, J., and Wang, J. (2018). Bidirectional and long-lasting control of alcohol-seeking behavior by corticostriatal LTP and LTD. *Nat Neurosci* 21, 373-383. 10.1038/s41593-018-0081-9.
- Ma, T., Huang, Z., Xie, X., Cheng, Y., Zhuang, X., Childs, M.J., Gangal, H., **Wang, X.**, Smith, L.N., Smith, R.J., et al. (2022). Chronic alcohol drinking persistently suppresses thalamostriatal excitation of cholinergic neurons to impair cognitive flexibility. *The Journal of clinical investigation* 132, e154969. 10.1172/JCI154969.
- Martone, M.E., Armstrong, D.M., Young, S.J., and Groves, P.M. (1992). Ultrastructural examination of enkephalin and substance P input to cholinergic neurons within the rat neostriatum. *Brain Res* 594, 253-262.
- Matamales, M., McGovern, A.E., Mi, J.D., Mazzone, S.B., Balleine, B.W., and Bertran-Gonzalez, J. (2020). Local D2- to D1-neuron transmodulation updates goal-directed learning in the striatum. *Science* 367, 549-555. 10.1126/science.aaz5751.
- Matamales, M., Skrbis, Z., Hatch, R.J., Balleine, B.W., Gotz, J., and Bertran-Gonzalez, J. (2016). Aging-related dysfunction of striatal cholinergic interneurons produces conflict in action selection. *Neuron* 90, 362-373. 10.1016/j.neuron.2016.03.006.
- Partridge, J.G., Apparsundaram, S., Gerhardt, G.A., Ronesi, J., and Lovinger, D.M. (2002). Nicotinic acetylcholine receptors interact with dopamine in induction of striatal long-term depression. *J Neurosci* 22, 2541-2549. 20026219.
- Sciamanna, G., Tassone, A., Mandolesi, G., Puglisi, F., Ponterio, G., Martella, G., Madeo, G., Bernardi, G., Standaert, D.G., Bonsi, P., and Pisani, A. (2012). Cholinergic dysfunction alters synaptic integration between thalamostriatal and corticostriatal inputs in DYT1 dystonia. *J Neurosci* 32, 11991-12004. 10.1523/JNEUROSCI.0041-12.2012.
- Shalev, U., Morales, M., Hope, B., Yap, J., and Shaham, Y. (2001). Time-dependent changes in extinction behavior and stress-induced reinstatement of drug seeking following withdrawal from heroin in rats. *Psychopharmacology (Berl)* 156, 98-107. 10.1007/s002130100748.

- Son, J.H., Kuhn, J., and Keefe, K.A. (2013). Perseverative behavior in rats with methamphetamine-induced neurotoxicity. *Neuropharmacology* 67, 95-103. 10.1016/j.neuropharm.2012.09.021.
- Stalnaker, T.A., Berg, B., Aujla, N., and Schoenbaum, G. (2016). Cholinergic Interneurons Use Orbitofrontal Input to Track Beliefs about Current State. *J Neurosci* 36, 6242-6257. 10.1523/jneurosci.0157-16.2016.
- Straub, C., Tritsch, N.X., Hagan, N.A., Gu, C., and Sabatini, B.L. (2014). Multiphasic modulation of cholinergic interneurons by nigrostriatal afferents. *J Neurosci* 34, 8557-8569. 10.1523/JNEUROSCI.0589-14.2014.
- Sullivan, M.A., Chen, H., and Morikawa, H. (2008). Recurrent inhibitory network among striatal cholinergic interneurons. *J Neurosci* 28, 8682-8690. 10.1523/JNEUROSCI.2411-08.2008.
- Thorn, C.A., and Graybiel, A.M. (2010). Pausing to regroup: thalamic gating of cortico-basal ganglia networks. *Neuron* 67, 175-178. 10.1016/j.neuron.2010.07.010.
- Tritsch, N.X., and Sabatini, B.L. (2012). Dopaminergic modulation of synaptic transmission in cortex and striatum. *Neuron* 76, 33-50. 10.1016/j.neuron.2012.09.023 S0896-6273(12)00858-6 [pii].
- Ungless, M.A., Whistler, J.L., Malenka, R.C., and Bonci, A. (2001). Single cocaine exposure in vivo induces long-term potentiation in dopamine neurons. *Nature* 411, 583-587. 10.1038/35079077.
- Wang, J., Cheng, Y., Wang, X., Roltsch Hellard, E., Ma, T., Gil, H., Ben Hamida, S., and Ron, D. (2015). Alcohol elicits functional and structural plasticity selectively in dopamine D1 receptor-expressing neurons of the dorsomedial striatum. *J Neurosci* 35, 11634-11643. 10.1523/JNEUROSCI.0003-15.2015.
- Warnault, V., Darcq, E., Levine, A., Barak, S., and Ron, D. (2013). Chromatin remodeling--a novel strategy to control excessive alcohol drinking. *Transl Psychiatry* 3, e231. 10.1038/tp.2013.4.
- Wieland, S., Du, D., Oswald, M.J., Parlato, R., Kohr, G., and Kelsch, W. (2014). Phasic dopaminergic activity exerts fast control of cholinergic interneuron firing via sequential NMDA, D2, and D1 receptor activation. *J Neurosci* 34, 11549-11559. 10.1523/JNEUROSCI.1175-14.2014.

REVIEWER COMMENTS

Reviewer #1 (Remarks to the Author):

The authors have addressed my concerns.

Reviewer #2 (Remarks to the Author):

The authors have been very responsive to the comments of the reviewers and have added considerable new data. It is welcome to see better controlled versions of the previous studies, or at least less confounded by spatial changes with reversal. There is, therefore, much to like about the attempt but there are still considerable issues to be addressed:

1. Replacing the previous work with the current studies is appropriate; indeed, I think they could go further and simply remove the previous studies altogether given the difficulty with interpretation. The authors argument in the letter that: "These results suggest that cocaine exposure disrupted reversal learning in mice and that the spatial configuration in Figure 6 (and Old Fig. 1) unlikely contributes to a cocaine-induced reversal learning deficits." cannot be ascertained from the results without a direct within experiment comparison. Perhaps those earlier studies would be better in the supplement with a supplementary discussion to address these issues there.
2. There are still problems with the introduction, specifically the rationale for the current experiments. The rationale as it stands is not at all clear. Why should cocaine be even hypothesized to affect reversal learning by modifying activity at cholinergic interneurons when it has been established previously that natural rewards do not have this effect; indeed, that they rely on CINs for reversal? Thus, the statement: "Given that positive reinforcers enhance striatal dMSN activity, we hypothesize that inhibitory dMSN CIN transmission plays an essential role in mediating drug-induced suppression of cognitive flexibility" really make much sense as it stands. Something else must be induced by cocaine exposure to produce the effects the authors observe but there is no suggestion either in the introduction - to motivate the investigation - or in the discussion - to interpret the authors' findings. Arguing that dSPNs always inhibit CINs suggests that animal could never learn reversals. Why do drugs increase dSPN inhibition but other rewards, like sucrose, do not? There is some mechanism missing here that differentiates drug reward from other rewards.
3. It is not clear to me what the authors mean by this statement at the end of the introduction: "Importantly, SNr-projecting dMSNs in the direct pathway that mediate reinforcement also sent collaterals to inhibit CINs. Our data demonstrate that the inhibitory dMSN CIN connection mediates the reinforcement-induced reduction in cognitive flexibility." What do the authors mean by "reinforcement" here? Are they contrasting these effects with reward? Does cocaine induce reinforcement - some form of enhanced effect on dopamine neurons resulting in compulsive performance during reversal compared to sucrose that does not? It is simply not clear and needs to be given the prominence of this statement.
4. The behavioral data are incorrectly analysed in a couple of places. The authors use independent t-tests to test devaluation effects in vehicle and cocaine groups in experiments in which they are comparing the size of those devaluation effects between the groups. The appropriate analysis is to use a mixed model ANOVA and include both groups in an F-test so as to test for a significant group x devaluation interaction. Without a reliable interaction the data cannot be interpreted.
Dior example: on P.6-7 they state: "However, the saline, but not the cocaine group, was sensitive to outcome devaluation after reversal training (saline: Fig. 1k left, $t(21) = 2.37$, $p < 0.05$; cocaine: Fig. 1k right, $t(23) = -0.750$, $p > 0.05$)." This problem is found in the analysis of the data in Figure 1 panels f, h, j, k; and Figure 7 panels e, and h.
Discussion: The authors emphasize the effects of stimulant exposure on reversal; they find no effect of drug exposure on initial learning. However, this actually is not the common result. There have been assessments of drug exposure- and drug context exposure-induced effects on devaluation previously. For example: Nelson & Killcross (2006) *J. Neurosci*; Nelson & Killcross (2013) *Frontiers in Neuroscience*; Corbit et al (2014), *Neuropsychopharmacology*; Halbout et al, (2016) *Frontiers in Psychiatry*; Leong et al (2016) *Physiol Behav*; Furlong et al (2017) *addiction biol*; Jones et al, (2022) *learning and memory*. These reported effects are not effects on reversal but on devaluation conducted after initial training. They are not discussed or referenced here. What is the difference between the authors treatments and those previously reported? Is it a matter of dose or intensity of the effects of drug exposure When does drug exposure affect initial

learning and when does it only affect reversal?

Reviewer #3 (Remarks to the Author):

The authors have done a very nice job of addressing almost all of my concerns, including a significant number of additional experiments that have added to an already interesting and impactful body of work.

Below is the only relatively minor issue.

It is appreciated that the authors provided references to previous manuscripts that have used this behavioral paradigm. It is also appreciated that cognitive flexibility is indeed loosely defined as the ability of an animal to adjust to environmental changes. However, it is worth noting that the provided references almost exclusively refer to this task as changes in action-outcome or goal-directed learning and not specifically cognitive flexibility (with the exception of Matamales et al., 2016). This is also an important point when citing this work. For example saying CINs regulate cognitive flexibility and then citing those references.

While this task is not generally used to measure changes in cognitive flexibility as it is often referred to in the literature versus something like an attentional set-shifting task, I do feel like the behavior construct "loosly" falls within a cognitive construct. I think it would be helpful to be more specific about the type of cognitive construct being tested at the very least in the title of the manuscript and then make an inference as to what that aligns with (i.e. cognitive flexibility) in the main text. For example, it would seem to more appropriate for the title to read "Drug Reinforcement impairs flexibility of goal-directed actions (or learning) by inhibiting striatal cholinergic interneurons in mice. Additionally, the authors can note that they will refer to alterations in this learning as cognitive flexibility. The distinction is small but important in the eyes of this reviewer.

We thank the reviewers for their insightful comments and appreciate the opportunity to revise the manuscript. We have carefully revised the manuscript and provided our detailed responses below. The original concerns expressed by the reviewers are shown in *italics*.

Reviewer 1

The authors have addressed my concerns.

Response: We thank the reviewer for their positive feedback.

Reviewer 2

Concern 1

The authors have been very responsive to the comments of the reviewers and have added considerable new data. It is welcome to see better controlled versions of the previous studies, or at least less confounded by spatial changes with reversal. There is, therefore, much to like about the attempt but there are still considerable issues to be addressed:

*1. Replacing the previous work with the current studies is appropriate; indeed, I think they could go further and simply remove the previous studies altogether given the difficulty with interpretation. (see **Response#1 below**) The authors argument in the letter that: “These results suggest that cocaine exposure disrupted reversal learning in mice and that the spatial configuration in Figure 6 (and Old Fig. 1) unlikely contributes to a cocaine-induced reversal learning deficits.” cannot be ascertained from the results without a direct within experiment comparison (see **Response#2 below**). Perhaps those earlier studies would be better in the supplement with a supplementary discussion to address these issues there (see **Response#1 below**).*

Response #1: We appreciate the reviewer’s suggestion to replace previous studies, which had spatial configuration issues, with the current studies that have addressed this problem in our manuscript. In response to this feedback, we relocated the chemogenetics result (old Fig. 6), which still exhibited the spatial issue, to the supplementary material. We also added a supplementary discussion to discuss this result and its limitations in detail. This approach enables us to maintain a clear focus on the studies without the spatial configuration issue in the main text, while providing interested readers access to the earlier work and an in-depth discussion of its limitations. By doing so, we aim to strengthen the overall clarity of the manuscript. We thank the reviewer for their valuable input that has improved our work.

Response #2: We concur that the statement in our first rebuttal letter, “These results suggest that cocaine exposure disrupted reversal learning in mice and that the spatial configuration in Figure 6 (and Old Fig. 1) unlikely contributes to a cocaine-induced reversal learning deficits.”, cannot be ascertained from the results without a direct within-experiment comparison. We acknowledge that conducting such an experiment is beyond the scope of our current study.

Reviewer 2, concern 2

*2. There are still problems with the introduction, specifically the rationale for the current experiments. The rationale as it stands is not at all clear. Why should cocaine be even hypothesized to affect reversal learning by modifying activity at cholinergic interneurons (see **Response#1 below**) when it has been established previously that natural rewards do not have this effect; indeed, that they rely on CINs for reversal? Thus, the statement: “Given that positive reinforcers enhance striatal dMSN activity, we hypothesize that inhibitory dMSN CIN transmission plays an essential role in mediating drug-induced suppression of cognitive flexibility”*

really make much sense as it stands. **(see Response#2 below)** Something else must be induced by cocaine exposure to produce the effects the authors observe but there is no suggestion either in the introduction - to motivate the investigation - or in the discussion - to interpret the authors' findings. **(see Response#3 below)** Arguing that dSPNs always inhibit CINs suggests that animal could never learn reversals. **(see Response#4 below)** Why do drugs increase dSPN inhibition but other rewards, like sucrose, do not? There is some mechanism missing here that differentiates drug reward from other rewards. **(see Response#3 below)**

Response #1: We acknowledge the reviewer's concern regarding the rationale for our current experiments and would like to further clarify our hypothesis that cocaine may affect reversal learning through alterations in cholinergic interneuron (CIN) activity. Firstly, previous studies have demonstrated that cocaine exposure impairs reversal learning (Jentsch *et al.*, 2002; West *et al.*, 2021b). Secondly, it has been well established that CINs play a crucial role in mediating reversal learning (Bradfield *et al.*, 2013; Matamales *et al.*, 2016). Thirdly, a recent study has shown an association between inflexible compulsive cocaine self-administration and reduced CIN activity (Lee *et al.*, 2020), indicating that cocaine-induced CIN dysfunction may contribute to reversal learning deficits. Considering these findings, we hypothesize that cocaine impairs reversal learning by reducing CIN activity. We have updated our introduction to incorporate this statement and provide a clearer rationale for our experiments.

Response #2: We appreciate the reviewer's critique. We observed that cocaine reduced DMS CIN activity and increased presynaptic GABAergic inputs onto DMS CINs. Consequently, we found a strong local dMSN to CIN inhibition in the striatum. Since drug-exposure is known to upregulate dMSN activity, the dMSN→CIN inhibitory transmission became a candidate mechanism to explain the observed reduction in CIN activity after drug exposure. Interestingly, while self-administration of natural rewards can also upregulate dMSN activity, the plasticity generated after natural reward administration has been shown, both in previous studies and in our research, to be weaker or short lasting compared to drug-induced changes (Chen *et al.*, 2008; Shan *et al.*, 2014)(Fig. 5f-h). This could explain why drug-exposure has a more potent effect on CIN activity and, consequently, on cognitive flexibility. Nevertheless, we propose that this mechanism may apply to all types of reinforcers, with the severity of its impact being dependent on the strength and potency of the given reinforcer to affect dMSN activity. Intriguingly, there is evidence suggesting that excessive or chronic reinforcement of natural rewards can also cause a decline in decision making and flexibility (Chawla *et al.*, 2017; Perez Diaz *et al.*, 2019; Thiebaud *et al.*, 2014). The weaker effect of natural rewards on cognitive flexibility could potentially be due to a transient or weaker change in dMSN→CIN inhibitory transmission. We have incorporated some of these statements into the introduction.

Response #3: We thank the reviewer for their question. Our results showed that drug self-administration led to greater inhibitory dMSN→CIN transmission than sucrose self-administration, and this difference was observed 3 weeks after the last drug exposure (Fig. 5f-h). This increased inhibition may be attributed to the selective facilitation of NMDA receptor and extracellular signal-regulated kinase (ERK) activity in dMSNs by drugs. Cocaine and nicotine exposure has been demonstrated to facilitate NMDAR function in the dorsal striatum (Ortinski *et al.*, 2013; Valjent *et al.*, 2000; Valjent *et al.*, 2005; Wang *et al.*, 2018; Xia *et al.*, 2017; Yamamoto and Zahner, 2012). Additionally, it has been reported that *ex vivo* (i.e., in slices) or *in vivo* exposure to, and withdrawal from, alcohol caused long-term facilitation of NMDAR activity in dMSNs (Kash *et al.*, 2009; Lovinger, 2010; Wang *et al.*, 2007; Wang *et al.*, 2010), and NMDARs are required for corticostriatal LTP induction in the dorsal striatum (Lovinger, 2010; Lovinger and Kash, 2015; McCool, 2011) and for operant alcohol self-administration (Wang *et al.*, 2010). In contrast, sucrose self-administration did not alter NMDAR activity (Fig. 1 below)(Ma *et al.*, 2018). Furthermore, optogenetic corticostriatal LTP induction is successfully achieved in DMS slices from alcohol- but not sucrose-drinking rats. This finding is consistent with previous studies showing that repeated exposure of rats to alcohol facilitates LTP induction (Wang *et al.*, 2012; Wills *et al.*, 2012), and with another study showing that repeated cocaine exposure *in vivo* also facilitates LTP induction (Liu *et al.*, 2005). The selective enhancement of NMDA receptor activity by drugs, as opposed to natural rewards, may explain the persistent strengthening of dMSN→CIN transmission observed following drug exposure.

Figure 1. Operant alcohol self-administration caused higher NMDAR activity than did operant sucrose self-administration. Comparison of input-output relation for NMDAR-EPSCs between alcohol and sucrose groups. $F_{(1, 94)} = 9.62$, $^{##}P = 0.0049$; $n = 13$ neurons from 4 rats (Sucrose) and 13 neurons from 5 rats (EtOH). Adapted from (Ma et al., 2018).

Cocaine exposure may also impact cognitive flexibility through mechanisms beyond those explored in our study. In particular, the prefrontal cortex has been shown to play a key role in mediating executive function and impulsivity, both of which are heavily impacted by cocaine use (Leyrer-Jackson et al., 2021; McCracken and Grace, 2013; Melugin et al., 2021; Smith and Laiks, 2018). Selective activation of the prelimbic cortex-nucleus accumbens core pathway was able to restore cognitive flexibility deficits induced by cocaine (West et al., 2021a). Additionally, dysfunction of the orbitofrontal cortex (OFC) and amygdala has also been linked to cocaine-induced inflexibility (Li et al., 2022; McCracken and Grace, 2013; Stalnaker et al., 2007; Stalnaker et al., 2009).

Importantly, OFC→DMS projections are required for multiple learning and memory phases that are necessary for sustaining response flexibility, which is disrupted after cocaine exposure (Li et al., 2022). Cocaine has also been shown to cause the DMS to encode trial-specific information and rules more strongly, which can interfere with decision-making and reversal learning (Mueller et al., 2021). Interestingly, CINs are crucial modulators of striatal function (Zucca et al., 2018). CINs temporarily inhibit cortico-striatal synapses by binding to the M4 muscarinic receptors (Shen et al., 2015) and can preferentially facilitate cortical activation of D2-MSNs via postsynaptic M1 muscarinic receptors (Oldenburg and Ding, 2011). Drug exposure has also been shown to impair the inhibitory control of striatal CINs on MSN activity (Fleming et al., 2022; Ma et al., 2022).

The primary focus of our study is to identify a pathway mediating drug-induced cognitive inflexibility. Although we briefly consider physiological changes after natural reward-induced reinforcement, the distinction between drug- and natural reward-induced changes is not the main focus of our manuscript. We have updated our Discussion section to include parts of the above response.

Response #4: We appreciate the reviewer’s feedback. Firstly, previous research has shown that glutamatergic projections from the thalamus→CINs mediate reversal learning (Bradfield et al., 2013; Ma et al., 2022; Matamales et al., 2016; Saund et al., 2017). In our study, we provide evidence supporting the idea that inhibitory dMSN→CIN transmission plays a modulatory role in regulating reversal learning. This inhibition is upregulated after drug exposure, causing enhanced suppression of CIN activity, and thereby reducing cognitive flexibility. Secondly, dMSNs are not continuously excited *in vivo*, rather dMSN activity increases during action initiation (Cui et al., 2013). Therefore, inhibition of CINs via dMSNs is not constantly maintained. However, long-lasting dMSN excitation (via drug exposure/excessive reinforcing behaviors) could lead to a decline in CIN activity. We thank the reviewer for this question and have incorporated this response into the discussion.

Reviewer 2, concern 3

3. It is not clear to me what the authors mean by this statement at the end of the introduction: “Importantly, SNr-projecting dMSNs in the direct pathway that mediate reinforcement also sent collaterals to inhibit CINs. Our data demonstrate that the inhibitory dMSN CIN connection mediates the reinforcement-induced reduction in cognitive flexibility.” What do the authors mean by “reinforcement” here? Are they contrasting these effects with reward? **(see Response#1 below)** Does cocaine induce reinforcement – some form of enhanced effect on dopamine neurons resulting in compulsive performance during reversal compared to sucrose that does not? It is simply not clear and needs to be given the prominence of this statement **(see Response#2 below)**.

Response #1: We appreciate the reviewer's question regarding the clarity of the statement in the introduction. We understand that the term "reinforcement" may be causing confusion, and we would like to clarify our intended meaning. In this context, we propose changing the phrase "reinforcement-induced reduction" to "reinforcer-induced reduction" instead of "reward-induced reduction." This distinction is important because rewards are generally perceived as pleasurable stimuli, while reinforcers directly influence the likelihood of a specific behavior being repeated in the future. Although rewards can potentially act as reinforcers, not all rewards are effective reinforcers. The following paragraph conveys the meaning of the last statement of the introduction.

Since reinforcers can affect the likelihood of future behaviors associated with them, exposure to reinforcers is more likely to increase dMSN activity, given that dMSNs contribute to the selection of actions for reinforcer-seeking and taking behavior (Cheng et al., 2017; Gerfen and Surmeier, 2011; Hellard et al., 2019; Ma et al., 2018). Moreover, in our study, we found that reinforcer-induced enhanced dMSN→CIN inhibitory transmission and subsequent suppression of CIN activity, leading to cognitive inflexibility. This suggests that the dMSN→CIN transmission mediates the reinforcer-induced reduction in cognitive flexibility. However, the degree of cognitive inflexibility should vary depending on the potency and strength of the primary reinforcer. We thank the reviewer for this question and have updated the statement in the introduction section.

Response #2: We acknowledge that cocaine-induced reinforcement could result in more compulsive performance during reversal learning than sucrose-induced reinforcement. This notion is consistent with a previous study demonstrating that cocaine but not sucrose self-administration produces persistent synaptic plasticity in the VTA (Chen et al., 2008). Please also see our Response #3 to Reviewer 2, Concern 2.

Reviewer 2, concern 4

4. The behavioral data are incorrectly analysed in a couple of places. The authors use independent t-tests to test devaluation effects in vehicle and cocaine groups in experiments in which they are comparing the size of those devaluation effects between the groups. The appropriate analysis is to use a mixed model ANOVA and include both groups in an F-test so as to test for a significant group x devaluation interaction. Without a reliable interaction the data cannot be interpreted.

Dior example: on P.6-7 they state: “However, the saline, but not the cocaine group, was sensitive to outcome devaluation after reversal training (saline: Fig. 1k left, $t(21) = 2.37$, $p < 0.05$; cocaine: Fig. 1k right, $t(23) = -0.750$, $p > 0.05$.” This problem is found in the analysis of the data in Figure 1 panels f, h, j, k; and Figure 7 panels e, and h. **(see Response #1 below)**

Discussion: The authors emphasize the effects of stimulant exposure on reversal; they find no effect of drug exposure on initial learning. However, this actually is not the common result. There have been assessments of drug exposure- and drug context exposure-induced effects on devaluation previously. For example: Nelson & Killcross (2006) *J. Neurosci*; Nelson & Killcross (2013) *Frontiers in Neuroscience*; Corbit et al (2014), *Neuropsychopharmacology*; Halbout et al, (2016) *Frontiers in Psychiatry*; Leong et al (2016) *Physiol Behav*; Furlong et al (2017) *addiction biol*; Jones et al, (2022) *learning and memory*. These reported effects are not effects on reversal but on devaluation conducted after initial training. They are not discussed or referenced here. What is the difference between the authors treatments and those previously reported? Is it a matter of dose or intensity of the effects of drug exposure **(see Response #2 below)** When does drug exposure affect initial learning and when does it only affect reversal? **(see Response #3 below)**

Response #1: We thank the reviewer for the correction. We have now reanalyzed the devaluation data for the three experiments using mixed model ANOVA followed by simple effects analysis in SPSS, results of which are presented in Table 2 below. Old statistics have been replaced with the new analysis in the manuscript.

Table 2. Revised Statistics

Fig.	Analysis	Test	Dep. Variable	Factor(s)	Statistic	P value
1f	Initial devaluation	Mixed model ANOVA	Lever Press rate (press/min)	Devaluation (valued/devalued; within-subject) Group (Saline/Cocaine; between subjects)	Main Effect (Devaluation): $F(1,16) = 67.376$ Group x Devaluation interaction: $F(1,16) = 0.386$	<0.001 0.543
		Simple-effects			Lever within Sal (Val vs Dev): $F(1,16) = 32.378$ Lever within Coc (Val vs Dev): $F(1,16) = 35.084$	<0.001 <0.001
1h	Second devaluation	Mixed model ANOVA	Lever Press rate (press/min)	Devaluation (valued/devalued; within-subject) Group (Saline/Cocaine; between subjects)	Main Effect (Devaluation): $F(1,16) = 29.762$ Group x Devaluation interaction: $F(1,16) = 7.368$	<0.001 0.015
		Simple-effects			Lever within Sal (Val vs Dev): $F(1,16) = 35.545$ Lever within Coc (Val vs Dev): $F(1,16) = 3.381$	<0.001 0.085
1j	Initial devaluation	Mixed model ANOVA	Lever Presses	Devaluation (valued/devalued; within-subject) Group (Saline/Cocaine; between subjects)	Main Effect (Devaluation): $F(1,21) = 19.780$ Group x Devaluation interaction: $F(1,21) = 0.044$	<0.001 0.837
		Simple-effects			Lever within Sal (Val vs Dev): $F(1,21) = 10.389$ Lever within Coc (Val vs Dev): $F(1,21) = 9.391$	0.004 0.006
1k	Second devaluation	Mixed model ANOVA	Lever Presses	Devaluation (valued/devalued; within-subject) Group (Saline/Cocaine; between subjects)	Main Effect (Devaluation): $F(1,21) = 0.400$ Group x Devaluation interaction: $F(1,21) = 6.665$	0.534 0.017
		Simple-effects			Lever within Sal (Val vs Dev): $F(1,21) = 4.949$ Lever within Coc (Val vs Dev): $F(1,21) = 1.987$	0.037 0.173
6e	Initial devaluation	Mixed model ANOVA	Lever Presses	Devaluation (valued/devalued; within-subject) Group (eYFP/eNpHR; between subjects)	Main Effect (Devaluation): $F(1,16) = 48.994$ Group x Devaluation interaction: $F(1,16) = 1.304$	<0.001 0.270
		Simple-effects			Lever within eYFP (Val vs Dev): $F(1,16) = 33.142$ Lever within eNpHR (Val vs Dev): $F(1,16) = 17.156$	<0.001 <0.001
6h	Second devaluation	Mixed model ANOVA	Lever Presses	Devaluation (valued/devalued; within-subject) Group (eYFP/eNpHR; between subjects)	Main Effect (Devaluation): $F(1,16) = 0.791$ Group x Devaluation interaction: $F(1,16) = 10.937$	0.387 0.004
		Simple-effects			Lever within eYFP (Val vs Dev): $F(1,16) = 8.805$ Lever within eNpHR (Val vs Dev): $F(1,16) = 2.923$	0.009 0.107

Response #2: We thank the reviewer for their valuable feedback and for providing a list of publications that examine the impact of drug exposure on the initial acquisition of goal-directed learning. We have summarized

the details regarding drug treatments, behavioral tasks employed, and obtained results for these and our study in Table 3 below.

As evident from the table, some of these studies demonstrate that drug exposure disrupted the initial learning of action-outcome associations (Corbit *et al.*, 2014; Nelson and Killcross, 2006; Nelson and Killcross, 2013), while others suggest that drug exposure did not disrupt the initial learning (Furlong *et al.*, 2017; Halbout *et al.*, 2016), which is consistent with our study. This discrepancy is unlikely to be due to differences in dose or intensity of drug exposure, as most of the studies employed a similar drug treatment intensity to ours (i.e., ~7 d of intraperitoneal injections before the behavioral task). There seems to be no direct relationship between the dose/drug used and the discrepancy in initial goal-directed learning.

However, we believe that the discrepancy could be attributed to the training schedule. Two training schedules are used in the listed references and our current study: random interval (RI) schedule and random ratio (RR) schedule. When the RI schedule was employed, previous studies found that drug (cocaine or amphetamine) exposure disrupted goal-directed learning after initial training (Corbit *et al.*, 2014; Nelson and Killcross, 2006; Nelson and Killcross, 2013). In contrast, when the RR schedule was used, two previous studies observed that drug exposure did not impair goal-directed learning after initial training (Furlong *et al.*, 2017; Halbout *et al.*, 2016), which is consistent with our findings (Fig. 1). Notably, in one study, both RR and RI schedules were used in separate groups, and it was found that RR training, but not RI training, preserved goal-directed learning (Jones *et al.*, 2022). Additionally, when a fixed-ratio schedule was used, short (1 h/day) but not long (6 hr/day) cocaine intravenous self-administration (IVSA) training preserved goal-directed learning (Leong *et al.*, 2016). These findings suggest that prior drug exposure disrupts instrumental learning of tasks that employ the RI schedule but not the RR schedule.

The reason for drug exposure impacting schedule-specific instrumental learning remains unclear. RR schedules, which provide reinforcement rewards after a random number of responses, encourage animals to actively seek and engage in action (lever pressing)-outcome (reward delivery) association learning to obtain the reward, thus resulting in a higher lever-press rate than the RI schedule (Dickinson *et al.*, 1983). Therefore, goal-directed behavior under RR training is more likely to be affected by the **current** value of the reward, rather than drug exposure history. In contrast, RI schedules, which provide reinforcement rewards after a random amount of **time**, tend to promote habitual behaviors (Dickinson *et al.*, 1983). However, it is unclear why habitual behavior under RI training is more likely to be influenced by drug exposure history and exploring the underlying mechanism is out of scope of this manuscript. We thank the reviewer for raising this question and have incorporated parts of the above response into the discussion section of our manuscript.

Table 3. Drug-treatment, behavior task employed and obtained results

Papers	Drug treatment	Task	Result (Drug-exposed group)
Present study	Cocaine (15 mg/kg, i.p., 7 d)	Two action-outcome association (food and sucrose, for rats; grain and purified food for mice); RR schedule; 11-d instrumental training	Initial goal-directed learning preserved , reversal goal-directed learning disrupted .
(Halbout et al. , 2016)	Cocaine (15/30 mg/kg, i.p., 15 d); 30-d before training.	Two action-outcome association (grain and chocolate-based purified food); RR schedule; 10-d instrumental training	Initial goal-directed learning preserved
(Furlong et al. , 2017)	Methamphetamine (1/2 mg/kg, i.p., 7 injections in 14 d)	Two action-outcome association (food and sucrose); RR schedule; 7-d instrumental training	Initial goal-directed learning preserved in saline- but not Methamphetamine -paired context, suggesting that contextual cues interfere with the expression of goal-directed action selection.
(Nelson and Killcross, 2006)	Amphetamine (2 mg/kg, i.p., 7 d)	Single action-outcome association (sucrose or maltodextrin solution); RI schedule; alternative reinforcer provided separately at random time schedule; 5-d instrumental training	Exp 1: Initial goal-directed learning disrupted . Exp 2: i.p. inj. after training; result: initial goal-directed learning preserved.
(Nelson and Killcross, 2013)	Amphetamine (2 mg/kg, i.p., 7 d)	Single action-outcome association (sucrose or maltodextrin solution); RI schedule; 5-d instrumental training	Initial goal-directed learning disrupted .

(Corbit et al. , 2014)	Cocaine (30 mg/kg, i.p., 6 d)	Single action-outcome association (pellets or sucrose solution); RI schedule; 8-d or 16-d instrumental training	Initial goal-directed learning preserved after 8 days but not after 16 days of training.
(Jones et al. , 2022)	Cocaine (0.5 mg/kg/infusion, IVSA seeking-taking chained schedule, > 18 sessions)	One group on RR schedule of reinforcement; another on RI schedule of reinforcement.	RR schedule preserved initial goal-directed learning whereas RI schedule disrupted learning.
(Leong et al. , 2016)	Cocaine (0.2 mg/50 µl bolus)	Instrumental IVSA training (Short access: 1 h/d; Long access: 6 h/d); Cocaine-induced outcome devaluation.	Initial goal-directed learning preserved in short access but not in long access rats.

Response #3: We appreciate the reviewer’s question. In a reversal learning paradigm, subjects already possess prior experience with the task and understand the necessary actions to obtain a reward. An example of this is a lever-pressing task, where mice are initially trained to press one lever for food and another for sucrose. After mastering this association, the contingencies are reversed, requiring the mice to learn and adapt to the new associations (Bradfield *et al.*, 2013). Although there are some similarities, the neural mechanisms involved in initial learning and reversal learning are distinct. Research has shown that ablation of DMS CINs impairs reversal learning without affecting initial learning (Bradfield *et al.*, 2013; Matamales *et al.*, 2016). This finding suggests that cholinergic activity is necessary for reversal learning, but not for initial learning. Consequently, drug exposure, which inhibits cholinergic activity as reported in this manuscript, only impacts reversal learning and not initial learning. We have revised the Discussion section to incorporate these points and provide a clearer explanation of the relationship between drug exposure, initial learning, and reversal learning.

Reviewer 3, Concern 1

The authors have done a very nice job of addressing almost all of my concerns, including a significant number of additional experiments that have added to an already interesting and impactful body of work.

*It is appreciated that the authors provided references to previous manuscripts that have used this behavioral paradigm. It is also appreciated that cognitive flexibility is indeed loosely defined as the ability of an animal to adjust to environmental changes. However, it is worth noting that the provided references almost exclusively refer to this task as changes in action-outcome or goal-directed learning and not specifically cognitive flexibility (with the exception of Matamales *et al.*, 2016). This is also an important point when citing this work. For example saying CINs regulate cognitive flexibility and then citing those references.*

While this task is not generally used to measure changes in cognitive flexibility as it is often referred to in the literature versus something like an attentional set-shifting task, I do feel like the behavior construct "loosly" falls within a cognitive construct. I think it would be helpful to be more specific about the type of cognitive construct being tested at the very least in the title of the manuscript and then make an inference as to what that aligns with (i.e. cognitive flexibility) in the main text. For example, it would seem to more appropriate for the title to read "Drug Reinforcement impairs flexibility of goal-directed actions (or learning) by inhibiting striatal cholinergic interneurons in mice. Additionally, the authors can note that they will refer to alterations in this learning as cognitive flexibility. The distinction is small but important in the eyes of this reviewer.

Response: We appreciate the reviewer's suggestion to be more specific about the cognitive construct being tested in the title of our manuscript. We acknowledge that the task used in our study primarily examines changes in action-outcome or goal-directed learning, as referenced in most of our citations, with the exception of Matamales *et al.* (2016). We concur that "cognitive flexibility" is a broader concept encompassing an individual's ability to adapt their thinking and behavior in response to environmental or task changes. It involves the capacity to switch between mental sets, strategies, or perspectives when encountering new information or situations.

Despite "cognitive flexibility" being a broad term that includes various cognitive processes, we believe it is suitable for our study's title for the following reasons:

- The behavioral task design in our manuscript was first introduced by Dr. Bernard Balleine's group in 2013 (Bradfield et al., 2013). Using the same task design, his group recently published an article titled "CRF-receptor1 modulation of the dopamine projection to prelimbic cortex facilitates cognitive flexibility after acute and chronic stress" (Mor et al., 2022). Notably, the first sentence in their discussion states, "The capacity to update action-outcome encoding is essential for goal-directed behavior to remain adaptive in a changing environment, a core component of cognitive flexibility," implying that flexibility of goal-directed learning plays a significant role in regulating cognitive flexibility.
- Cognitive flexibility is a higher-order cognitive function, and no single experiment can uniquely assess all its dimensions. In the Nature Communications journal, the term "cognitive flexibility" has been used in various contexts, including tasks involving attentional set-shifting, reversal learning, or goal-directed learning (Table 4 below).

Table 4. List of articles published in Nature Communications that used "cognitive flexibility"

References	Title	Behavioral Task
(Kosillo et al., 2019)	Tsc1-mTORC1 signaling controls striatal dopamine release and cognitive flexibility	Four-choice odor-based reversal task
(Xu et al., 2019)	Caspase-2 promotes AMPA receptor internalization and cognitive flexibility via mTORC2-AKT-GSK3 β signaling	Morris water-maze task
(Eales et al., 2014)	The MK2/3 cascade regulates AMPAR trafficking and cognitive flexibility	Modified Barnes-maze task
(Hu et al., 2020)	Activity-dependent isomerization of Kv4.2 by Pin1 regulates cognitive flexibility	Morris water-maze task

- After conducting a PubMed search, we found 839 articles with "cognitive flexibility" in their titles, suggesting the term is widely recognized and extensively used in scientific literature. To effectively communicate our research and reach a broader audience, it would be advantageous to use this same terminology in our work. By employing this recognized term, we can increase the visibility and accessibility of our work to other researchers and professionals interested in this topic.

Considering these points, we believe our study contributes to understanding cognitive flexibility, specifically from the aspect of goal-directed learning. We propose retaining the term "cognitive flexibility" in our title, understanding that our task particularly addresses the flexibility of goal-directed learning. To address the reviewer's concern, we could amend the title from "Drug Reinforcement Impairs **Cognitive Flexibility** by Inhibiting Striatal Cholinergic Interneurons" to "Drug Reinforcement Impairs **Flexibility of Goal-directed Learning** by Inhibiting Striatal Cholinergic Interneurons". We do not think it is appropriate to specify "in mice" in the title because our studies were conducted in both mice and rats. We would like to leave the final decision regarding the use of terms "cognitive flexibility" or "flexibility of goal-directed learning" in the title to the editor's discretion. We hope this response addresses the reviewer's concerns and provides a balanced representation of the cognitive construct being tested in our work.

In addition, we made the following changes in the manuscript:

- In the abstract, we modified the statement to read: "Furthermore, chemogenetic and time-locked optogenetic inhibition of DMS CINs suppressed flexibility of goal-directed behavior in an instrumental reversal learning task", instead of "Furthermore, chemogenetic and time-locked optogenetic inhibition of DMS CINs suppressed cognitive flexibility in an instrumental reversal learning task."
- In the introduction, we added the sentence: "The ability to modify action-outcome associations is critical for maintaining adaptive goal-directed behavior in a dynamic environment, which is a fundamental aspect of cognitive flexibility." (Mor et al., 2022)

- We did not include “cognitive” in our running title. The running title reads, “Reinforcement impairs flexibility by inhibiting cholinergic neurons.”
- We added “in rodents” in the abstract to specify the subject type.

Other changes:

1. All changes in the main manuscript use a blue font

References:

- Bradfield, L.A., Bertran-Gonzalez, J., Chieng, B., and Balleine, B.W. (2013). The thalamostriatal pathway and cholinergic control of goal-directed action: interlacing new with existing learning in the striatum. *Neuron* 79, 153-166
- Chawla, A., Cordner, Z.A., Boersma, G., and Moran, T.H. (2017). Cognitive impairment and gene expression alterations in a rodent model of binge eating disorder. *Physiol Behav* 180, 78-90
- Chen, B.T., Bowers, M.S., Martin, M., Hopf, F.W., Guillory, A.M., Carelli, R.M., Chou, J.K., and Bonci, A. (2008). Cocaine but not natural reward self-administration nor passive cocaine infusion produces persistent LTP in the VTA. *Neuron* 59, 288-297
- Cheng, Y., Huang, C.C.Y., Ma, T., Wei, X., Wang, X., Lu, J., and Wang, J. (2017). Distinct synaptic strengthening of the striatal direct and indirect pathways drives alcohol consumption. *Biological psychiatry* 81, 918-929
- Corbit, L.H., Chieng, B.C., and Balleine, B.W. (2014). Effects of repeated cocaine exposure on habit learning and reversal by N-acetylcysteine. *Neuropsychopharmacology* 39, 1893-1901
- Cui, G., Jun, S.B., Jin, X., Pham, M.D., Vogel, S.S., Lovinger, D.M., and Costa, R.M. (2013). Concurrent activation of striatal direct and indirect pathways during action initiation. *Nature* 494, 238-242
- Dickinson, A., Nicholas, D.J., and Adams, C.D. (1983). The Effect of the Instrumental Training Contingency on Susceptibility to Reinforcer Devaluation. *Q J Exp Psychol-B* 35, 35-51
- Eales, K.L., Palygin, O., O'Loughlin, T., Rasooli-Nejad, S., Gaestel, M., Müller, J., Collins, D.R., Pankratov, Y., and Corrêa, S.A. (2014). The MK2/3 cascade regulates AMPAR trafficking and cognitive flexibility. *Nat Commun* 5, 4701
- Fleming, W., Lee, J., Briones, B.A., Bolkan, S.S., and Witten, I.B. (2022). Cholinergic interneurons mediate cocaine extinction in male mice through plasticity across medium spiny neuron subtypes. *Cell reports* 39, 110874
- Furlong, T.M., Supit, A.S., Corbit, L.H., Killcross, S., and Balleine, B.W. (2017). Pulling habits out of rats: adenosine 2A receptor antagonism in dorsomedial striatum rescues meth-amphetamine-induced deficits in goal-directed action. *Addict Biol* 22, 172-183
- Gerfen, C.R., and Surmeier, D.J. (2011). Modulation of striatal projection systems by dopamine. *Annu Rev Neurosci* 34, 441-466
- Halbout, B., Liu, A.T., and Ostlund, S.B. (2016). A Closer Look at the Effects of Repeated Cocaine Exposure on Adaptive Decision-Making under Conditions That Promote Goal-Directed Control. *Front Psychiatry* 7, 44

- Hellard, E.R., Binette, A., Zhuang, X., Lu, J., Ma, T., Jones, B., Williams, E., Jayavelu, S., and Wang, J. (2019). Optogenetic control of alcohol-seeking behavior via the dorsomedial striatal circuit. *Neuropharmacology* *155*, 89-97
- Hu, J.H., Malloy, C., Tabor, G.T., Gutzmann, J.J., Liu, Y., Abebe, D., Karlsson, R.M., Durell, S., Cameron, H.A., and Hoffman, D.A. (2020). Activity-dependent isomerization of Kv4.2 by Pin1 regulates cognitive flexibility. *Nat Commun* *11*, 1567
- Jentsch, J.D., Olausson, P., De La Garza, R., 2nd, and Taylor, J.R. (2002). Impairments of reversal learning and response perseveration after repeated, intermittent cocaine administrations to monkeys. *Neuropsychopharmacology* *26*, 183-190
- Jones, B.O., Cruz, A.M., Kim, T.H., Spencer, H.F., and Smith, R.J. (2022). Discriminating goal-directed and habitual cocaine seeking in rats using a novel outcome devaluation procedure. *Learn Mem* *29*, 447-457
- Kash, T.L., Baucum, A.J., 2nd, Conrad, K.L., Colbran, R.J., and Winder, D.G. (2009). Alcohol exposure alters NMDAR function in the bed nucleus of the stria terminalis. *Neuropsychopharmacology* *34*, 2420-2429
- Kosillo, P., Doig, N.M., Ahmed, K.M., Agopyan-Miu, A., Wong, C.D., Conyers, L., Threlfell, S., Magill, P.J., and Bateup, H.S. (2019). Tsc1-mTORC1 signaling controls striatal dopamine release and cognitive flexibility. *Nat Commun* *10*, 5426
- Lee, J.H., Ribeiro, E.A., Kim, J., Ko, B., Kronman, H., Jeong, Y.H., Kim, J.K., Janak, P.H., Nestler, E.J., Koo, J.W., and Kim, J.H. (2020). Dopaminergic Regulation of Nucleus Accumbens Cholinergic Interneurons Demarcates Susceptibility to Cocaine Addiction. *Biol Psychiatry* *88*, 746-757
- Leong, K.C., Berini, C.R., Ghee, S.M., and Reichel, C.M. (2016). Extended cocaine-seeking produces a shift from goal-directed to habitual responding in rats. *Physiol Behav* *164*, 330-335
- Leyrer-Jackson, J.M., Hood, L.E., and Olive, M.F. (2021). Drugs of Abuse Differentially Alter the Neuronal Excitability of Prefrontal Layer V Pyramidal Cell Subtypes. *Front Cell Neurosci* *15*, 703655
- Li, D.C., Dighe, N.M., Barbee, B.R., Pitts, E.G., Kochoian, B., Blumenthal, S.A., Figueroa, J., Leong, T., and Gourley, S.L. (2022). A molecularly integrated amygdalo-fronto-striatal network coordinates flexible learning and memory. *Nat Neurosci* *25*, 1213-1224
- Liu, Q.S., Pu, L., and Poo, M.M. (2005). Repeated cocaine exposure in vivo facilitates LTP induction in midbrain dopamine neurons. *Nature* *437*, 1027-1031
- Lovinger, D.M. (2010). Neurotransmitter roles in synaptic modulation, plasticity and learning in the dorsal striatum. *Neuropharmacology* *58*, 951-961
- Lovinger, D.M., and Kash, T.L. (2015). Mechanisms of Neuroplasticity and Ethanol's Effects on Plasticity in the Striatum and Bed Nucleus of the Stria Terminalis. *Alcohol research : current reviews* *37*, 109-124
- Ma, T., Cheng, Y., Roltsch Hellard, E., Wang, X., Lu, J., Gao, X., Huang, C.C.Y., Wei, X., Ji, J., and Wang, J. (2018). Bidirectional and long-lasting control of alcohol-seeking behavior by corticostriatal LTP and LTD. *Nat Neurosci* *21*, 373-383
- Ma, T., Huang, Z., Xie, X., Cheng, Y., Zhuang, X., Childs, M.J., Gangal, H., Wang, X., Smith, L.N., Smith, R.J., et al. (2022). Chronic alcohol drinking persistently suppresses thalamostriatal excitation of cholinergic neurons to impair cognitive flexibility. *The Journal of clinical investigation* *132*, e154969
- Matamales, M., Skrbis, Z., Hatch, R.J., Balleine, B.W., Gotz, J., and Bertran-Gonzalez, J. (2016). Aging-related dysfunction of striatal cholinergic interneurons produces conflict in action selection. *Neuron* *90*, 362-373

- McCool, B.A. (2011). Ethanol modulation of synaptic plasticity. *Neuropharmacology* 61, 1097-1108
- McCracken, C.B., and Grace, A.A. (2013). Persistent cocaine-induced reversal learning deficits are associated with altered limbic cortico-striatal local field potential synchronization. *J Neurosci* 33, 17469-17482
- Melugin, P.R., Nolan, S.O., and Siciliano, C.A. (2021). Bidirectional causality between addiction and cognitive deficits. *Int Rev Neurobiol* 157, 371-407
- Mor, D., Becchi, S., Bowring, J., Tsoukalas, M., and Balleine, B.W. (2022). CRF-receptor1 modulation of the dopamine projection to prelimbic cortex facilitates cognitive flexibility after acute and chronic stress. *Neurobiol Stress* 16, 100424
- Mueller, L.E., Sharpe, M.J., Stalnaker, T.A., Wikenheiser, A.M., and Schoenbaum, G. (2021). Prior Cocaine Use Alters the Normal Evolution of Information Coding in Striatal Ensembles during Value-Guided Decision-Making. *J Neurosci* 41, 342-353
- Nelson, A., and Killcross, S. (2006). Amphetamine exposure enhances habit formation. *J Neurosci* 26, 3805-3812
- Nelson, A.J., and Killcross, S. (2013). Accelerated habit formation following amphetamine exposure is reversed by D1, but enhanced by D2, receptor antagonists. *Front Neurosci* 7, 76
- Ortinski, P.I., Turner, J.R., and Pierce, R.C. (2013). Extrasynaptic targeting of NMDA receptors following D1 dopamine receptor activation and cocaine self-administration. *J Neurosci* 33, 9451-9461
- Perez Diaz, M., Wilson, M.E., and Howell, L.L. (2019). Effects of long-term high-fat food or methamphetamine intake and serotonin 2C receptors on reversal learning in female rhesus macaques. *Neuropsychopharmacology* 44, 478-486
- Saund, J., Dautan, D., Rostron, C., Urcelay, G.P., and Gerdjikov, T.V. (2017). Thalamic inputs to dorsomedial striatum are involved in inhibitory control: evidence from the five-choice serial reaction time task in rats. *Psychopharmacology (Berl)* 234, 2399-2407
- Shan, Q., Ge, M., Christie, M.J., and Balleine, B.W. (2014). The acquisition of goal-directed actions generates opposing plasticity in direct and indirect pathways in dorsomedial striatum. *J Neurosci* 34, 9196-9201
- Shen, W., Plotkin, J.L., Francardo, V., Ko, W.K., Xie, Z., Li, Q., Fieblinger, T., Wess, J., Neubig, R.R., Lindsley, C.W., et al. (2015). M4 muscarinic receptor signaling ameliorates striatal plasticity deficits in models of L-DOPA-induced dyskinesia. *Neuron* 88, 762-773
- Smith, R.J., and Laiks, L.S. (2018). Behavioral and neural mechanisms underlying habitual and compulsive drug seeking. *Prog Neuropsychopharmacol Biol Psychiatry* 87, 11-21
- Stalnaker, T.A., Roesch, M.R., Calu, D.J., Burke, K.A., Singh, T., and Schoenbaum, G. (2007). Neural correlates of inflexible behavior in the orbitofrontal-amygdalar circuit after cocaine exposure. *Ann N Y Acad Sci* 1121, 598-609
- Stalnaker, T.A., Takahashi, Y., Roesch, M.R., and Schoenbaum, G. (2009). Neural substrates of cognitive inflexibility after chronic cocaine exposure. *Neuropharmacology* 56 Suppl 1, 63-72
- Thiebaud, N., Johnson, M.C., Butler, J.L., Bell, G.A., Ferguson, K.L., Fadool, A.R., Fadool, J.C., Gale, A.M., Gale, D.S., and Fadool, D.A. (2014). Hyperlipidemic diet causes loss of olfactory sensory neurons, reduces olfactory discrimination, and disrupts odor-reversal learning. *J Neurosci* 34, 6970-6984
- Valjent, E., Corvol, J.C., Pages, C., Besson, M.J., Maldonado, R., and Caboche, J. (2000). Involvement of the extracellular signal-regulated kinase cascade for cocaine-rewarding properties. *J Neurosci* 20, 8701-8709

- Valjent, E., Pascoli, V., Svenningsson, P., Paul, S., Enslen, H., Corvol, J.C., Stipanovich, A., Caboche, J., Lombroso, P.J., Nairn, A.C., et al. (2005). Regulation of a protein phosphatase cascade allows convergent dopamine and glutamate signals to activate ERK in the striatum. *Proc Natl Acad Sci U S A* *102*, 491-496
- Wang, J., Ben Hamida, S., Darcq, E., Zhu, W., Gibb, S.L., Lanfranco, M.F., Carnicella, S., and Ron, D. (2012). Ethanol-mediated facilitation of AMPA receptor function in the dorsomedial striatum: implications for alcohol drinking behavior. *J Neurosci* *32*, 15124-15132
- Wang, J., Carnicella, S., Phamluong, K., Jeanblanc, J., Ronesi, J.A., Chaudhri, N., Janak, P.H., Lovinger, D.M., and Ron, D. (2007). Ethanol induces long-term facilitation of NR2B-NMDA receptor activity in the dorsal striatum: implications for alcohol drinking behavior. *J Neurosci* *27*, 3593-3602
- Wang, J., Ishikawa, M., Yang, Y., Otaka, M., Kim, J.Y., Gardner, G.R., Stefanik, M.T., Milovanovic, M., Huang, Y.H., Hell, J.W., et al. (2018). Cascades of Homeostatic Dysregulation Promote Incubation of Cocaine Craving. *J Neurosci* *38*, 4316-4328
- Wang, J., Lanfranco, M.F., Gibb, S.L., Yowell, Q.V., Carnicella, S., and Ron, D. (2010). Long-lasting adaptations of the NR2B-containing NMDA receptors in the dorsomedial striatum play a crucial role in alcohol consumption and relapse. *J Neurosci* *30*, 10187-10198
- West, E.A., Niedringhaus, M., Ortega, H.K., Haake, R.M., Frohlich, F., and Carelli, R.M. (2021a). Noninvasive Brain Stimulation Rescues Cocaine-Induced Prefrontal Hypoactivity and Restores Flexible Behavior. *Biol Psychiatry* *89*, 1001-1011
- West, E.A., Niedringhaus, M., Ortega, H.K., Haake, R.M., Frohlich, F., and Carelli, R.M. (2021b). Noninvasive Brain Stimulation Rescues Cocaine-Induced Prefrontal Hypoactivity and Restores Flexible Behavior. *Biol Psychiatry*, *89*(10):1001-1011
- Wills, T.A., Klug, J.R., Silberman, Y., Baucum, A.J., Weitlauf, C., Colbran, R.J., Delpire, E., and Winder, D.G. (2012). GluN2B subunit deletion reveals key role in acute and chronic ethanol sensitivity of glutamate synapses in bed nucleus of the stria terminalis. *Proc Natl Acad Sci U S A* *109*, E278-287
- Xia, J., Meyers, A.M., and Beeler, J.A. (2017). Chronic Nicotine Alters Corticostriatal Plasticity in the Striatopallidal Pathway Mediated By NR2B-Containing Silent Synapses. *Neuropsychopharmacology* *42*, 2314-2324
- Xu, Z.X., Tan, J.W., Xu, H., Hill, C.J., Ostrovskaya, O., Martemyanov, K.A., and Xu, B. (2019). Caspase-2 promotes AMPA receptor internalization and cognitive flexibility via mTORC2-AKT-GSK3 β signaling. *Nat Commun* *10*, 3622
- Yamamoto, D.J., and Zahner, N.R. (2012). Differences in rat dorsal striatal NMDA and AMPA receptors following acute and repeated cocaine-induced locomotor activation. *PLoS One* *7*, e37673
- Zucca, S., Zucca, A., Nakano, T., Aoki, S., and Wickens, J. (2018). Pauses in cholinergic interneuron firing exert an inhibitory control on striatal output in vivo. *Elife* *7*,